# The age of *Homo naledi* and associated sediments in the Rising Star Cave, South Africa

Paul HGM Dirks[1,2]*, Eric M Roberts[1,2], Hannah Hilbert-Wolf[1], Jan D Kramers[3], John Hawks[2,4], Anthony Dosseto[5], Mathieu Duval[6,7], Marina Elliott[2], Mary Evans[8], Rainer Grün[6,9], John Hellstrom[10], Andy IR Herries[11], Renaud Joannes-Boyau[12], Tebogo V Makhubela[3], Christa J Placzek[1], Jessie Robbins[1], Carl Spandler[1], Jelle Wiersma[1], Jon Woodhead[10], Lee R Berger[2]

[1]Department of Geoscience, James Cook University, Townsville, Australia; [2]Evolutionary Studies Institute and the National Centre for Excellence in PalaeoSciences, University of the Witwatersrand, Wits, South Africa; [3]Department of Geology, University of Johannesburg, Johannesburg, South Africa; [4]Department of Anthropology, University of Wisconsin-Madison, Madison, United States; [5]School of Earth and Environmental Sciences, University of Wollongong, Wollongong, Australia; [6]Australian Research Centre for Human Evolution, Environmental Futures Research Institute, Griffith University, Nathan, Australia; [7]Geochronology, Centro Nacional de Investigación sobre la Evolución Humana (CENIEH), Burgos, Spain; [8]School of Geosciences, University of the Witwatersrand, Wits, South Africa; [9]Research School of Earth Sciences, The Australian National University, Canberra, Australia; [10]School of Earth Sciences, The University of Melbourne, Parkville, Australia; [11]The Australian Archaeomagnetism Laboratory, Department of Archaeology and History, La Trobe University, Melbourne, Australia; [12]Geoarchaeology and Archaeometry Research Group, Department of GeoScience, Southern Cross University, Lismore, Australia

*For correspondence: paul.dirks@jcu.edu.au

**Competing interests:** The authors declare that no competing interests exist.

**Abstract** New ages for flowstone, sediments and fossil bones from the Dinaledi Chamber are presented. We combined optically stimulated luminescence dating of sediments with U-Th and palaeomagnetic analyses of flowstones to establish that all sediments containing *Homo naledi* fossils can be allocated to a single stratigraphic entity (sub-unit 3b), interpreted to be deposited between 236 ka and 414 ka. This result has been confirmed independently by dating three *H. naledi* teeth with combined U-series and electron spin resonance (US-ESR) dating. Two dating scenarios for the fossils were tested by varying the assumed levels of $^{222}$Rn loss in the encasing sediments: a maximum age scenario provides an average age for the two least altered fossil teeth of 253 +82/–70 ka, whilst a minimum age scenario yields an average age of 200 +70/–61 ka. We consider the maximum age scenario to more closely reflect conditions in the cave, and therefore, the true age of the fossils. By combining the US-ESR maximum age estimate obtained from the teeth, with the U-Th age for the oldest flowstone overlying *Homo naledi* fossils, we have constrained the depositional age of *Homo naledi* to a period between 236 ka and 335 ka. These age results demonstrate that a morphologically primitive hominin, *Homo naledi,* survived into the later parts of the Pleistocene in Africa, and indicate a much younger age for the *Homo naledi* fossils than have previously been hypothesized based on their morphology.

**eLife digest** Species of ancient humans and the extinct relatives of our ancestors are typically described from a limited number of fossils. However, this was not the case with *Homo naledi*. More than 1500 fossils representing at least 15 individuals of this species were unearthed from the Rising Star cave system in South Africa between 2013 and 2014. Found deep underground in the Dinaledi Chamber, the *H. naledi* fossils are the largest collection of a single species of an ancient human-relative discovered in Africa.

After the discovery was reported, a number of questions still remained. Not least among these questions was: how old were the fossils? The material was undated, and predictions ranged from anywhere between 2 million years old and 100,000 years old. *H. naledi* shared several traits with the most primitive of our ancient relatives, including its small brain. As a result, many scientists guessed that *H. naledi* was an old species in our family tree, and possibly one of the earliest species to evolve in the genus *Homo*.

Now, Dirks et al. – who include many of the researchers who were involved in the discovery of *H. naledi* – report that the fossils are most likely between 236,000 and 335,000 years old. These dates are based on measuring the concentration of radioactive elements, and the damage caused by these elements (which accumulates over time), in three fossilized teeth, plus surrounding rock and sediments from the cave chamber. Importantly, the most crucial tests were carried out at independent laboratories around the world, and the scientists conducted the tests without knowing the results of the other laboratories. Dirks et al. took these extra steps to make sure that the results obtained were reproducible and unbiased.

The estimated dates are much more recent than many had predicted, and mean that *H. naledi* was alive at the same time as the earliest members of our own species – which most likely evolved between 300,000 and 200,000 years ago. These new findings demonstrate why it can be unwise to try to predict the age of a fossil based only on its appearance, and emphasize the importance of dating specimens via independent tests. Finally in two related reports, Berger et al. suggest how a primitive-looking species like *H. naledi* survived more recently than many would have predicted, while Hawks et al. describe the discovery of more *H. naledi* fossils from a separate chamber in the same cave system.

## Introduction

The fossil assemblage attributed to *Homo naledi* from the Rising Star Cave in the Cradle of Humankind, UNESCO World Heritage Area, South Africa (CoH) (*Berger et al., 2015*), represents one of the richest and most unusual taphonomic assemblages yet discovered in the hominin fossil record (*Dirks et al., 2015*). The remains are exceptionally well preserved and represent the largest collection of fossils from a single primitive hominin species ever discovered in Africa. The *H. naledi* fossils occur without a direct association with non-hominin macrofossil remains, and are found deep inside the difficult to access U.W.101-Dinaledi Chamber (*Dirks et al., 2015*). The Dinaledi Chamber is characterised by a sedimentary environment that is geochemically and sedimentologically distinct from the rest of the Rising Star Cave (*Dirks et al., 2015*), and the fossiliferous deposit it contains is profoundly different from other known hominin-bearing cave assemblages in the CoH (e.g., *Reynolds and Kibii, 2011*; *Dirks et al., 2010*; *Pickering et al., 2011a*; *Dirks and Berger, 2013*; *Bruxelles et al., 2014*). The fossils occur as a dense bone accumulation in mostly unconsolidated muddy sediment that largely originated from within the cave through weathering of the dolomite host rock (*Dirks et al., 2015*). The fossils have not been dated until now.

In this paper we present results of uranium-thorium (U-Th) disequilibrium, electron spin resonance (ESR), radiocarbon, and optically stimulated luminescence (OSL) dating in combination with palaeomagnetic analyses, to provide ages for the fossils and surrounding deposits in the Dinaledi Chamber, and build upon the geological context described in *Dirks et al. (2015)*. Dates acquired via U-Th and ESR techniques were obtained using a double blind approach for each technique to ensure robust,

reproducible results, with each laboratory using their own analytical and computational approach. Approaches taken by each laboratory that contributed to this paper are described in detail in the methodology section.

The age of the hominins in the Dinaledi Chamber has implications for our understanding of the mode and tempo of the morphological evolution of hominins (*Hawks and Berger, 2016*), raising questions about evolutionary stasis and the role of refugia. The results challenge our ability to associate given hominin species to specific cultures and behaviours in the past. These issues are discussed in greater detail in an accompanying paper (*Berger et al., 2017*).

## Geological setting

The caves in the Cradle of Humankind (CoH), South Africa have yielded rich fossil assemblages of late Pliocene to early Pleistocene age, which include a range of hominin species (*A. africanus, A. prometheus, A. sediba, P. robustus, H. ergaster, H. naledi* and early *Homo*) and associated mammals, reptiles, and birds (e.g., *Vrba, 1975*, *1995*; *Brain, 1993*; *Tobias, 2000*; *Berger et al., 2010*, *2015*). For the past 3 million years, hominin-bearing deposits in caves formed in broadly similar settings, involving debris cone accumulations near cave openings (*Partridge, 1973*; *Wilkinson, 1985*; *Brain, 1993*; *Pickering et al., 2007*; *de Ruiter et al., 2009*; *Dirks and Berger, 2013*; *Herries and Adams, 2013*; *Dirks et al., 2010*, *2016b*; *Bruxelles et al., 2014*; *Stratford et al., 2014*), with deposits cemented by carbonate-rich waters dripping from cave ceilings (e.g., *Wilkinson, 1985*; *Pickering et al., 2011b*). In contrast to all other hominin deposits in the CoH, the deposits that host *H. naledi* in Rising Star Cave are composed of largely unconsolidated, mud-clast breccia in a mud matrix with no evidence of coarse clastic sediment being carried in by water flow. This suggests a different depositional regime and timing for the sediments and the fossils (*Dirks et al., 2015*, *Dirks et al., 2016a*).

Rising Star Cave is situated in the Bloubank River valley, 2.2 km W of Sterkfontein Cave. The cave system comprises several kilometres of mapped passageways (*Figure 1a*) that are stratigraphically bound to a 20–30 m-thick, chert-poor dolomite horizon capped by a 1–1.3 m-thick chert unit that forms the roof to the cave system (*Dirks et al., 2015*). Geological mapping and laser-theodolite surveys indicate that this roof is intact and not penetrated by significant shafts that open to surface (*Dirks et al., 2015*; *Kruger et al., 2016*). The broader geological setting of the cave is discussed in *Dirks et al. (2015)*, (*Dirks et al., 2016a*).

The Dinaledi Chamber, which contains most of the fossils of *H. naledi,* is ~30 m below surface and ~80 m in a straight line from the nearest present-day opening to the surface (*Figure 1a*). The main cavity forming the Dinaledi Chamber is ~15 m long with variable widths not exceeding 2.5 meters (*Figure 1b*), and expands near the intersection with a crosscutting passage, which is the location of the main excavation site to date (*Figure 1b*). There is no evidence that the present entrance into the Dinaledi Chamber has significantly changed since the deposition of the fossil hominins, with sediment accumulating mostly near the current access point (*Dirks et al., 2015*, *Dirks et al., 2016a*; *Figure 2*). Samples for dating were collected from the various flowstone horizons and stratigraphic units exposed in the Dinaledi Chamber (*Figures 1b*, *2*, *3*, *4* and *5*) as well as from fossil material itself (*Figures 4*, *6* and *7*).

## Lithologic and stratigraphic context for dating

The Dinaledi Chamber contains deposits of fine-grained, muddy sediments intercalated with flowstone drapes. The sediments include various types of orange, laminated mudstone and mud clast breccia distributed across three broad lithostratigraphic units (Units 1, 2 and 3; *Dirks et al., 2015*) that filled parts of the chamber over time. Based on variations in sediment composition, fossil content and/or stratigraphic position of each unit, we have divided Unit 1 into sub-units 1a, 1b and 1c, and Unit 3 into sub-units 3a and 3b, to more precisely define the stratigraphic packages targeted for dating (*Figure 2*). The units are separated by erosional unconformities or flowstone intercalations, but do not all necessarily occur in direct contact with one another due to the complex nature of caves as depositional systems (e.g., *Brain, 1993*; *Martini et al., 2003*). In addition, apart from sediments accumulating along the floor of the cave chamber, sediment in the form of orange mud deposits also accumulated inside fractures and along ledges higher up in the Dinaledi Chamber (*Figure 2b*), where it formed as a result of the combined effect of in situ weathering and

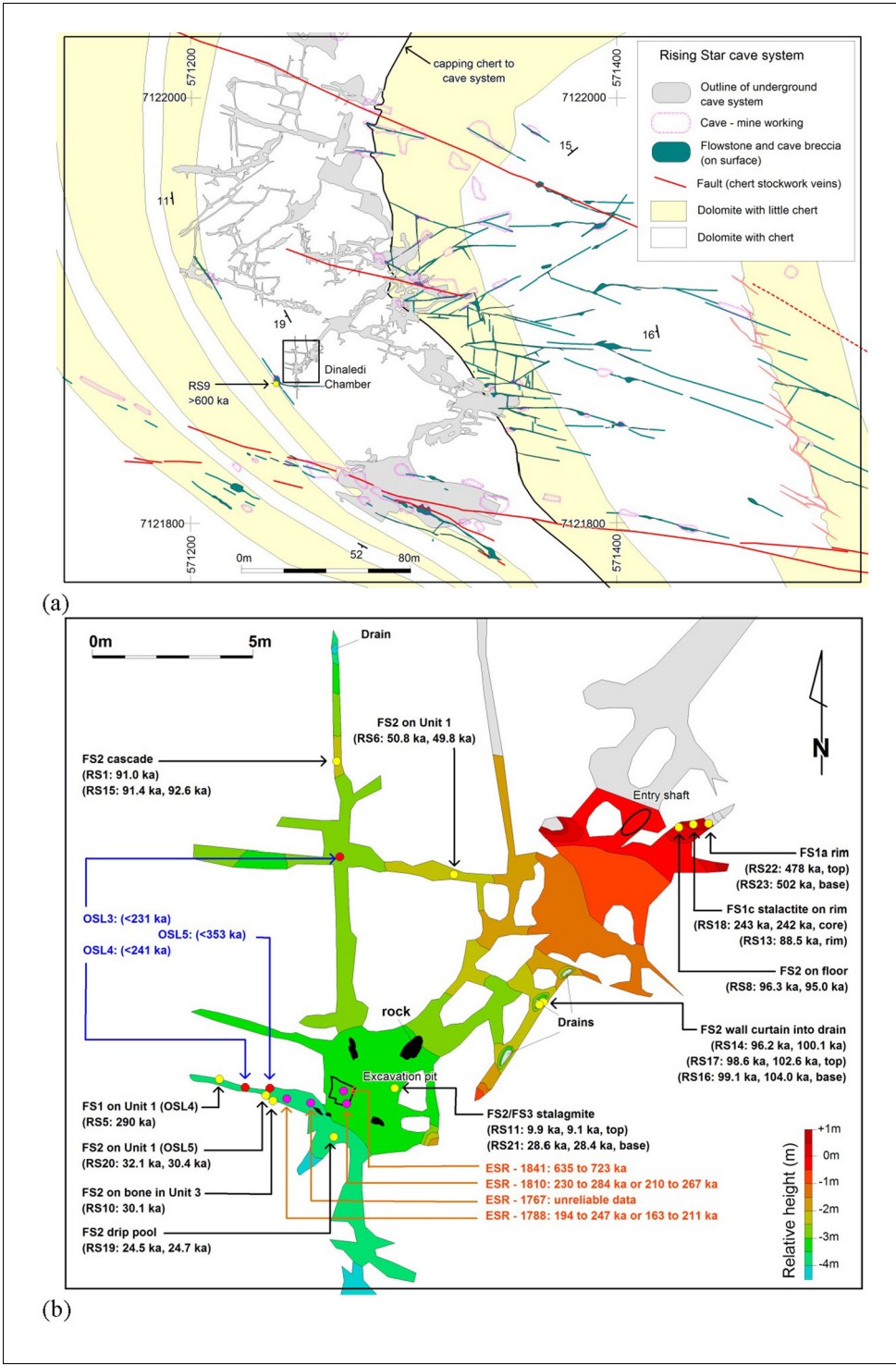

**Figure 1.** Location of Rising Star Cave and the Dinaledi Chamber. (a) Simplified geological map showing the position of the Rising Star Cave (in grey); (b) close-up map of the Dinaledi Chamber showing the distribution of the dating samples, including: U-Th flowstone samples (yellow dots, black text); ESR samples (purple dots, orange text); and OSL samples (red dots, blue text). Age estimates for the different samples are shown, with cross reference to *Tables 1*, *7* and *8*.

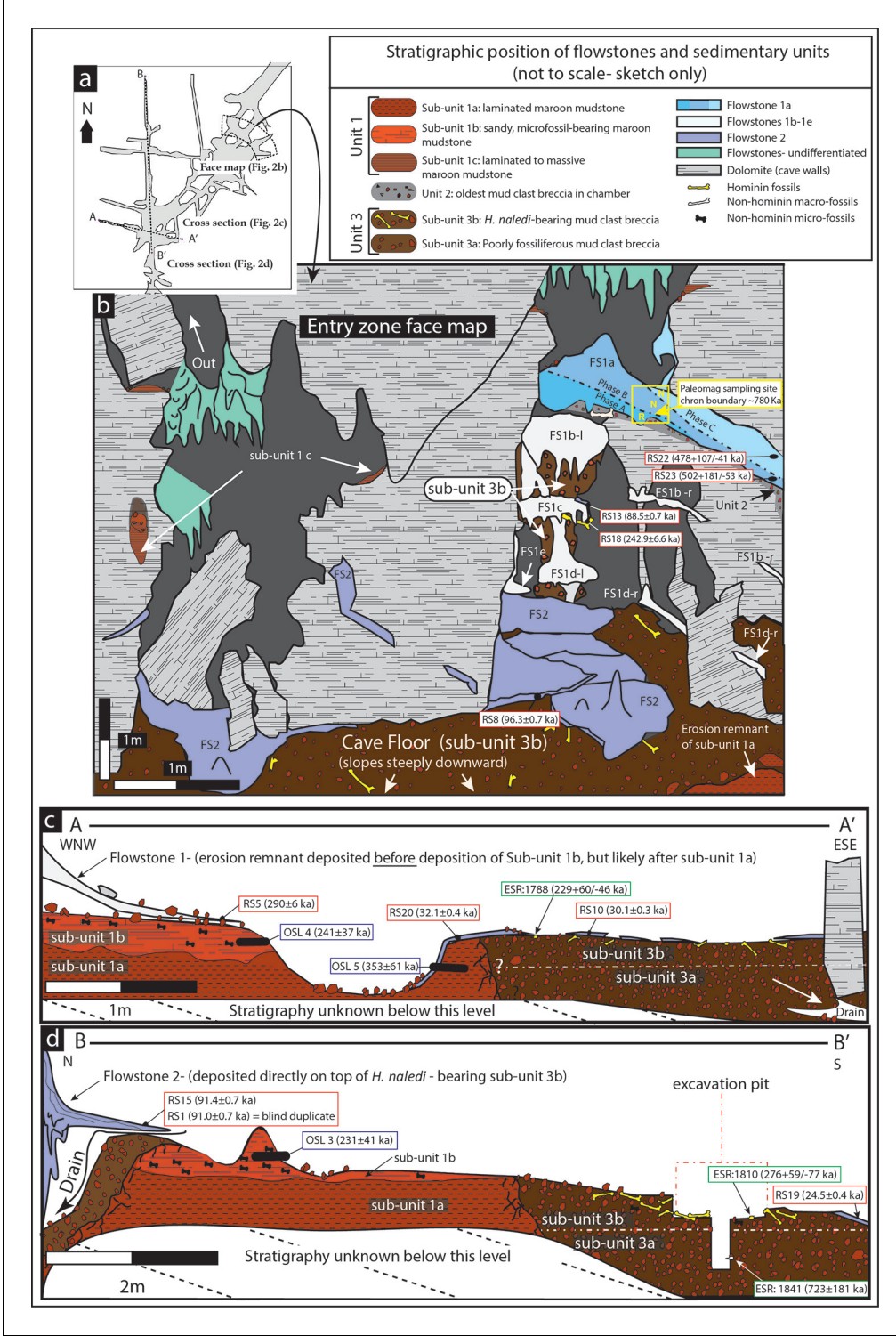

**Figure 2.** Geological face map and cross-sections through the sediment pile at different locations in the Dinaledi Chamber, illustrating the relationships between the flowstone groups and sedimentary units. The positions of the section lines are shown in (**a**); a face map of the entry zone of the Dinaledi Chamber (looking NE) is shown in (**b**); geological cross-sections through the central part of the Dinaledi Chamber near the excavation pit are shown in (**c**) and (**d**).

deposition from water flowing down fractures and side walls. All units and sub-units are time-transgressive, meaning that they are lithostratigraphic units and not chronostratigraphic units that occur in strict temporal order. Periods of sedimentation alternated with periods of erosion, during which sediments were either redeposited or removed from the chamber via floor drains, resulting in erosional remnants of all units occurring in a variety of stratigraphic positions (*Dirks et al., 2015*).

Stalactites have formed at drip points along the roof and associated stalagmites formed below these points. In one area below the entrance to the chamber, these drip points repeatedly formed flowstone aprons over cave sediments that dip towards the deeper part of the chamber. Flowstone also formed as cascades and curtains that developed where water seeped down fractures and ran along the walls to locally spread out, horizontally, across the sediments comprising the cave floor (*Dirks et al., 2015*). The flowstones have preliminarily been sub-divided into three groups demarcating semi-contemporaneous generations of formation, which we named Flowstone Groups 1, 2 and 3 based on their appearance and relationships with each other, and with the floor sediments and other litho-stratigraphic units in the chamber. In making this subdivision it was realised that each group of flowstones will probably comprise a range of ages representing separate flowstone forming events (*Dirks et al., 2015*), a fact borne out by the ages presented below (*Table 1*).

**Flowstone Group 1** (FS1 in *Table 1*; *Figures 1b*, *2* and *3*) includes remnants of what are interpreted to be generally older flowstone units that were partly dissolved and resorbed to leave behind rims or aprons along the side walls of the cave chamber, some with sediment attached below them. Flowstone remnants interpreted as Flowstone Group 1 are mostly restricted to five staggered remnants (Flowstones 1a-e), one above the other in reverse stratigraphic order (oldest on top, youngest at the bottom), near the entry shaft into the Dinaledi Chamber (*Figure 2b*). **Flowstone Group 2**, the most extensive group of flowstones in the chamber (FS2 in *Table 1*, and *Figures 1b*, *2* and *3*), comprises wall aprons and sheets that have spread out across the floor of the Dinaledi Chamber together with drip pools, cascades, curtains, stalactites and stalagmites that connect to these sheets, and, therefore, formed in conjunction with them. **Flowstone Group 3** (FS3 in *Table 1* and *Figures 1b*, *2* and *3*) comprises the flowstone deposits that are actively forming below existing drip points, and include fresh growth of delicate crystals of aragonite and calcite in floor sediments and along cave walls.

Sedimentary deposits within the Dinaledi Chamber can be organized into three primary stratigraphic units (*Dirks et al., 2015*). **Unit 1** consists of deposits of non-lithified, laminated, orange mud interpreted as suspension deposits in standing water (Facies 1a of *Dirks et al., 2015*), and laminated mud with fine sand containing small-scale ripple cross laminations and rodent remains (Facies 1b of *Dirks et al., 2015*), reflecting deposition by shallow, flowing water along the cave floor, with additional sandy material accumulating near local entry points, where fractures higher in the chamber act as sediment conduits (*Dirks et al., 2015*, *Dirks et al., 2016a*).

Within the Dinaledi Chamber **Unit 1** deposits can be divided into three sub-units provisionally called sub-units 1a, 1b and 1c. It is assumed that Unit 1 is time-transgressive and future work may reveal additional sub-units. Sub-unit 1a is composed of laminated orange mudstone with isolated lenses of sandy material, occurs as erosion remnants along the cave floor, and is possibly more extensive beneath younger deposits in the chamber. Sub-unit 1b is dominated by sandy orange mud deposits that are rich in micro-faunal remain, stratigraphically overlies deposits of sub-unit 1a (*Figure 2c and d*), and may have formed through the partial erosion and re-deposition of sub-unit 1a. Deposits of sub-unit 1c are similar in appearance and composition to the laminated, muddy sediments of sub-unit 1a, but they occur along chert ledges, solution pockets and fractures in the chamber walls and along the entry shaft, higher up in the cave chamber (*Figure 2*). The orange mud is mostly the product of the cave formation process, representing the insoluble residue left over when cavities develop via dissolution of dolomite (*Dirks et al., 2015*). Some of the mud-bearing waters seeping out of the fractures would have flowed as water films along the cave walls to deposit mud on ledges and in fractures to form sub-unit 1c, whilst elsewhere this water would have dripped to the floor to contribute to the deposition of sub-unit 1a and 1b.

**Unit 2** is composed of largely lithified mud clast breccia consisting of angular to sub-angular clasts of laminated orange mudstone (similar to that found in Unit 1), embedded in a brown mud matrix (Facies 2 of *Dirks et al., 2015*). The mud clasts are interpreted to be derived locally due to wetting and drying of orange mud deposits, which led to auto-brecciation, and subsequent erosion and re-deposition of angular mud clasts (*Dirks et al., 2015*). We hypothesize that the mud clasts

forming Unit 2 are partly derived from erosion of deposits of sub-unit 1c, and partly from a yet unidentified unit that was likely deposited in fractures within and above the chamber entry zone. Two macro-fossils (partial shafts of long bones) that are non-specific, but not hominin, have been found in Unit 2.

Unit 2 sediments are only exposed as hanging remnants attached below the remains of a composite flowstone sheet (Flowstone 1a) near the entrance shaft into the chamber (*Figure 2b*; *Dirks et al., 2015*). Note that in *Dirks et al. (2015)* Unit 2 was originally defined to also include sediments below Flowstones 1b-e; however, based on our new dating results, the revised definition of Unit 2 has been narrowed to only include the more indurated and distinctly darker coloured erosional remnants of mud clast breccia under Flowstone 1a, which are notable for their absence of hominin fossils. Unit 2 sediments accumulated as a sloping debris cone of mud clast breccia below a vertical fracture system before being covered by flowstone (Flowstone 1a). The debris cone of mud clast breccia was subsequently eroded leaving behind hanging erosion remnants of Unit 2 below a flowstone apron (*Figures 2b* and *3l*). The processes that caused erosion of the Unit 2 debris cone led to the deposition of Unit 3 sediment along the floor of the Dinaledi Chamber as shown in *Figure 8*.

**Unit 3** is composed of largely unlithified, clast-supported, mud clast breccia (Facies 2 of *Dirks et al., 2015*), dominated by reworked angular to sub-angular mud clasts, which are interpreted as being locally derived from the reworking of Units 1 and 2. Unit 3 accumulated below the hanging remnants of the Unit 2 debris cone near the entry shaft, and also extends along the current, sloping cave floor to the SW end of the chamber (*Figures 2c* and *8*). Unit 3 sediments are dynamic in the sense that they are poorly lithified in most places and actively slump towards, and erode into, floor drains that occur in parts of the chamber where sediment is being washed down to deeper levels in the cave (likely as a result of fluctuations in the ground water level). Remains of Unit 3 sediment are attached to apron-like erosional remnants of Flowstones 1b-e near the entrance shaft (*Figure 2a* and *3l*). Erosional remnants of Unit 3 under Flowstone 1c contain in situ long bones consistent with *H. naledi*, which are actively eroding out and accumulating along the present cave floor. Note that *Dirks et al. (2015)* originally included these erosional remnants as part of Unit 2. Everywhere else, Unit 3 deposits are spread across the cave floor as loosely packed, semi-moist, orange mud clasts of varying sizes in which bone material of *H. naledi* is distributed. Unit 3 is partly covered by sheets of Flowstone Groups 1, 2 and 3.

Unit 3 has been divided into a lower and an upper sub-unit, termed sub-unit 3a and 3b (*Figure 2*), based on the respective absence or presence of hominin fossils. Sediments belonging to sub-unit 3a are not directly exposed in the chamber, but their presence has been confirmed in the deepest part of the excavation area (*Figure 2d*). In contrast sub-unit 3b is exposed within the talus cone near the entry shaft and along the cave floor, and contains all of the known *H. naledi* fossils in the chamber (*Figure 2c and d*). The thickness of sub-unit 3b is thought to be no more than 20–30 cm (see below).

## The distribution of fossils, units, and flowstones

All hominin bones identified in the Dinaledi Chamber are contained in deposits of sub-unit 3b. Bones attributed to *H. naledi* have been recovered as: (a) isolated elements that weathered out from erosion remnants of sub-unit 3b below Flowstones 1b-e; (b) as fragmented remains scattered across the cave floor; and (c) as partly articulated remains from a single excavation pit down to a depth of ~20 cm below the current floor level (*Dirks et al., 2015*).

Preliminary ground penetrating radar work (*Naidoo, 2016*) suggests that Unit 3 deposits along the floor of the Dinaledi Chamber could be up to 1.5 m thick. A 50 cm-deep sondage was dug in the centre of the excavation pit, which itself is 20 cm deep, to indicate a minimum depth of 70 cm for the mud clast breccia pile of Unit 3. The top 20 cm of this sediment contains *H. naledi* remains and is part of sub-unit 3b (*Figure 2d*). A discrete contact occurs at 15–20 cm depth, below which no more fossils were encountered with the exception of a single juvenile baboon tooth (sample 1841; *Figure 7*) that was recovered from a depth of 55–60 cm below the original cave floor surface in sediment of sub-unit 3a (*Figures 2* and *8*).

Staining patterns on bone fragments, skeletal element representation, and the fact that bones can be seen to weather out from erosional remnants of sub-unit 3b, indicate that part of the fossil

assemblage has been reworked (*Dirks et al., 2015*). The presence of well-articulated remains in the excavation pit away from the chamber entrance indicates that some of the remains entered the cave intact. The mixed taphonomic signature suggests that fossils entered the cave over a period of time, which is minimally assumed to be during deposition of sub-unit 3b, and before deposition of Flowstone 1c. Fossil entry may have continued as sediment accumulations of sub-unit 3b near the entry shaft were reworked and redistributed along the cave floor (*Figure 8*).

The stratigraphic relationships in the Dinaledi Chamber suggest that Unit 1 sediments were deposited over a long period, which both predates and spans the more limited depositional time-frames of Units 2 and 3. Hence, Unit 1 is time-transgressive, meaning that these sediments were (and are) constantly forming in different parts of the chamber due to weathering of the dolomitic cave walls (i.e., wad formation *sensu Martini et al., 2003*), and that their age is dependent upon where in the cave the material is located. At present, we can only divide Unit 1 into three sub-units, but we hypothesize that an older sub-unit consisting of laminated orange mudstone exists (or existed) higher up in the chamber as well (possibly only on ledges and in fractures), which was eroded to provide some of the sediment that formed Unit 2 and parts of Unit 3, near the entry shaft.

Flowstone 1a, which overlies remnants of Unit 2, is the oldest flowstone unit in the chamber, and displays evidence of multiple phases of flowstone formation followed by partial dissolution (*Figure 2b*). Flowstone dissolution occurred during time periods when the water table was elevated and the chamber was filled with standing water. The erosion remnants of Flowstone 1a dip towards the deeper part of the chamber, indicating that at the time of its formation, a sloping debris cone of Unit 2 sediment was present. Erosion of Unit 2 sediments from underneath Flowstone 1a only occurred after the flowstone had formed and lithified the top of Unit 2. Following erosion of Unit 2, deposition of Unit 3 began, as sediment and mud clasts spread out over the cave floor and also filled much of the space underneath Flowstone 1a. This has led to an inverted stratigraphy near the cave entrance, although a normal stratigraphy is documented at the bottom of the chamber, where the cave floor is flat lying and sediment of Unit 3 progressively built up (*Figures 2b* and *8*). At some point during these processes remains of *H. naledi* entered the cave chamber, marking the start of deposition of sub-unit 3b. Following deposition of sub-unit 3b and the hominin remains, Flowstones 1b-e were deposited over sub-unit 3b in the entry zone. These flowstones have been interpreted as younger than Flowstone 1a, but older than the Flowstone Group 2 sheets along the cave floor. In other words, after deposition of Unit 3 commenced to form the talus cone near the entrance of the chamber, parts of the cone slumped and eroded down towards deeper parts of the chamber after Flowstones 1b-e were deposited, but before Flowstone Group 2 was deposited. This slumping motion was probably driven by sediment being removed from the base of the stratigraphic pile through floor drains.

Flowstone Group 2 covers erosion remnants of Flowstones 1a-e as coatings and stalactites along drip rims. In places, Flowstone Group 2 also covers erosion remnants of Unit 1 and Unit 3 along the floor and displays variable relationships with Unit 3 (*Figure 2*). Where parts of Unit 3 have been eroded via floor drains, hanging remnants of Flowstone Group 2 can be found attached to the walls as fringing aprons, up to 10 cm above the current floor level, establishing the fact that parts of the floor are currently in a state of erosion. In other places, Flowstone Group 2 sheets directly overlie Unit 3 and the *H. naledi* fossils it contains. These varying relationships indicate that Flowstone Group 2 sheets were deposited over an extended period of time, post-dating deposition and partial reworking of sub-unit 3b.

In summary, the stratigraphic context indicates that the *H. naledi* fossils entered the cave during deposition (and possibly during partial reworking) of sub-unit 3b, after deposition of the older sediments of Unit 1 (sub-unit 1a) and Unit 2. Several isolated, non-hominin bone fragments in hanging erosion remnants of Unit 2 and a single baboon tooth in floor sediments in sub-unit 3a were deposited prior to the entry of the hominin remains. The accumulation of Unit 3 along the cave floor involved a dynamic interplay between the accumulation of mud clast breccia below sediment entry points or in situ sediment sources (Unit 1 and Unit 2) in the chamber, and erosion through floor drains resulting in contrasting stratigraphic relationships across the chamber (*Figures 2* and *8*).

## Dating the *H. naledi* fossils

Most fossil deposits in the Cradle of Humankind that have been dated are between 0.5 and 3.7 Ma old and consist of bone material encased in well-cemented hard clastic rocks commonly referred to as cave breccia (e.g., *Wilkinson, 1985*; *O'Regan and Reynolds, 2009*; *Herries et al., 2009*; *Pickering et al., 2011b*; *Granger et al., 2015*). In the absence of volcanic deposits, it is generally difficult to obtain accurate ages for the fossils, not just because reliable techniques are few, but mostly because the stratigraphic sequences in the caves are complex, discontinuous and frequently reworked (e.g., *Brain, 1993*; *Pickering et al., 2011a*; *Bruxelles et al., 2014*; *Stratford et al., 2014*). Workers have relied on a combination of biochronology of faunal remains, palaeomagnetic work and a range of radiometric methods, including U-Pb, U-Th and ESR dating targeting flowstones and fossil teeth (e.g., *Vrba, 1975*; *Partridge et al., 1999*; *Berger et al., 2002*; *Walker et al., 2006*; *Herries et al., 2006*, *2013*, *2014*; *Herries and Shaw, 2011*; *Dirks et al., 2010*; *Pickering and Kramers, 2010*; *Pickering et al., 2011a*; *Herries and Adams, 2013*), as well as limited cosmogenic ($^{10}$Be, $^{16}$Al) dating (e.g., *Partridge et al., 2003*; *Granger et al., 2015*; *Dirks et al., 2016b*). Whilst some of these techniques are well established, others such as the application of cosmogenic isochrons (e.g., *Granger et al., 2015*) are relatively new and not without significant analytical (and interpretative) challenges (*Kramers and Dirks, 2017*), and all efforts are strongly dependent on the stratigraphic interpretation of the fossils or units that are being dated.

Unlike other fossil deposits in the Cradle of Humankind, the remains in the Dinaledi Chamber are largely restricted to hominins. This makes it impossible to use biochronology as a preliminary technique to assess the age of the fossils. In addition, the fossils are contained in mostly unconsolidated muddy sediment with clear evidence of a mixed taphonomic signature indicative of repeated cycles of reworking and more than one episode of primary deposition (*Dirks et al., 2015*). This indicates that caution is required when interpreting the stratigraphy and the age of the fossils they contain.

In preparation for this study, trial dating of the deposits in the Dinaledi Chamber was undertaken to obtain an indication of the age of the deposit and the best techniques to apply. Preliminary work was focussed on assessing the viability of U-series techniques for flowstone dating, using $^{14}$C for dating bone fragments, and using OSL to test samples of quartz-bearing Unit 1 (*Dirks et al., 2015*). Initial tests were carried out at the University of Johannesburg (UJ) to assess suitability for U-Pb dating, which allows for the dating of older (>500 ka) flowstone material (e.g., *Walker et al., 2006*; *Pickering et al., 2010*; *Pickering and Kramers, 2010*), on the assumption that the *H. naledi* material could be older than 1 Ma based on its primitive morphology (*Berger et al., 2015*; *Dembo et al., 2016*; *Hawks and Berger, 2016*; *Thackeray, 2016*; *Hawks et al., 2017*). It was found that the older flowstones in the Dinaledi Chamber contained excessive common Pb caused by the inclusion of detrital material (mainly clays) making them unsuitable for U-Pb dating (*Dirks et al., 2015*). In contrast, preliminary tests with U-Th disequilibrium dating at James Cook University (JCU) returned promising results. U-Th dating is more precise in the <500 ka range than U-Pb dating, and is much less critically affected by detrital material. The initial tests with U-Th disequilibrium dating revealed that the fossils may be much younger than originally anticipated (e.g., *Dembo et al., 2016*; *Thackeray, 2016*), and mostly well within the range of the U-Th technique. Therefore, U-Pb dating was not pursued further.

Preliminary tests with OSL were conducted at the University of the Witwatersrand (Wits) on samples from Unit 1, which were assumed to be older than the fossils of *H. naledi*. These preliminary studies, and the results contained in this paper, are the first OSL results for cave sediments from the CoH, and again indicated that the *H. naledi* fossils were probably relatively young (i.e.,<500 ka).

Tests with radiocarbon ($^{14}$C) dating were undertaken through a commercial facility (Beta Analytic Inc. in Florida), to ensure a fast turn-around time for results. At the time these dating tests were done, it was already known from U-Th and OSL tests that the *H. naledi* fossils would be too old to be dated by $^{14}$C. Nevertheless, analyses were carried out as part of the due diligence process, and the results of these tests are presented here. Following this initial work, no further radiocarbon studies were carried out.

The preliminary results have guided the subsequent dating strategy and sampling approach reported here. The dating strategy was designed to achieve three objectives: (i) establish a detailed

stratigraphy for the cave sediments in the Dinaledi Chamber; (ii) date sedimentary units that potentially bracket the fossil-bearing deposits; and (iii) date the fossils directly.

To obtain an upper age limit for the fossil-bearing deposits of Unit 3 (i.e. sub-unit 3b), we conducted U-Th dating of flowstones that directly overlie Unit 3. A large number of such flowstones were sampled with the aim of finding the oldest flowstone directly overlying *H. naledi* fossils. To obtain a lower age limit for sub-unit 3b, erosional remnants of Unit 1 sediments that were at least partially covered by fossil-bearing sub-unit 3b sediments, were sampled for OSL dating on the assumption that sub-units 1a and 1b in these areas are older than sub-unit 3b (*Dirks et al., 2015*). This was done in the full knowledge that OSL dating of cave sediments is complex and difficult to interpret (e.g., *Roberts et al., 2009*), and probably imprecise. As an internal control, we also sampled flowstones that cover the outcrops of sub-units 1a and 1b from which OSL samples were taken. These flowstones were dated with U-Th with the expectation that they are younger than the underlying Unit 1 sediments. In addition to OSL, Flowstone 1a, which overlies Unit 2 sediments, was sampled for palaeomagnetic analyses. This flowstone was targeted, because it was expected to be the oldest flowstone in the chamber and possibly older than 780 ka, and hence could potentially record reverse magnetic polarity (e.g., *Singer, 2014*). In this case, this would constrain the minimum age of Unit 2.

The best age estimates for *H. naledi* can be obtained by directly dating fossil material. It was clear from preliminary tests that this could not be achieved with $^{14}$C, and instead combined ESR and U-Th disequilibrium dating techniques (US-ESR; *Grün et al., 1988*) were applied to three *H. naledi* teeth that were freshly collected from near the site of the original excavation (*Figures 1*, *2*, *4* and *6*), as well as a single baboon tooth (cf. *Papio*) that had been recovered from sub-unit 3a below the hominin-bearing horizon (*Figures 2d* and *7*).

Once results were obtained for ESR and U-Th dating, it became apparent that OSL dating would only provide general age constraints that confirmed the ESR results, but in their own right did not return additional age constraints for the fossils. OSL results were also difficult to interpret in the complex cave environment that was strongly affected by Rn loss (see Discussion). It was, therefore, decided not to pursue more detailed OSL studies at this stage, even though we did carry out preliminary tests for single grain and feldspar analyses at the University of Wollongong, to assess the suitability of these techniques. Pilot results are encouraging, and suggest that future, detailed OSL studies are worth pursuing.

## Results

### U-Th dating of flowstones

U-Th dating of 17 flowstone samples (*Figure 3*) has yielded minimum depositional age estimates for the sedimentary units they overlie, and has provided insights into the timing of flowstone formation events (*Tables 1*, *2* and *3*). Three separate checks were built into the U-Th dating strategy to ensure robust results would be obtained. Independent dates for the same samples were obtained by laboratories at JCU and at the University of Melbourne (UoM), with results displaying a high degree of concordance. In instances where samples were obtained from the same flowstone layer, but at different stratigraphic levels (e.g., sample pairs RS13 and RS18, RS22 and RS23, RS16 and RS17, and RS11 and RS21) all ages are consistent with stratigraphic order, and blind duplicates of the same sample (RS1 and RS15) returned identical results within error, indicating that results are both accurate and precise. Double blind U-Th results from JCU and UoM are shown in *Tables 2* and *3*, respectively. The distribution of flowstone ages across the Dinaledi Chamber is shown in *Figures 1b* and *2*.

The oldest dated flowstone in the assemblage is Flowstone 1a overlying Unit 2, which yields age estimates of 502 +181/–53 ka (RS 23) and 478 +107/–41 ka (RS 22) (*Table 1*). The next oldest age comes from a flowstone interpreted as Flowstone Group 1 overlying sediment of sub-unit 1b to the W of the excavation pit (*Figure 1b*) with an age of 290 ± 6 ka (RS5; *Table 1*). This age is younger than the OSL age derived from sub-unit 1a (OSL5), but is slightly older than OSL ages derived from sub-unit 1b in this location (OSL4; *Figure 2c*). This suggests that sub-unit 1a was deposited prior to precipitation of this flowstone and that sub-unit 1b formed out of stratigraphic order due to erosion and redeposition of the top of sub-unit 1a beneath this flowstone. However, the U-Th date for RS5

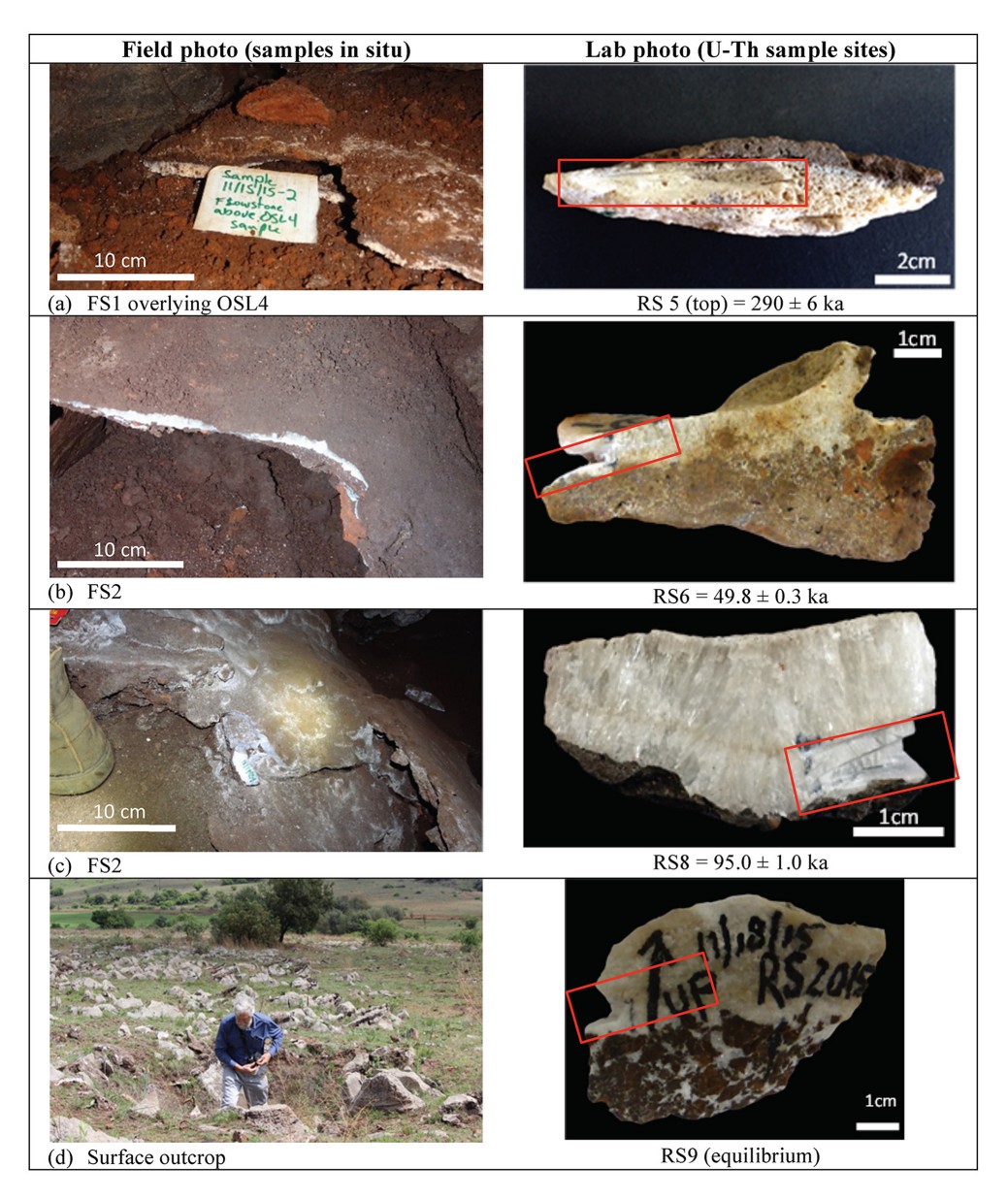

| Field photo (samples in situ) | Lab photo (U-Th sample sites) |
|---|---|
| (a) FS1 overlying OSL4 | RS 5 (top) = 290 ± 6 ka |
| (b) FS2 | RS6 = 49.8 ± 0.3 ka |
| (c) FS2 | RS8 = 95.0 ± 1.0 ka |
| (d) Surface outcrop | RS9 (equilibrium) |

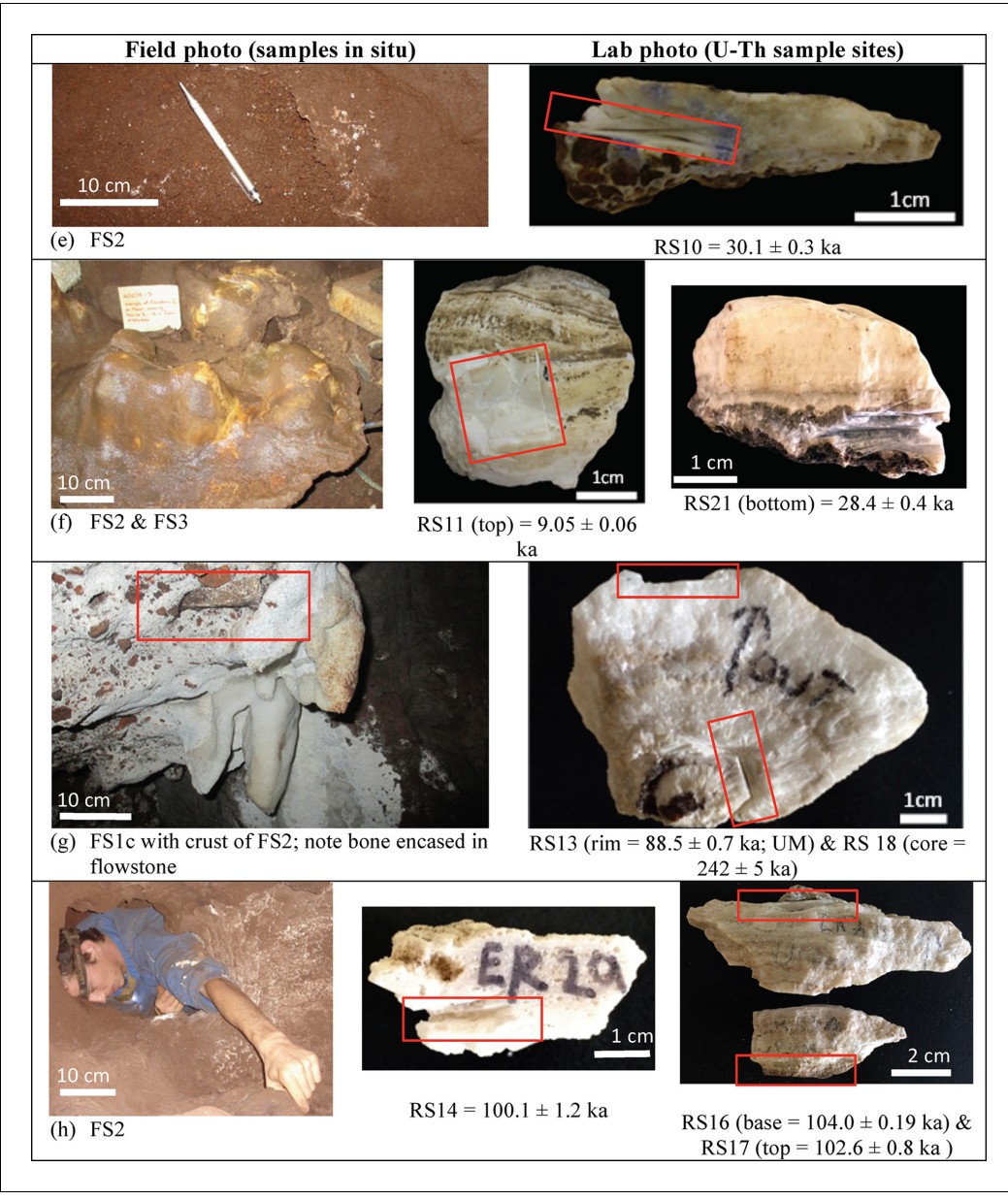

| Field photo (samples in situ) | Lab photo (U-Th sample sites) |
|---|---|
| (e)  FS2 | RS10 = 30.1 ± 0.3 ka |
| (f)  FS2 & FS3 | RS11 (top) = 9.05 ± 0.06 ka    RS21 (bottom) = 28.4 ± 0.4 ka |
| (g)  FS1c with crust of FS2; note bone encased in flowstone | RS13 (rim = 88.5 ± 0.7 ka; UM) & RS 18 (core = 242 ± 5 ka) |
| (h)  FS2 | RS14 = 100.1 ± 1.2 ka    RS16 (base = 104.0 ± 0.19 ka) & RS17 (top = 102.6 ± 0.8 ka ) |

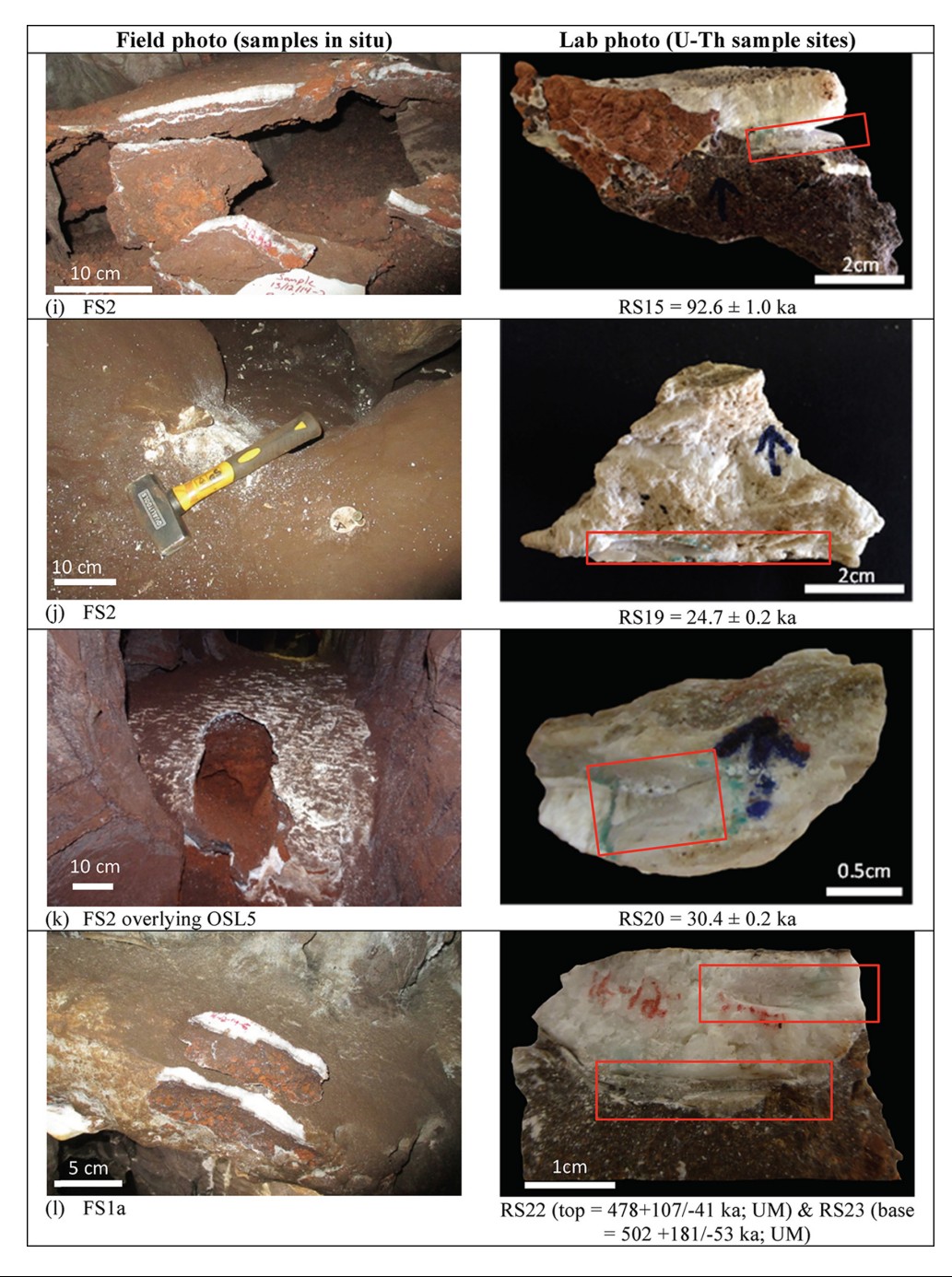

**Figure 3.** Field and close-up photographs of all flowstone samples collected for U-Th dating.  The flowstone groups (i.e., Flowstone Groups 1, 2 or 3), sample numbers, and ages (2σ uncertainty), as listed in *Table 1*, are shown below each sample. Ages reported here are from JCU, unless otherwise stated.

should be interpreted with caution as the flowstone has a porous texture (*Figure 3a*), which probably indicates some degree of dissolution and/or recrystallization of the primary calcite, and may have affected the U-Th systematics (see Discussion).

Flowstone samples that overlie sub-unit 3b, which contains the *H. naledi* fossils, yield age estimates that fall within four distinct time periods: ~242 ka (RS18 = 242 ± 5 ka [JCU] and 242.9 ± 6.6 ka

**Table 1.** Summary table of U-Th disequilibrium ages obtained for samples from the Dinaledi Chamber by James Cook University (JCU - [1]) and the University of Melbourne (UoM - [2]). The detailed analytical results are shown in **Tables 2** and **3**. Sample locations are shown in **Figure 1b**. The data are ranked by increasing age of the oldest flowstone horizon within the sample, based on the JCU ages. The grey shading highlights the different age groupings observed within the flowstones: 24–32 ka, ~50 ka, 88–105 ka, ~242 ka, ~290 ka and >440 ka. Ages are reported relative to 1950.

| Sample ID | Flowstone group | Underlying unit | Age[1] (ka) | 2σ[1] (ka) | Age[2] (ka) | 2σ[2] (ka) |
|---|---|---|---|---|---|---|
| RS19 | FS2 | sub-unit 3b | 24.7 | 0.2 | 24.53 | 0.43 |
| RS11 | FS3 (top to RS21) | FS2 | 9.05 | 0.06 | 9.946 | 0.063 |
| RS21 | FS2 (base to RS11) | sub-unit 3b | 28.4 | 0.4 | 28.62 | 0.29 |
| RS10 | FS2 | sub-unit 3b (and bone) | 30.1 | 0.3 | – | – |
| RS20 | FS2 | sub-unit 1a (Facies 1a; OSL5) | 30.4 | 0.2 | 32.12 | 0.38 |
| RS6 | FS2 | sub-unit 1a, sub-unit 3b | 49.8 | 0.3 | 50.82 | 0.43 |
| RS15 | FS2 (blind duplicate of RS1) | sub-unit 3b | 92.6 | 1.0 | 91.40 | 0.65 |
| RS1 | FS2 (blind duplicate of RS15) | sub-unit 3b | – | – | 91.04 | 0.72 |
| RS8 | FS2 (below FS1a-e) | sub-unit 3b | 95.0 | 1.0 | 96.29 | 0.69 |
| RS14 | FS2 | in drain, along dolostone wall | 100.1 | 1.2 | 96.20 | 0.36 |
| RS17 | FS2 (top to RS16) | in drain, along dolostone wall | 102.6 | 0.8 | 98.6 | 1.4 |
| RS16 | FS2 (base to RS17) | in drain, along dolostone wall | 104.0 | 1.9 | 99.1 | 1.4 |
| RS13 | FS2 (rim to RS18) | sub-unit 3b | – | – | 88.46 | 0.67 |
| RS18 | FS1c (core to RS13) | sub-unit 3b | 242 | 5 | 242.9 | 6.6 |
| RS5 | FS1 | sub-unit 1b (Facies 1b; OSL4) | 290 | 6 | – | – |
| RS22 | FS1a (top to RS23) | Unit 2 | equilibrium | – | 478 | +107/−41 |
| RS23 | FS1a (base to RS22) | Unit 2 | equilibrium | – | 502 | +181/−53 |
| RS9 | n/a (surface outcrop) | n/a | equilibrium | – | equilibrium | – |

[1]James Cook University (JCU), Advanced Analytical Centre.

[2]University of Melbourne (UoM), paleochronology laboratory.

[UoM]) for Flowstone 1c; 88–106 ka (RS1, RS8, RS13-17); 50 ka (RS6); and 24–32 ka (RS10, RS19-21) for Flowstone Group 2 deposits. The results for RS18 provide a minimum age for the *H. naledi* fossils in this part of the cave.

An actively forming (i.e., dripping) stalagmite of Flowstone Group 3 (RS11) on top of an older base of Flowstone Group 2 returned a younger age of 9–10 ka (RS11 = 9.05 ± 0.06 [JCU] and 9.95 ± 0.06 ka [UoM]). The final flowstone sample that was dated in this study was collected on the land surface above the cave system. It was sampled from a ~14 cm wide vertical flowstone-filled fracture exposed on the surface above the southern end of the Dinaledi Chamber itself. This is the only possible alternative entry-way into the Dinaledi Chamber that we have observed. The replicate samples (RS9) analysed at JCU and UoM both indicated secular equilibrium, which confirms that the flowstone sealed this fracture sometime before ~600 ka, eliminating this thin flowstone-filled fracture as a possible alternative entrance for *H. naledi* into the Dinaledi Chamber. It should also be noted that no evidence of a talus cone or any other evidence of sediment entry into the chamber below this point has been observed.

The spatial distribution of the flowstones belonging to the different age groups (**Figure 1b**) indicates that the oldest flowstones (Flowstone Group 1) occur near the entry zone into the chamber and as an erosional remnant (RS 5) near the back of a WNW-trending fracture W of the excavation pit. The 88–106 ka flowstones formed in three separate parts of the chamber (**Figure 1b**): (a) on top of older flowstones near the entry; (b) as wall drapes above a drain 6 m SW of the entry; and (c)

**Table 2.** U-Th data table for James Cook University. Uncertainties include: analytical error, decay constant uncertainty, and uncertainty on initial $^{230}Th/^{232}Th$. Ages are reported relative to 1950 and assume an initial $^{230}Th/^{232}Th$ activity of 0.83 ± 0.5, and the equation given in **Placzek et al. (2006)**. Decay constants for $^{234}U$ and $^{230}Th$ are from **Cheng et al. (2013)**.

| Sample ID | U (ppm) | $^{234}U/^{238}U$ | 2σ | $^{230}Th/^{238}U$ | 2σ | $^{232}Th/^{238}U$ | 2σ | Age (ka BP) | 2σ (ka) | $^{234}U/^{238}U_{initial}$ | 2σ |
|---|---|---|---|---|---|---|---|---|---|---|---|
| | | (activity) | | (activity) | | | | (corrected) | | (activity) | |
| RS11 | 2.314 | 1.772 | 0.050 | 0.144 | 0.001 | 0.0001088 | 0.0000005 | 9.05 | 0.06 | 1.8184 | 0.0003 |
| RS19 | 0.652 | 1.855 | 0.001 | 0.387 | 0.002 | 0.002176 | 0.000008 | 24.7 | 0.2 | 1.989 | 0.001 |
| RS21 | 0.421 | 1.946 | 0.001 | 0.460 | 0.004 | 0.001920 | 0.000015 | 28.4 | 0.4 | 2.109 | 0.002 |
| RS10 | 0.846 | 1.885 | 0.001 | 0.466 | 0.003 | 0.000792 | 0.000003 | 30.1 | 0.3 | 2.053 | 0.001 |
| RS20 | 0.795 | 1.855 | 0.001 | 0.463 | 0.003 | 0.001363 | 0.000005 | 30.4 | 0.2 | 2.022 | 0.001 |
| RS6 | 0.560 | 1.966 | 0.001 | 0.747 | 0.003 | 0.000974 | 0.000002 | 49.8 | 0.3 | 2.263 | 0.002 |
| RS15 | 0.400 | 1.912 | 0.001 | 1.164 | 0.008 | 0.00472 | 0.00001 | 92.6 | 1.0 | 2.484 | 0.007 |
| RS8 | 0.328 | 1.813 | 0.003 | 1.120 | 0.008 | 0.00316 | 0.00002 | 95.0 | 1.0 | 2.373 | 0.007 |
| RS14 | 0.734 | 1.639 | 0.095 | 1.039 | 0.008 | 0.00298 | 0.00002 | 100.1 | 1.2 | 2.175 | 0.008 |
| RS17 | 0.680 | 1.609 | 0.001 | 1.032 | 0.005 | 0.000679 | 0.000001 | 102.6 | 0.8 | 2.150 | 0.005 |
| RS16 | 0.973 | 1.583 | 0.000 | 1.024 | 0.011 | 0.000403 | 0.000006 | 104.0 | 1.9 | 2.12 | 0.01 |
| RS18 | 0.152 | 1.848 | 0.001 | 1.856 | 0.013 | 0.01175 | 0.00005 | 242 | 5 | 3.66 | 0.05 |
| RS5 | 0.090 | 1.728 | 0.001 | 1.818 | 0.009 | 0.01732 | 0.00005 | 290 | 6 | 3.92 | 0.07 |
| RS23 | 0.314 | 1.187 | 0.002 | 1.315 | 0.011 | 0.00346 | 0.00002 | >400 | – | – | |
| RS22 | 0.367 | 1.209 | 0.001 | 1.322 | 0.008 | 0.000125 | 0.000001 | >400 | – | – | |
| RS9 | 0.737 | 1.007 | 0.002 | 1.029 | 0.004 | 0.000462 | 0.000001 | >400 | – | – | |

**Table 3.** U-Th data table for the University of Melbourne. Activity ratios are determined after **Hellstrom (2003)** and **Drysdale et al. (2012)**. Ages are corrected for initial $^{230}Th$ using **Equation 1** of **Hellstrom (2006)**, the decay constants of **Cheng et al. (2013)**, and an initial $^{230}Th/^{232}Th$ activity of 1.5 ± 1.5. The initial $^{234}U/^{238}U$ ratios are calculated using corrected ages, which are reported relative to 1950.

| Sample ID | U (ppm) | $^{234}U/^{238}U$ | 2σ | $^{230}Th/^{238}U$ | 2σ | $^{232}Th/^{238}U$ | 2σ | Age (ka BP) | 2σ (ka) | $^{234}U/^{238}U_{initial}$ | 2σ |
|---|---|---|---|---|---|---|---|---|---|---|---|
| | | (activity) | | (activity) | | | | (corrected) | | (activity) | |
| RS11 | 1.518 | 1.808 | 0.003 | 0.1597 | 0.0009 | 0.0000875 | 0.0000004 | 9.946 | 0.063 | 1.831 | 0.004 |
| RS19 | 0.501 | 1.884 | 0.011 | 0.3916 | 0.0026 | 0.004322 | 0.000010 | 24.53 | 0.43 | 1.947 | 0.011 |
| RS21 | 0.361 | 1.968 | 0.011 | 0.4654 | 0.0030 | 0.0011342 | 0.0000019 | 28.62 | 0.29 | 2.049 | 0.011 |
| RS20 | 0.626 | 1.878 | 0.011 | 0.4925 | 0.0032 | 0.0023837 | 0.0000040 | 32.12 | 0.38 | 1.961 | 0.011 |
| RS6 | 0.276 | 2.023 | 0.004 | 0.7856 | 0.0021 | 0.00496 | 0.00010 | 50.82 | 0.43 | 2.181 | 0.004 |
| RS13 | 0.076 | 2.006 | 0.004 | 1.1837 | 0.0047 | 0.004786 | 0.000058 | 88.46 | 0.67 | 2.291 | 0.005 |
| RS15 | 0.381 | 1.934 | 0.004 | 1.1661 | 0.0029 | 0.00639 | 0.00012 | 91.37 | 0.65 | 2.209 | 0.005 |
| RS14 | 0.665 | 1.626 | 0.003 | 1.0010 | 0.0015 | 0.001262 | 0.000014 | 96.24 | 0.36 | 1.822 | 0.003 |
| RS8 | 0.257 | 1.831 | 0.004 | 1.1397 | 0.0034 | 0.005746 | 0.000060 | 96.29 | 0.69 | 2.091 | 0.004 |
| RS17 | 0.517 | 1.637 | 0.009 | 1.0248 | 0.0066 | 0.0023963 | 0.0000037 | 98.6 | 1.4 | 1.841 | 0.010 |
| RS16 | 0.905 | 1.590 | 0.010 | 0.9963 | 0.0067 | 0.0017099 | 0.0000037 | 99.1 | 1.4 | 1.780 | 0.011 |
| RS18 | 0.104 | 2.001 | 0.011 | 2.0320 | 0.0140 | 0.020557 | 0.000041 | 242.9 | 6.6 | 2.987 | 0.027 |
| RS22 | 0.324 | 1.228 | 0.007 | 1.3017 | 0.0083 | 0.0001201 | 0.0000008 | 478 | +107/–41 | – | – |
| RS23 | 0.206 | 1.225 | 0.007 | 1.3016 | 0.0093 | 0.007818 | 0.000016 | 502 | +181/–53 | – | – |
| RS9 | 0.896 | 1.010 | 0.002 | 1.0204 | 0.0018 | 0.000916 | 0.000012 | – | – | – | – |

**Table 4.** Summary table of U-Th disequilibrium ages obtained for the three *H. naledi* teeth (samples 1767, 1788 and 1810) and the baboon tooth (sample 1841) from the Dinaledi Chamber obtained by SCU-UoW. No age calculations were carried out for U concentrations of ≤0.5 ppm or U/Th ≤250 (indicated in red and underlined). Mean values in this table only incorporate values from which meaningful ages could be calculated (indicated in black), however all values (i.e., red and black) were averaged to obtain the relevant mean values reported in *Table 4*. All uncertainties are given as 2σ.

| Sample1767 | U (ppm) | U/Th | $^{230}$Th/$^{238}$U | 2σ | $^{234}$U/$^{238}$U | 2σ | Age (ka) | 2s (ka) | $(^{234}$U/$^{238}$U)i | 2σ |
|---|---|---|---|---|---|---|---|---|---|---|
| 1767-1 D | 7.22 | 685 | 2.167 | 0.024 | 6.259 | 0.009 | 43.5 | 1.1 | 6.949 | 0.026 |
| 1767-2 D | 7.75 | 996 | 2.261 | 0.023 | 6.282 | 0.010 | 45.5 | 1.1 | 7.009 | 0.030 |
| 1767-3 D | 8.03 | 196 | 2.225 | 0.825 | 6.276 | 0.012 | – | – | – | – |
| 1767-4 D | 8.55 | 951 | 2.209 | 0.031 | 6.301 | 0.009 | 44.1 | 1.4 | 7.007 | 0.030 |
| 1767-5* E | 3.69 | 1238 | 2.259 | 0.031 | 6.197 | 0.055 | 46.2 | 1.8 | 6.924 | 0.126 |
| 1767-6* E | 1.76 | 108 | 2.239 | 1.133 | 6.165 | 0.038 | – | – | – | – |
| 1767-7* E | 2.15 | 109 | 2.337 | 0.947 | 6.231 | 0.024 | – | – | – | – |
| 1767-8* E | 2.46 | 518 | 2.276 | 0.021 | 6.253 | 0.019 | 46.1 | 1.1 | 6.986 | 0.048 |
| Mean: | | | | | | | | | | |
| **1767 D** | **7.84** | **877** | **2.212** | **0.026** | **6.281** | **0.009** | **44.5** | **1.2** | **6.988** | **0.029** |
| **1767 E** | **3.08** | **878** | **2.268** | **0.026** | **6.225** | **0.037** | **46.2** | **1.4** | **6.955** | **0.087** |
| Sample1788 | U (ppm) | U/Th | $^{230}$Th/$^{238}$U | 2σ | $^{234}$U/$^{238}$U | 2σ | Age (ka) | 2s (ka) | $(^{234}$U/$^{238}$U)i | 2σ |
| 1788-1 D | 6.67 | 390 | 2.967 | 0.026 | 6.423 | 0.011 | 61.4 | 1.5 | 7.453 | 0.054 |
| 1788-2 D | 7.08 | 176 | 3.370 | 0.833 | 6.441 | 0.010 | – | – | – | – |
| 1788-3 D | 7.17 | 60 | 3.206 | 3.126 | 6.394 | 0.049 | – | – | – | – |
| 1788-4 D | 7.45 | 1391 | 3.313 | 0.023 | 6.445 | 0.010 | 70.3 | 1.4 | 7.645 | 0.056 |
| 1788-5 D | 5.52 | 4423 | 3.269 | 0.023 | 6.349 | 0.010 | 70.4 | 1.4 | 7.531 | 0.052 |
| 1788-6 D | 5.07 | 4090 | 3.416 | 0.014 | 6.378 | 0.014 | 74.1 | 1.1 | 7.634 | 0.054 |
| 1788-7 D | 5.39 | 4729 | 3.385 | 0.020 | 6.400 | 0.014 | 72.9 | 1.4 | 7.640 | 0.054 |
| 1788-8 D | 5.93 | 3209 | 3.427 | 0.015 | 6.393 | 0.013 | 74.2 | 1.1 | 7.654 | 0.054 |
| 1788-9 D | 5.24 | 4329 | 3.449 | 0.014 | 6.413 | 0.014 | 74.5 | 1.0 | 7.685 | 0.052 |
| 1788-10 D | 4.89 | 3161 | 3.390 | 0.010 | 6.403 | 0.011 | 73.0 | 0.9 | 7.645 | 0.052 |
| 1788-11 D | 4.8 | 2556 | 3.394 | 0.014 | 6.416 | 0.014 | 72.9 | 1.0 | 7.659 | 0.052 |
| 1788-12 D | 5.48 | 1606 | 3.356 | 0.017 | 6.384 | 0.014 | 72.3 | 1.1 | 7.609 | 0.052 |
| 1788-13 D | 5.04 | 838 | 3.317 | 0.025 | 6.420 | 0.014 | 70.7 | 1.5 | 7.623 | 0.058 |
| 1788-14 D | 5.69 | 93 | 3.281 | 2.426 | 6.408 | 0.013 | – | – | – | – |
| 1788-15 D | 5.03 | 72 | 3.315 | 3.731 | 6.427 | 0.014 | – | – | – | – |
| 1788-16 E | 0.13 | 3 | 1.786 | 18.149 | 3.834 | 0.267 | – | – | – | – |
| 1788-17 E | 0.68 | 25 | 0.752 | 9.149 | 6.248 | 0.273 | – | – | – | – |
| 1788-18 E | 0.4 | 16 | 0.801 | 13.053 | 6.236 | 0.050 | – | – | i | – |
| 1788-19 E | 0.08 | 3 | 1.783 | 36.231 | 4.301 | 0.288 | – | – | – | – |
| 1788-20 E | 1.02 | 306 | 2.990 | 0.117 | 5.541 | 0.154 | 75.1 | 9.3 | 6.617 | 0.394 |
| 1788-21* E | 0.33 | 50 | 2.041 | 27.135 | 5.793 | 0.141 | – | – | – | – |
| 1788-22* E | 0.12 | 30 | 1.513 | 24.801 | 5.975 | 0.098 | – | – | – | – |
| 1788-23* E | 0.25 | 34 | 1.368 | 17.071 | 5.988 | 0.079 | – | – | – | – |
| 1788-24* E | 0.36 | 90 | 1.237 | 13.555 | 6.167 | 0.055 | – | – | – | – |
| 1788-25* E | 0.41 | 107 | 1.084 | 8.672 | 6.206 | 0.033 | – | – | – | – |
| 1788-26* E | 0.48 | 102 | 1.302 | 11.333 | 6.384 | 0.081 | – | – | – | – |
| 1788-27* E | 0.49 | 165 | 0.686 | 7.733 | 6.367 | 0.037 | – | – | – | – |
| 1788-28* E | 0.31 | 167 | 1.615 | 6.975 | 5.602 | 0.246 | – | – | – | – |

*Table 4 continued on next page*

*Table 4 continued*

| Sample1767 | U (ppm) | U/Th | 230Th/238U | 2σ | 234U/238U | 2σ | Age (ka) | 2s (ka) | (234U/238U)i | 2σ |
|---|---|---|---|---|---|---|---|---|---|---|
| 1788-29* E | 0.44 | 6 | 2.311 | 11.898 | 5.576 | 0.306 | – | – | – | – |
| 1788-30 E | 0.44 | 62 | 0.988 | 5.310 | 6.089 | 0.075 | – | – | – | – |
| 1788-31 E | 0.29 | 8 | 1.066 | 19.256 | 6.151 | 0.056 | – | – | – | – |
| 1788-32 E | 0.23 | 95 | 0.994 | 17.451 | 6.352 | 0.064 | – | – | – | – |
| 1788-33 E | 0.41 | 6 | 1.103 | 21.651 | 6.344 | 0.049 | – | – | – | – |
| 1788-34 E | 0.28 | 51 | 1.340 | 11.450 | 6.382 | 0.061 | – | – | – | – |
| 1788-35 E | 0.35 | 4 | 1.286 | 21.088 | 6.321 | 0.062 | – | – | – | – |
| 1788-36 E | 0.4 | 115 | 1.216 | 12.896 | 6.372 | 0.041 | – | – | – | – |
| 1788-37 E | 0.3 | 61 | 1.106 | 17.059 | 6.313 | 0.073 | – | – | – | – |
| 1788-38 E | 0.54 | 279 | 2.810 | 0.237 | 6.300 | 0.064 | 58.9 | 12.2 | 7.262 | 0.270 |
| Mean: | | | | | | | | | | |
| 1788 D | 5.59 | 2793 | 3.335 | 0.018 | 6.402 | 0.013 | 71.5 | 1.2 | 7.616 | 0.054 |
| 1788 E | 0.78 | 293 | 2.900 | 0.177 | 5.920 | 0.109 | 67.0 | 10.8 | 6.936 | 0.332 |
| Sample1810 | U (ppm) | U/Th | 230Th/238U | 2σ | 234U/238U | 2σ | Age (ka) | 2s (ka) | (234U/238U)i | 2σ |
| 1810-1 D | 7.07 | 348 | 3.231 | 0.021 | 5.814 | 0.017 | 77.9 | 1.6 | 7.003 | 0.056 |
| 1810-2 D | 8.29 | 411 | 3.112 | 0.030 | 5.863 | 0.010 | 73.4 | 2.1 | 6.986 | 0.062 |
| 1810-3 D | 8.88 | 979 | 3.106 | 0.027 | 5.929 | 0.010 | 72.1 | 1.8 | 7.046 | 0.060 |
| 1810-4 D | 9.19 | 833 | 3.049 | 0.044 | 5.993 | 0.011 | 69.4 | 2.6 | 7.079 | 0.066 |
| 1810-5 D | 9.17 | 508 | 2.937 | 0.047 | 5.990 | 0.007 | 66.2 | 2.8 | 7.020 | 0.066 |
| 1810-6 D | 9.12 | 55 | 3.143 | 6.919 | 5.981 | 0.012 | – | – | – | – |
| 1810-7 D | 7.95 | 432 | 3.099 | 0.018 | 5.977 | 0.013 | 71.1 | 1.3 | 7.089 | 0.054 |
| 1810-8 D | 8.84 | 489 | 2.986 | 0.074 | 6.035 | 0.060 | 67 | 4.1 | 7.088 | 0.084 |
| 1810-9 D | 9.39 | 15905 | 3.122 | 0.013 | 5.870 | 0.006 | 73.6 | 1.1 | 6.999 | 0.052 |
| 1810-10 D | 9.78 | 7839 | 3.165 | 0.017 | 5.873 | 0.011 | 74.8 | 1.3 | 7.024 | 0.054 |
| 1810-11 D | 9.03 | 7242 | 3.174 | 0.030 | 5.888 | 0.015 | 74.8 | 2.0 | 7.043 | 0.058 |
| 1810-12 D | 9.53 | 9626 | 3.157 | 0.019 | 5.889 | 0.009 | 74.3 | 1.4 | 7.036 | 0.054 |
| 1810-13 D | 10.19 | 10240 | 3.094 | 0.018 | 5.904 | 0.008 | 72.2 | 1.3 | 7.016 | 0.052 |
| 1810-14 D | 10.64 | 14463 | 3.155 | 0.030 | 5.958 | 0.010 | 73.1 | 1.9 | 7.099 | 0.058 |
| 1810-15 E | 0.005 | 1 | −0.384 | 146.036 | 1.965 | 0.186 | – | – | – | – |
| 1810-16 E | 0.002 | 2 | −1.060 | 48.168 | 1.014 | 0.108 | – | – | – | – |
| 1810-17 E | 0.004 | 1 | 5.403 | 17.573 | 2.357 | 0.194 | – | – | – | – |
| 1810-18 E | 0.24 | 55 | 3.195 | 9.109 | 4.054 | 0.062 | – | – | – | – |
| 1810-19 E | 0.54 | 426 | 4.009 | 0.186 | 5.054 | 0.109 | 130.8 | 21.8 | 6.872 | 0.466 |
| 1810-20 E | 0.85 | 328 | 3.625 | 0.119 | 4.287 | 0.137 | 146.8 | 22.7 | 5.984 | 0.524 |
| 1810-21 E | 0.41 | 48 | 5.009 | 4.474 | 4.494 | 0.186 | – | – | – | – |
| 1810-22 E | 0.15 | 7 | 7.690 | 8.225 | 4.349 | 0.351 | – | – | – | – |
| 1810-23 E | 0.03 | 2 | 9.912 | 1.843 | 5.153 | 0.599 | – | – | – | – |
| 1810-24 E | 0.01 | 0 | 0.661 | 166.056 | 1.877 | 0.144 | – | – | – | – |
| 1810-25 E | 0.02 | 2 | 7.408 | 9.386 | 4.964 | 0.557 | – | – | – | – |
| 1810-26* E | 0.73 | 3 | 3.762 | 4.100 | 4.735 | 0.100 | – | – | – | – |
| 1810-27* E | 0.14 | 3 | 5.271 | 4.510 | 4.255 | 0.112 | – | – | – | – |
| 1810-28* E | 0.18 | 5 | 3.541 | 8.079 | 4.562 | 0.081 | – | – | – | – |
| 1810-29* E | 0.25 | 9 | 3.511 | 7.095 | 4.562 | 0.058 | – | – | – | – |

*Table 4 continued*

| Sample1767 | U (ppm) | U/Th | 230Th/238U | 2σ | 234U/238U | 2σ | Age (ka) | 2s (ka) | (234U/238U)i | 2σ |
|---|---|---|---|---|---|---|---|---|---|---|
| 1810-30* E | 0.21 | 3 | 4.027 | 5.291 | 4.073 | 0.149 | – | – | – | – |
| 1810-31* E | 0.09 | 1 | 3.875 | 39.136 | 4.029 | 0.072 | – | – | – | – |
| 1810-32* E | 0.05 | 2 | 2.469 | 9.099 | 4.187 | 0.072 | – | – | – | – |
| 1810-33* E | 0.06 | 1 | 2.602 | 24.169 | 4.426 | 0.141 | – | – | – | – |
| 1810-34* E | 0.91 | 2010 | 3.131 | 0.068 | 4.561 | 0.032 | 105.2 | 7.0 | 5.798 | 0.128 |
| 1810-35 E | 1.01 | 3 | 4.291 | 15.514 | 4.085 | 0.025 | – | – | – | – |
| 1810-36 E | 0.04 | 99 | 6.297 | 8.890 | 4.060 | 0.242 | – | – | – | – |
| 1810-37 E | 0.14 | 528 | 5.753 | 3.932 | 4.385 | 0.234 | – | – | – | – |
| 1810-38 E | 0.02 | 55 | 5.687 | 28.490 | 4.211 | 0.437 | – | – | – | – |
| 1810-39 E | 0.01 | 17 | 4.203 | 31.048 | 4.474 | 0.314 | – | – | – | – |
| 1810-40 E | 2.09 | 1586 | 3.993 | 0.049 | 4.993 | 0.037 | 132.5 | 6.2 | 6.814 | 0.146 |
| Mean: | | | | | | | | | | |
| 1810 D | 9.07 | 5332 | 3.107 | 0.030 | 5.922 | 0.0145 | 72.3 | 1.9 | 7.040 | 0.060 |
| 1810 E | 1.10 | 1088 | 3.690 | 0.105 | 4.724 | 0.0788 | 128.8 | 14.4 | 6.595 | 0.316 |
| Sample1841 | U (ppm) | U/Th | 230Th/238U | 2σ | 234U/238U | 2σ | Age (ka) | 2s (ka) | (234U/238U)i | 2σ |
| 1841-1 E | 2.51 | 78 | 4.415 | 3.252 | 5.851 | 0.035 | – | – | – | – |
| 1841-2 E | 1.96 | 51 | 4.268 | 7.631 | 5.842 | 0.044 | – | – | – | – |
| 1841-3 E | 2.37 | 218 | 4.319 | 0.041 | 5.871 | 0.021 | 115.5 | 3.5 | 7.758 | 0.090 |
| 1841-4 E | 1.88 | 350 | 4.261 | 0.046 | 5.891 | 0.016 | 112.6 | 3.6 | 7.730 | 0.082 |
| 1841-5 E | 2.5 | 214 | 4.378 | 0.045 | 5.846 | 0.032 | 118.7 | 4.3 | 7.784 | 0.124 |
| 1841-6 E | 2.5 | 12 | 4.428 | 2.744 | 5.881 | 0.044 | – | – | – | – |
| 1841-7 E | 2.4 | 63 | 4.484 | 1.744 | 5.946 | 0.044 | – | – | – | – |
| 1841-8 E | 2.14 | 47 | 4.499 | 2.467 | 5.962 | 0.037 | – | – | – | – |
| Mean: | | | | | | | | | | |
| 1841 E | 2.25 | 261 | 4.319 | 0.044 | 5.869 | 0.023 | 115.6 | 3.8 | 7.757 | 0.099 |

deep within a N-trending fracture, 8 m N of the excavation pit. The ~ 50 ka flowstones originate from a W-trending fracture, 6 m W of the entry shaft where it overlies sub-unit 1a and sub-unit 3b sediments. The 24–32 ka flowstones all originate from around the area where the excavation pit is located at the intersection point of three fracture sets (*Figure 1b*). The youngest flowstone sample comes from below an active drip point, 1.5 m E of the excavation pit, and similar actively forming flowstones can be seen in other parts of the chamber. The flowstone age groupings indicate that episodic wet periods in the Dinaledi Chamber alternated with periods during which no flowstone was deposited.

## U-Th analyses on teeth

U-Th disequilibrium analyses of four tooth samples were conducted to constrain U uptake models into dental tissues used in ESR dating. The analyses were also used to provide apparent U-Th age estimates (*Tables 4* and *5*). Analyses of all four teeth (samples 1767, 1788 and1810 from *H. naledi*, and sample 1841 from *Papio sp.)* were performed at the University of Wollongong (UoW), in collaboration with Southern Cross University (SCU). Duplicate analyses of two of the tooth samples (samples 1788 and 1810) were conducted at Griffith University (GU) in collaboration with the Australian National University (ANU).

**Table 5.** Summary table of U-Th disequilibrium ages obtained for two *H. naledi* teeth (samples 1788 and 1810) from the Dinaledi Chamber obtained by GU-ANU. No age calculations were carried out for U concentrations of ≤0.5 ppm or U/Th ≤250 (indicated in red and underlined). Negative U/Th values are due to the Th background being higher than the measured values. Mean values in this table only incorporate values from which meaningful ages could be calculated (indicated in black). All uncertainties are given as 2σ. CS = Closed System; Diff = diffusion (i.e., calculated ages are based on the assumption of continuous diffusion after *Sambridge et al. (2012)*.

| Sample 1810a | U (ppm) | U/Th | $^{230}Th/^{238}U$ | 2σ | $^{234}U/^{238}U$ | 2σ | Age – CS (ka) | 2σ (ka) | Age – Diff (ka) | 2σ (ka) | $(^{234}U/^{238}U)$ i* | 2σ |
|---|---|---|---|---|---|---|---|---|---|---|---|---|
| 1 E | 0.03 | −27 | 3.7113 | 1.2508 | 4.4302 | 0.8881 | n/a | – | – | – | – | – |
| 2 E | 0.02 | −19 | 3.1648 | 0.9390 | 4.2083 | 0.4703 | n/a | – | – | – | – | – |
| 3 E | 0.04 | −33 | 3.0257 | 1.0531 | 5.1220 | 0.4988 | n/a | – | – | – | – | – |
| 4 E | 0.05 | −35 | 3.6352 | 1.3897 | 4.9224 | 0.4912 | n/a | – | – | – | – | – |
| 5 E | 0.19 | −258 | 3.4504 | 0.1965 | 4.8106 | 0.1376 | n/a | – | – | – | – | – |
| 6 D | 6.07 | −2972 | 3.2909 | 0.0666 | 5.9801 | 0.0559 | 77.2 | 2.3 | 87.3 | 2.7 | 7.19 | 0.11 |
| 7 D | 6.10 | −5354 | 3.2618 | 0.0824 | 5.9768 | 0.0312 | 76.3 | 2.6 | 86.2 | 3.3 | 7.17 | 0.08 |
| 8 D | 6.39 | 11436 | 3.3169 | 0.0800 | 5.9827 | 0.0514 | 77.9 | 2.6 | 88.3 | 3.3 | 7.21 | 0.11 |
| 9 D | 6.47 | 6193 | 3.3318 | 0.0899 | 5.9470 | 0.0873 | 79.0 | 3.2 | 89.7 | 3.8 | 7.18 | 0.17 |
| 10 D | 6.65 | −5055 | 3.4985 | 0.1048 | 6.0462 | 0.0403 | 82.5 | 3.4 | 94.4 | 4.5 | 7.37 | 0.11 |
| 11 D | 6.95 | 5149 | 3.5465 | 0.0910 | 6.0531 | 0.0406 | 83.8 | 3.0 | 96.3 | 4.0 | 7.40 | 0.11 |
| 12 D | 7.15 | 3244 | 3.5238 | 0.0997 | 6.0501 | 0.0423 | 83.2 | 3.3 | 95.4 | 4.3 | 7.39 | 0.11 |
| **Mean:** | | | | | | | | | | | | |
| 1–5 E | 0.07 ± 0.06 | | 3.4321 | 0.3003 | 4.7962 | 0.1504 | 112.0 | 15.7 | 137.1 | 25.5 | – | – |
| 6–12 D | 6.54 ± 0.31 | | 3.4018 | 0.0749 | 6.0070 | 0.0428 | 80.1 | 2.5 | 91.1 | 3.2 | 7.27 | 0.11 |

| Sample 1810b | U (ppm) | U/Th | $^{230}Th/^{238}U$ | 2σ | $^{234}U/^{238}U$ | 2σ | Age – CS (ka) | 2σ (ka) | Age – Diff (ka) | 2σ (ka) | $(^{234}U/^{238}U)$ i* | 2σ |
|---|---|---|---|---|---|---|---|---|---|---|---|---|
| 1 E | 0.01 | 17 | 6.8442 | 14.0435 | 0.0228 | 5.5626 | n/a | – | – | – | – | – |
| 2 E | 0.00 | 3 | 9.2330 | 16.2333 | −2.2838 | 4.6085 | n/a | – | – | – | – | – |
| 3 E | 0.00 | −2 | 16.1688 | 27.6564 | −0.1033 | 7.6336 | n/a | – | – | – | – | – |
| 4 E | 0.00 | −3 | 14.9980 | 967.0421 | −0.7695 | 259.2590 | n/a | – | – | – | – | – |
| 5 E | 0.02 | −182 | 7.1338 | 296.1136 | 3.6250 | 48.0742 | n/a | – | – | – | – | – |
| 6 E | 0.86 | −2493 | 4.5176 | 0.1786 | 4.6588 | 0.0795 | 189.1 | 16.7 | 381.3 | 137.9 | 7.24 | 0.44 |
| 7 E | 0.98 | −603 | 4.8797 | 0.1416 | 4.8737 | 0.0681 | 201.4 | 14.0 | 0.0 | 0.0 | 7.84 | 0.40 |
| 8 D | 4.49 | 20423 | 3.5778 | 0.0670 | 5.9327 | 0.0773 | 87.1 | 2.7 | 100.8 | 3.1 | 7.31 | 0.15 |
| 9 D | 5.35 | −10128 | 3.3046 | 0.0659 | 5.9142 | 0.0677 | 78.7 | 2.4 | 89.2 | 2.8 | 7.14 | 0.13 |
| 10 D | 5.67 | −4197 | 3.4077 | 0.0777 | 5.9480 | 0.0459 | 81.3 | 2.6 | 92.8 | 3.3 | 7.23 | 0.10 |
| **Mean:** | | | | | | | | | | | | |
| 1–5 E | 0.01 ± 0.01 | | 8.7750 | 204.6988 | 1.6547 | 93.9630 | n/a | - | - | - | - | - |
| 6–7 E | 0.92 ± 0.12 | | 4.7101 | 0.1465 | 4.7730 | 0.0602 | 195.7 | 13.8 | 471.0 | 269.4 | 7.54 | 0.42 |
| 8–10 D | 5.17 ± 0.70 | | 3.4214 | 0.0790 | 5.9319 | 0.0523 | 82.1 | 2.7 | 93.7 | 3.6 | 7.23 | 0.13 |

| Sample 1788a | U (ppm) | U/Th | $^{230th}/^{238}U$ | 2σ | $^{234}U/^{238}U$ | 2σ | Age – CS (ka) | 2σ (ka) | Age – Diff (ka) | 2σ (ka) | $(^{234}U/^{238}U)$ i* | 2σ |
|---|---|---|---|---|---|---|---|---|---|---|---|---|
| 1E | 0.03 | 4877 | 2.1095 | 3.3058 | 3.5740 | 1.6107 | n/a | – | – | – | – | – |
| 2E | 0.01 | −21 | 3.7845 | 5.5271 | 1.3525 | 2.7713 | n/a | – | – | – | – | – |
| 3E | 0.00 | −1 | 10.5030 | 27.8940 | −2.4909 | 10.1171 | n/a | – | – | – | – | – |
| 4E | 0.00 | −1 | 9.0249 | 113.8912 | −0.7120 | 32.3636 | n/a | – | – | – | – | – |
| 5E | 0.00 | −2 | 6.6795 | 66.2750 | 0.7769 | 18.7506 | n/a | – | – | – | – | – |
| 6E | 0.01 | −6 | 3.0231 | 0.9844 | 2.1904 | 0.4875 | n/a | – | – | – | – | – |
| 7E | 0.24 | −105 | 2.8139 | 0.2076 | 6.3791 | 0.1624 | n/a | – | – | – | – | – |

*Table 5 continued on next page*

*Table 5 continued*

| Sample 1810a | U (ppm) | U/Th | $^{230}$Th/$^{238}$U | 2σ | $^{234}$U/$^{238}$U | 2σ | Age – CS (ka) | 2σ (ka) | Age – Diff (ka) | 2σ (ka) | ($^{234}$U/$^{238}$U) i* | 2σ |
|---|---|---|---|---|---|---|---|---|---|---|---|---|
| 8E | 0.24 | −204 | 1.6495 | 1.5811 | 6.0123 | 0.2961 | n/a | – | – | – | – | – |
| 9E | 0.19 | 579 | 2.4075 | 4.2620 | 6.3341 | 1.3187 | n/a | – | – | – | – | – |
| 10E | 0.48 | 189 | 3.1717 | 2.2862 | 6.1006 | 0.1341 | n/a | – | – | – | – | – |
| 11E | 1.34 | 13833 | 3.8792 | 0.2864 | 6.3521 | 0.1024 | 88.5 | 9.2 | 102.9 | 12.6 | 7.87 | 0.32 |
| 12E | 2.57 | 755 | 4.1770 | 0.0609 | 6.3275 | 0.0972 | 98.6 | 3.0 | 117.7 | 3.1 | 8.04 | 0.19 |
| Mean: | | | | | | | | | | | | |
| 1–6 E | 0.01 ± 0.01 | | 3.5188 | 4.4531 | 2.3471 | 7.1463 | n/a | - | - | - | – | – |
| 7–10 E | 0.29 ± 0.13 | | 2.6484 | 0.2469 | 6.1824 | 0.0978 | 56.2 | 6.5 | 61.0 | 7.7 | – | – |
| 11–12 E | 1.96 ± 1.23 | | 4.0746 | 0.0941 | 6.3361 | 0.0482 | 95.1 | 3.2 | 112.3 | 4.7 | 7.96 | 0.26 |

| Sample 1788b | U (ppm) | U/Th | $^{230}$Th/$^{238}$U | 2σ | $^{234}$U/$^{238}$U | 2σ | Age – CS (ka) | 2σ (ka) | Age – Diff (ka) | 2σ (ka) | ($^{234}$U/$^{238}$U) i* | 2σ |
|---|---|---|---|---|---|---|---|---|---|---|---|---|
| 1E | 0.02 | 4 | 2.3945 | 1.8774 | 3.5040 | 1.4368 | n/a | – | – | – | – | – |
| 2E | 0.02 | 14 | 1.9656 | 1.3299 | 3.3099 | 0.9022 | n/a | – | – | – | – | – |
| 3E | 0.01 | −8 | 2.8156 | 2.1034 | 2.5359 | 0.8082 | n/a | – | – | – | – | – |
| 4E | 0.02 | 160 | 2.1024 | 58.1854 | 3.2342 | 7.4009 | n/a | – | – | – | – | – |
| 5E | 0.03 | −31 | 2.3859 | 1.4084 | 4.1285 | 1.5222 | n/a | – | – | – | – | – |
| 6E | 0.03 | −20 | 2.8951 | 11.2911 | 4.1046 | 3.9747 | n/a | – | – | – | – | – |
| 7E | 0.02 | −10 | 2.8486 | 3.5383 | 4.9362 | 2.3343 | n/a | – | – | – | – | – |
| 8E | 0.03 | −18 | 2.8325 | 1.6113 | 5.7052 | 0.7139 | n/a | – | – | – | – | – |
| Mean: | | | | | | | | | | | | |
| 1–8 E | 0.02 ± 0.01 | | 2.5597 | 6.7618 | 4.1308 | 1.2209 | n/a | – | – | – | – | – |

| Sample 1788c | U (ppm) | U/Th | $^{230}$th/$^{238}$U | 2σ | $^{234}$U/$^{238}$U | 2σ | Age – CS (ka) | 2σ (ka) | Age – Diff (ka) | 2σ (ka) | ($^{234}$U/$^{238}$U) i* | 2σ |
|---|---|---|---|---|---|---|---|---|---|---|---|---|
| 1D | 5.44 | 21578 | 3.9281 | 0.0707 | 6.4260 | 0.0740 | 88.6 | 2.6 | 103.0 | 3.1 | 7.97 | 0.15 |
| 2D | 5.39 | 155037 | 3.8908 | 0.0565 | 6.4416 | 0.0524 | 87.2 | 2.0 | 101.0 | 2.4 | 7.96 | 0.11 |
| 3D | 4.95 | 1708 | 3.8901 | 0.0828 | 6.4085 | 0.0792 | 87.8 | 3.0 | 102.0 | 3.6 | 7.93 | 0.16 |
| 4D | 3.87 | 1653 | 3.8033 | 0.0859 | 6.3786 | 0.1068 | 85.8 | 3.3 | 99.0 | 3.6 | 7.85 | 0.20 |
| 5D | 4.25 | 1168 | 3.9569 | 0.0800 | 6.4051 | 0.0957 | 90.0 | 3.1 | 105.0 | 3.6 | 7.97 | 0.19 |
| 6D | 5.12 | 1493 | 3.9433 | 0.0579 | 6.4951 | 0.0961 | 87.8 | 2.5 | 102.0 | 2.5 | 8.04 | 0.17 |
| 7D | 5.34 | 2659 | 3.8020 | 0.0581 | 6.4713 | 0.0581 | 84.0 | 2.0 | 96.7 | 2.4 | 7.94 | 0.11 |
| 8D | 5.06 | 1093 | 3.9948 | 0.0672 | 6.4479 | 0.0630 | 90.3 | 2.4 | 105.5 | 3.0 | 8.03 | 0.13 |
| 9D | 4.78 | 1018 | 4.0481 | 0.0718 | 6.4468 | 0.0586 | 92.0 | 2.5 | 108.0 | 3.3 | 8.06 | 0.13 |
| 10D | 5.22 | 817 | 3.9011 | 0.0582 | 6.5187 | 0.0813 | 86.1 | 2.3 | 99.6 | 2.4 | 8.04 | 0.15 |
| 11D | 5.25 | 425 | 3.8872 | 0.0850 | 6.4415 | 0.0624 | 87.1 | 2.8 | 101.0 | 3.6 | 7.96 | 0.14 |
| 12D | 5.46 | 345 | 3.9561 | 0.0584 | 6.4658 | 0.0733 | 88.8 | 2.3 | 103.3 | 2.5 | 8.03 | 0.14 |
| Mean: | | | | | | | | | | | | |
| 1–12 D | 5.01 ± 0.28 | | 3.9175 | 0.0796 | 6.4479 | 0.0461 | 87.9 | 2.6 | 102.0 | 3.6 | 7.98 | 0.06 |

Summaries of the U-Th analytical data and ages are reported in *Table 4* (SCU-UoW) and *Table 5* (GU-ANU). In *Table 4* only closed system dates are reported, while *Table 5* also lists dates based on the continuous diffusion model of *Sambridge et al. (2012)*. In both datasets, the U content in enamel is much lower than in dentine. Note that apparent U-Th ages for the teeth are likely to

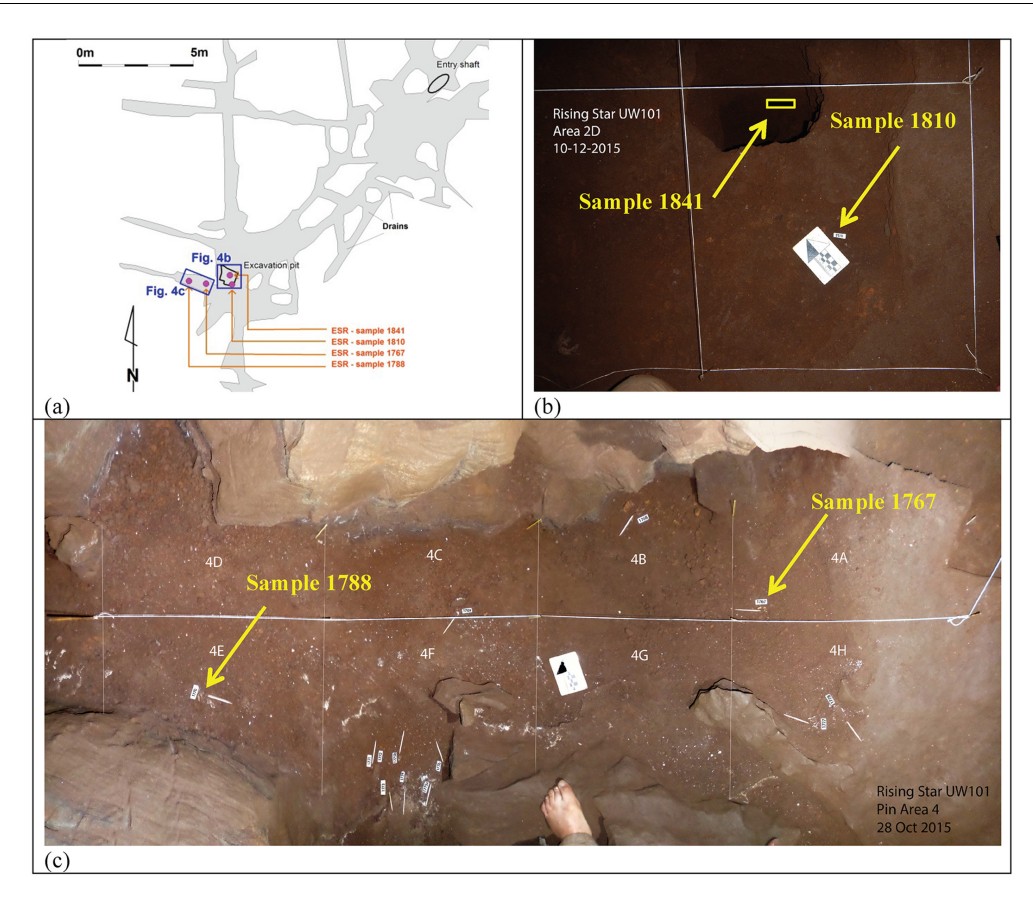

**Figure 4.** Location of the three *H. naledi* tooth samples (samples 1767, 1788 and 1810) and one baboon (cf. *Papio*) tooth sample (sample 1841) used for combined U-series and ESR dating. (a) Map of the Dinaledi Chamber showing the position of the excavation pit and the position of figures (b) and (c); (b) close-up of the SE corner of the excavation pit showing the sample site for sample 1810 and the 50 cm deep sondage from which sample 1841 was recovered; (c) the area to the W of the excavation pit from which samples 1767 and 1788 were collected. DOI: 10.7554/eLife.24231.011

provide apparent age estimates, which will approach the age for U uptake events that affected the teeth during wet periods in the chamber, typically after deposition. These ages should, therefore, be regarded as minimum age estimates for the teeth, and do not represent depositional ages for the fossils.

**Sample 1767:** This extremely worn upper premolar crown (*Figure 6a*) is heavily weathered, and only a small fragment of enamel was left attached to the dentine. It could, therefore, only be dated once (at SCU-UoW). Both dentine and enamel yield consistent results with apparent U-Th ages of 44.5 ± 0.6 ka for dentine and 46.1 ± 0.7 ka for enamel, and initial $^{234}U/^{238}U$ activity ratios at 6.99 ± 0.01 and 6.99 ± 0.04 respectively. These results suggest that a single uptake event is dated. The tooth is characterised by an extremely high U content in the enamel when compared to the other teeth (*Table 4*). Uranium concentration gradients show the effects of diffusion into the enamel from all external surfaces, with enrichment at the Enamel Dentine Junction (EDJ).

**Sample 1788:** This lower right second molar was covered by a thin layer of sediment and is well-preserved (*Figure 6b*). Uranium concentrations in enamel and dentine vary across the surface, with minor hotspots and leaching zones near enamel cracks and along the dentine canal. The EDJ is enriched in U, showing a diffusion gradient into the enamel, and resulting in elevated U concentrations in spots located close to the EDJ (*Figure 9*). The U uptake history appears complex and heterogeneous, and probably involved several episodes. Most of the U concentrations in the enamel are too low to provide a meaningful age. However, parts of the enamel and the dentine yielded

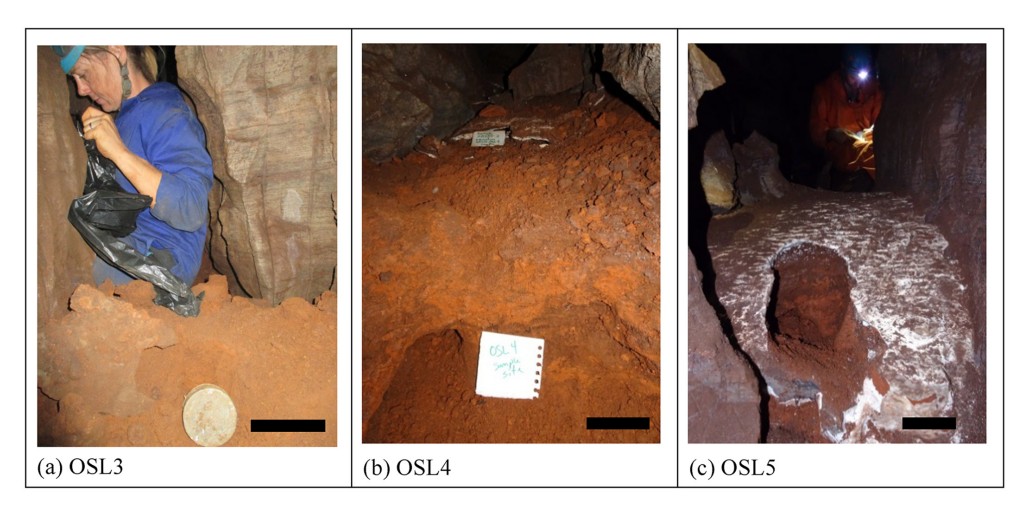

**Figure 5.** Samples of orange laminated mudstone of Unit 1 for OSL dating. (**a**) sample OSL3 with an estimated MAM age of 231 ± 41 ka taken from sub-unit 1b; (**b**) sample OSL4 with an estimated MAM age of 241 ± 37 ka, taken from sub-unit 1b and covered by a flowstone sheet dated at 290 ± 6 ka (RS5); (**c**) sample OSL5 with an estimated MAM age of 353 ± 61 ka, taken from sub-unit 1a and covered by a flowstone sheet dated at 32.1 ± 0.4 ka (RS20). The scale bar in each of the photographs is 10 cm.

consistent measurements for which SCU-UoW provide ages with mean values of 71.5 ± 0.6 ka for dentine and 67.0 ± 5.4 ka for enamel with initial $^{234}U/^{238}U$ activity ratios of 7.62 ± 0.03 and 6.94 ± 0.17 respectively. GU-ANU obtained a combined age of 95.1 ± 3.2 ka for two enamel spots with higher U concentrations, with parts of the enamel with lower U-enrichment yielding a combined age of 56.2 ± 6.5 ka. GU-ANU also provide a consistent mean apparent age of 87.9 ± 2.6 ka, associated with initial $^{234}U/^{238}U$ activity ratios of 7.98 ± 0.06 (individual spots agreeing within error) for dentine which is interpreted as the age of an U uptake event.

**Sample 1810:** This lower left molar or premolar from the excavation pit (*Figures 4* and *6c*), is near complete and only moderately weathered. Uranium diffusion patterns show U accumulating at the EDJ with slow diffusion into the enamel tissue. The U concentrations in most of the enamel are too low to calculate a meaningful age. Areas of enamel with higher U concentrations return older ages (*Tables 4* and *5*). SCU-UoW provide a mean age for high-U spots in enamel of 128.8 ± 7.2 ka and an associated initial $^{234}U/^{238}U$ activity ratio of 7.60 ± 0.16. GU-ANU report a mean apparent U-Th age of 195.7 ± 13.8 ka, which is much higher than the adjacent dentine spots (at 81.1 ± 2.7 ka), but is coupled with realistic initial $^{234}U/^{238}U$ activity ratios of 7.24 and 7.84 with overlapping error limits. This indicates that a secondary overprint of the dentine took place for which the U source had a similar $^{234}U/^{238}U$ composition as the source of the initial U uptake event. Dentine measurements are consistent along the measured sections with small regions affected by leaching and enrichment near cracks and the pulp cavity. The dentine analyses done by SCU-UoW (*Table 4*) yield similar ages with consistent initial $^{234}U/^{238}U$ activity ratios of around 7.04 ± 0.03, and a mean apparent age of 72.3 ± 1.0 ka. The combined analytical data for dentine from GU-ANU in samples 1810A and 1810B yield apparent U-Th ages of 80.1 ± 2.5 ka and 82.1 ± 2.7 ka respectively, coupled with consistent initial $^{234}U/^{238}U$ activity ratios (*Table 5*).

**Sample 1841** The baboon tooth consists of an enamel crown that is structurally intact, but the enamel is friable and weathered (*Figure 7*). The U distribution within the enamel appears homogenous, however, Th concentrations are low and the resolution of the elemental distribution is poor, which impairs the quality of the U-Th age estimates. A recent U uptake event may have occurred affecting enamel in contact with sediment, resulting in a mean apparent U-Th age estimate of 115.6 ± 1.9 ka with a mean initial $^{234}U/^{238}U$ activity ratio of 7.76 ± 0.05 (*Table 4*).

GU-ANU also provide age estimates in which the continuous diffusion assumptions of *Sambridge et al. (2012)* have been applied. The results obtained for samples 1788 and 1810 are

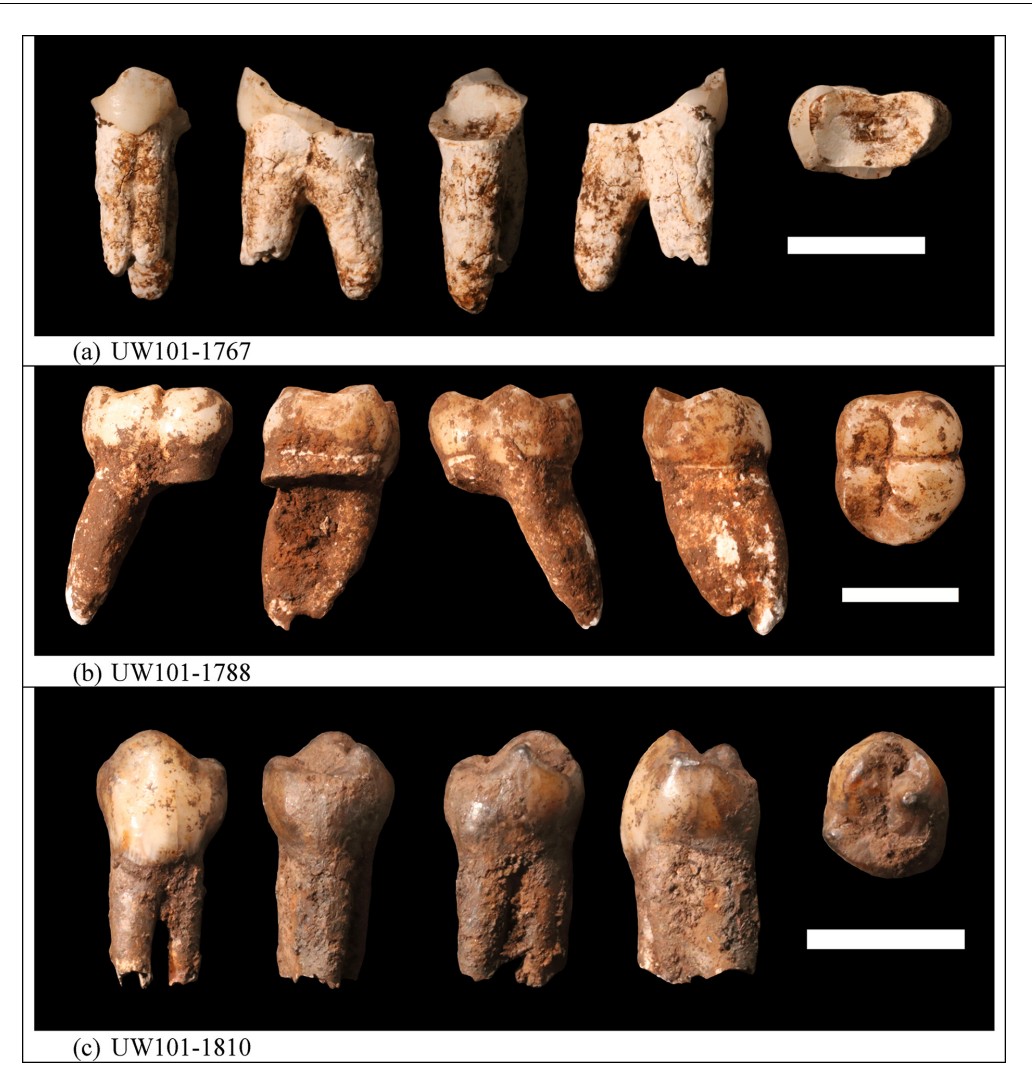

**Figure 6.** Photographs of *H. naledi* teeth used for ESR dating. (a) U.W.101–1767; (b) U.W.101–1788; (c) U.W.101–1810. The order of images for each panel is: buccal, distal, lingual, mesial, and occlusal views. The scale bar in each panel is 1 cm.

about 20% older than the closed system ages (*Table 5*), but show much less consistency and are not further considered.

Collectively, the results show that the teeth are older than 70 ka, and considering sample 1810 with a minimum age of around 200 ka, that the *H. naledi* fossils are probably older than 200 ka (see Discussion).

## US-ESR dating

Combined U-series and ESR dating (US-ESR; *Grün et al., 1988*) of three hominin teeth (samples 1767, 1788 and 1810) and a baboon tooth (sample 1841) was performed at SCU. Blind duplicate analyses of two of the hominin samples (samples 1788 and 1810) were performed at the 'Centro Nacional de Investigación sobre la Evolución Humana' (CENIEH), Spain in collaboration with GU (CENIEH-GU). In obtaining the ages, each laboratory carried out independent sample preparation, and ESR and U-series analyses of the fossil teeth. Estimates for the environmental dose rates used in the age calculations were standardized for both laboratories (*Table 6*) in order to produce comparable results (see discussion and methodology sections for details).

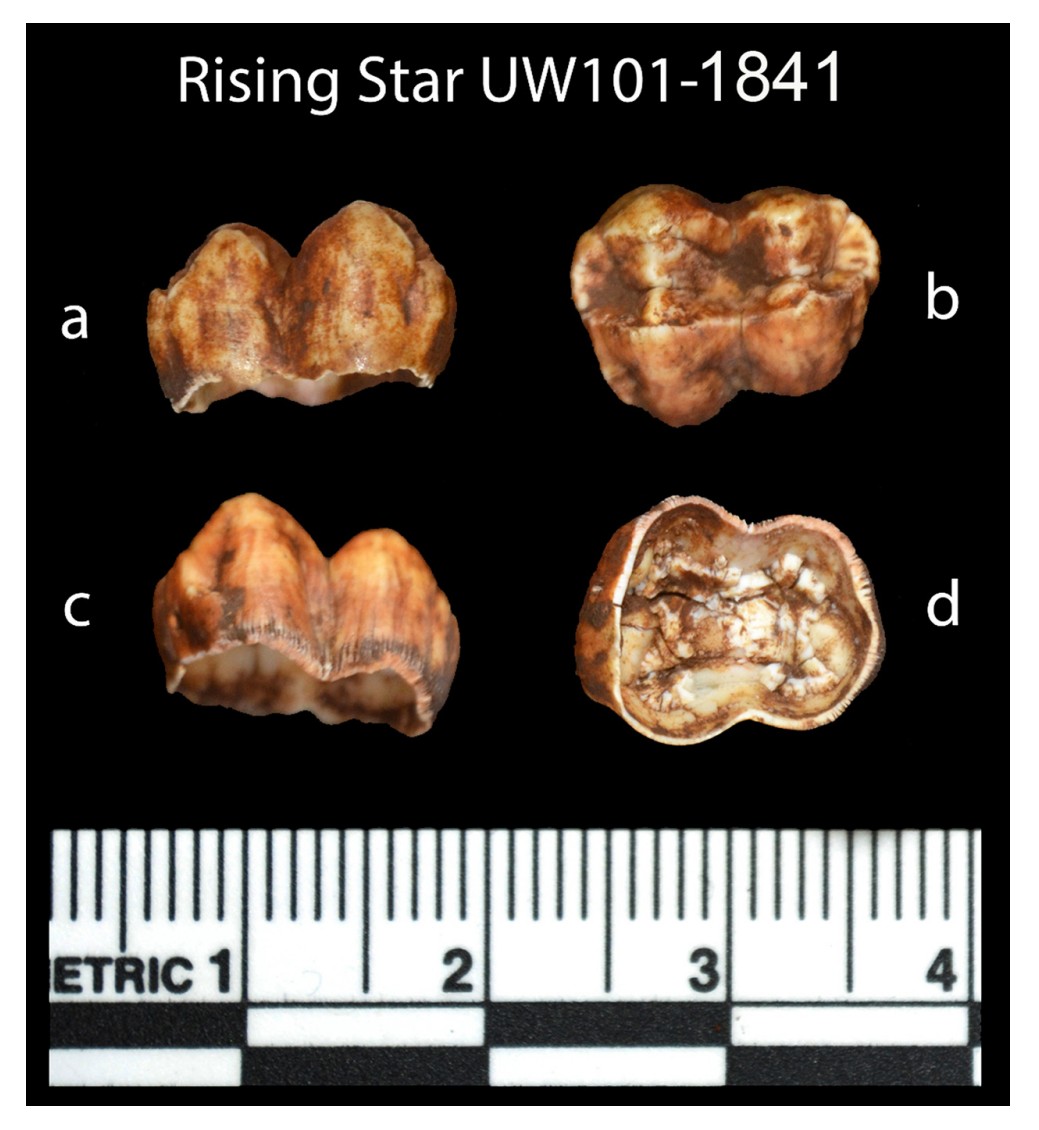

**Figure 7.** Photographs of the baboon (cf. *Papio*) tooth (sample 1841), recovered from the sondage in the excavation pit, and used for ESR dating. Views are: (**a**) buccal, (**b**) occlusal, (**c**) lingual, and (**d**) internal.

Analyses and results from both labs are presented in *Table 7* and *Figures 10*, *11* and *12*. Results are presented for two scenarios: scenario 1 in which the teeth are fully covered in sediment that contains 25 ± 10% water and experienced 80% Rn loss; and scenario 2 in which the teeth are fully covered in sediment that contains 25 ± 10% water and experienced no Rn loss. Scenario 1 reflects the measured present-day situation and is interpreted as a maximum age estimate. Scenario 2 provides a minimum age estimate (*Table 7*). Together these scenarios provide our best estimate for the age range of the fossil teeth.

Combined US-ESR ages determined by SCU for samples 1810, 1788 and 1767 under scenario 1 conditions (i.e., the maximum age scenario) are 284 ± 51 ka, 247 ± 41 ka and 104 ± 29 ka (2σ uncertainty), respectively (*Table 7*). Combined US-ESR ages determined by CENIEH-GU for samples 1810 and 1788 under scenario 1 conditions are 267 ± 68 ka and 211 ± 28 ka (2σ uncertainty), respectively (*Table 7*).

Combined US-ESR ages determined by SCU for samples 1810, 1788 and 1767 under scenario 2 conditions (i.e., the minimum age scenario) are 230 ± 40 ka, 194 ± 34 ka and 87 ± 22 ka (2σ

uncertainty), respectively (*Table 7*). Combined US-ESR ages determined by CENIEH-GU for samples 1810 and 1788 under this scenario are 210 ± 50 ka and 163 ± 24 ka (2σ uncertainty), respectively (*Table 7*).

Results for sample 1767 are based on anomalously high (~20 times) U concentrations in enamel, and probably yield anomalously low age estimates (*Duval et al., 2012*). This result is, therefore, considered to be unreliable and has been excluded from final age estimates (see Discussion).

The observed difference in age estimates obtained by SCU and CENIEH-GU for samples 1788 and 1810, are most likely explained by natural dose variations within the tested enamel layers (see Discussion and methodology sections), and we have no reason to prefer one age result over another. The optimal age estimate for the *H. naledi* fossils, therefore, combines the results from both laboratories with average maximum (i.e., scenario 1) age estimates for samples 1788 and 1810 of 229 + 60/–46 ka and 276 + 59/–77 ka (2 σ uncertainty) respectively, and average minimum (i.e., scenario 2) age estimates of 179 + 49/–40 ka and 220 + 50/–60 ka (2 σ uncertainty) respectively. Together these results provide an age range of 139–335 ka for the *H. naledi* remains, although dating of flowstone encasing *H. naledi* bones helps to better constrain this range (see Discussion).

Combined US-ESR ages for the baboon tooth (sample 1841) determined by SCU using scenario 1 and 2 conditions are 723 ± 181 ka and 635 ± 148 ka respectively. The tooth contained no inner dentine (*Figure 7*), and was filled with sediment. In calculating the age it was, therefore, assumed that sediment occurred on both sides of the enamel layer. Sample 1841 was recovered from sub-unit 3a directly below the occurrence of articulated *H. naledi* remains in the excavation pit (*Dirks et al., 2015*). The age results provide an upper age limit for the deposition of the *H. naledi* bearing layer, and mark an earlier stage of deposition of mud clast breccia in the cave assigned to sub-unit 3a, which predates the entry of *H. naledi* fossils into the cave.

## OSL dating

Optically stimulated luminescence (OSL) dating of three samples of sediment from Unit 1 in the Dinaledi Chamber (samples OSL3 and OSL4 from sub-unit 1b, and sample OSL5 from sub-unit 1a; *Figure 5*) was performed at the University of the Witwatersrand (Wits). The measurements were carried out on small aliquots containing ~30 grains. Summaries of the OSL analytical data and ages are reported in *Table 8*.

The reported dose rates for the samples range from 0.7 to 0.9 Gy ka$^{-1}$ (*Table 8*), with significant within-sample scatter, resulting in uncertainties on age estimates of 15–18%. Overdispersion in $D_e$ ranges from 50–70%, which is much higher than would be expected for a well-bleached sample, and indicates that it is most appropriate to apply a Minimum Age Model (MAM) to the dataset, in which the MAM age is likely to represent a maximum estimate for the age of the sediments (see Discussion). As with ESR, significant Rn loss was detected in the samples of Unit 1, and corrections to the measured U concentrations were applied (*Table 8*). The MAM calculations for the three samples yield maximum age estimates for the sediments of 231 ± 41 ka (OSL3), 241 ± 37 ka (OSL4), and 353 ± 61 ka (OSL5). The MAM apparent ages for OSL3 and OSL4 were obtained from the sandy facies of sub-unit 1b sediments and yield ages that are younger than the age of a Flowstone 1 sheet (sample RS5 at 290 ± 4 ka; *Table 1*) that covers the outcrop of sub-unit 1a from which sample OSL4 was taken (*Figure 2c*). This discrepancy can be attributed to inverted stratigraphy associated with erosion of the top of the older sub-unit 1a after the deposition of Flowstone 1 (RS5) by running water and subsequent deposition of sub-unit 1b between sub-unit 1a and the flowstone (*Figure 2c*). Sample OSL5 was obtained from muddy sediment of sub-unit 1a, and yields an older age than the covering flowstones (RS5 and RS20). Note that if a Central Age Model (CAM) is applied to the OSL data, results are significantly older at 560 ± 102 ka (OSL3), 546 ± 79 ka (OSL4), and 849 ± 132 ka (OSL5), however, this model is considered unrealistic within a cave environment (*Galbraith et al., 1999*).

## Palaeomagnetic analysis of flowstone

Palaeomagnetic analysis of one composite sample of Flowstone 1a (*Figure 13*), covering erosional remnants of Unit 2 near the entry shaft into the Dinaledi Chamber, was performed at La trobe University, Melbourne (LTU). The palaeomagnetic results for Flowstone 1a are presented in *Figure 13* and *Table 9*. The palaeomagnetic sample from Flowstone 1a comprises three distinct phases of

flowstone formation (from base to top: A, B and C; *Figure 13a,b*) that have been analysed for their palaeomagnetic orientation (*Figure 13c*). Note that the palaeomagnetic sample of Flowstone 1a was taken up-dip from the position where the U-Th sample of Flowstone 1a was taken (i.e., RS22 and RS23). The basal, phase A flowstone observed in the palaeomagnetic sample tapers out down-dip (*Figure 13a*) and is not present in the U-Th sample; thus, RS23 at the base of the U-Th sample corresponds to phase B carbonate in the palaeomagnetic sample, and RS22 to phase C carbonate (*Figure 2b*).

Phase A flowstone at the base of the sample records a consistent reversed magnetic polarity, with a clear overprint that is anti-parallel to the characteristic, reversed polarity remanence (ChRM) and is removed by ~10 mT. The reversed polarity ChRM is then stable between 10 and 40 mT, although within this range there are two reversed polarity components that can be identified in some samples (between 10 and 19 mT and then between 20 and 40 mT; *Figure 13c*). Both components have almost identical declination values, but the lower field component has a shallower inclination. This is not uncommon within speleothems (*Herries and Shaw, 2011*) because a single subsample of 2.5 cm depth is measuring the remanence recorded in multiple layers of speleothem as well as multiple layers of detrital contamination. Phase B flowstone records a weak, normal polarity ChRM that is consistently isolated between 7 and 36mT (*Figure 13c*). The results from this phase have the highest mean maximum angular deviation (MAD) values (*Table 9*) due to the small and oddly shaped nature of the samples that provide less consistent measurements between each spin in the magnetometer. Phase C flowstone also records a normal ChRM that is similar to that seen within phase B (*Figure 13c*), although with slightly steeper inclinations (*Figure 13c*, Stereo Plot).

The samples do not have strong secondary overprints as seen in many Plio-Pleistocene palaeo-cave deposits from the region (*Dirks et al., 2010*), which may indicate that they are younger or have a different sediment source and, thus, mineralogy holding the remanence. The coercivity of phase B and C flowstones is similar, and distinct from phase A flowstone, although all the demagnetisation spectra suggest the dominant mineral holding the remanence is ferrimagnetic (magnetite/maghaemite) as at many other CoH sites (*Herries et al., 2006*; *Herries and Shaw, 2011*; *Herries et al., 2014*).

### Radiocarbon dating

Three weathered bone fragments of *H. naledi* were analysed via radiocarbon dating at Beta Analytic Inc. (Florida, USA). Analyses indicated that no collagen was present in any of the samples and that the bone appeared possibly cremated. This was investigated with a bone carbonate extraction technique. Tests did not support cremation and indicated that extensive secondary $CaCO_3$ replacement had occurred, providing ages of 33.0 ± 0.2 ka and 35.50 ± 0.16 ka for two of the fragments. We interpret these ages to relate to late calcite precipitation in the bones that may reflect a wet period in the cave.

## Discussion

The strategy in dating the fossils has been built on three components: (i) construct a detailed stratigraphic model for the cave sediments in the Dinaledi Chamber in which the position of the hominin-bearing deposits can be securely placed; (ii) date the sedimentary units that potentially bracket the hominin-bearing deposits; and (iii) date the *H. naledi* fossils directly. We will start this discussion by commenting on the reliability of the various dating techniques, and how they should be viewed relative to each other. We will then discuss the outcomes of each of the three strategic components, and their implications for the age of the *H. naledi* material. The broader implications of the age for the morphological evolution of hominins in southern Africa is discussed in detail in *Berger et al., 2017*, and will be briefly summarized at the end of this discussion.

### Reliability of the age estimates

The dating techniques applied in this study do not all work in the same way, and hence the results must be viewed differently. U-series analysis on carbonates, [14]C analysis of bone and palaeomagnetic analyses of flowstones (e.g., *Herries and Shaw, 2011*; *Taylor and Bar-Yosef, 2014*; *Hellstrom and Pickering, 2015*) are well established dating techniques requiring few, if any, à priori

assumptions. In contrast, ESR and OSL results are strongly dependent on model assumptions for the environmental conditions that affected the locations from which the samples were taken.

## U-Th results

Sample preparation and analytical procedures for U-Th disequilibrium dating are well established (see methodology section), and are likely to return highly reproducible (i.e., high precision) results. This is certainly the case here, where samples analysed in duplicate in independent laboratories (JCU and UoM) returned identical results within analytical uncertainties (*Tables 1*, *2* and *3*). The only exception is that UoM was able to provide ages for older flowstones that date close to the detection limit for the technique, whereas JCU did not provide ages over 400 ka.

Although these results demonstrate high levels of precision, further evaluation of the geological meaning of the ages is required. In particular, the effects of possible post-depositional U uptake, which would result in apparent ages that are younger than the true age, can be assessed with textural analyses (e.g., *Pickering et al., 2010*), and the use of initial $^{234}U/^{238}U$ ratios (*Tables 2* and *3*; *Kronfeld et al., 1994*). Most of the analysed flowstones display similar initial $^{234}U/^{238}U$ ratios (1.8 to 2.4), with two notable exceptions in samples RS18 (243 ± 7 ka) and RS5 (290 ± 6 ka) that record $^{234}U/^{238}U$ ratios of 3 to 4. These are two of the oldest samples analysed, and both display evidence for recrystallization and secondary dissolution. Considering that the groundwater reservoir in the CoH has high (i.e., >7) initial $^{234}U/^{238}U$ ratios (*Kronfeld et al., 1994*) it is likely that the elevated $^{234}U/^{238}U$ ratios in samples RS5 and RS18 are due to U uptake during interaction with groundwater after flowstone deposition. In this case, these two age results should be viewed as minimum age estimates for the flowstones.

As with flowstone, analytical procedures for U-Th disequilibrium dating of teeth are well established (e.g., *Grün et al., 2014*). However, in this study there are significant differences in the U-Th disequilibrium ages of teeth reported by the two laboratories (*Tables 4* and *5*). This difference in measured ages is most likely due to the fact that the different laboratories dated different fragments of the teeth. The U-Th analyses show that the U distribution within the teeth, and especially within enamel, is highly variable, as reflected in the wide range of U-Th ages (*Tables 4* and *5*). Where a tooth displays consistent ages and initial $^{234}U/^{238}U$ ratios over a large domain, or where we have observed close coincidence in U-Th ages and initial $^{234}U/^{238}U$ ratios for dentine and adjacent enamel samples, we interpret the results to suggest that a U uptake event has been dated. Considering the variable distribution of U within the tooth samples, it is possible that the same tooth records more than one U uptake event in different domains within the tooth (most notably when comparing enamel vs dentine domains; *Grün et al., 2014*). Since Th is immobile, each U uptake event that affects the tooth will shift pre-existing U-Th systematics to portray a younger age (*Grün et al., 2014*). The variable U-Th disequilibrium ages are a clear indication that U uptake events took place, and provide a minimum age estimate for the true age of the teeth. Where apparent ages are consistent across much of the tooth (e.g., sample 1767) they may approach the true age of the U uptake events. The results presented in this study show that the hominin teeth are clearly older than 70 ka, as evidenced from the consistent results on the dentine of samples 1788 and 1810. An important result from this method is the older closed system age (ca. 200 ka) from enamel obtained for sample 1810B by GU-ANU (*Table 5*). This result indicates that there was an early phase of U uptake in the tooth that was later overprinted in the associated dentine. In this case, sample 1810 has a minimum age of around 200 ka, suggesting that the *H. naledi* fossils are older than 200 ka.

## US-ESR and OSL results

ESR and OSL dating techniques are dosimetric dating techniques (*Adamiec and Aitken, 1998*; *Murray and Wintle, 2000*), and deliver results that are generally much more variable than U-series dating due to the relative lack of control of the conditions under which the samples accumulated radiation damage. In addition, analytical protocols between labs may vary significantly (e.g., SCU used an X-ray gun whilst CENIEH-GU used gamma-rays as their irradiation source), and final results are heavily dependent on a range of à priori model assumptions (such as water content, Rn loss or burial history through time). For OSL, similar à priori model assumptions must be made, and the interpretation of results is further hindered by the use of composite samples in which individual grains may have experienced different burial histories (*Galbraith et al., 1999*, *2005*). Single grain

analyses are possible (*Duller, 2008*), but far more labour intensive and, for the purpose of this study, were not required. Moreover, this is the first time that OSL has been applied to cave deposits in the CoH, and there is no comparative data available for this area. However, the results are consistent with the interpreted stratigraphy and other dating in the chamber, and highlight the potential for additional work.

To assure that all US-ESR results obtained by the different laboratories can be compared in an objective manner, we standardized the à priori model assumptions (*Table 6*), that is, the same analytical results for sediment chemistry, back ground radiation, water content, radon-loss and cosmogenic radiation, needed to calculate the environmental dose rate, were used by both laboratories. Therefore, any differences in age estimates relate to the measured equivalent dose ($D_E$) and laboratory methodologies, and are not the result of model assumptions. Both laboratories also applied the same combinations of model parameters that would most likely result in either maximum or minimum age estimates. These are presented as two scenarios in which the determined age range will overlap with the true age of the fossils (*Table 7*).

When examining the US-ESR results for the teeth (*Table 7*), two trends are apparent: (i) The results for samples 1810 and 1788 from SCU and CENIEH-GU agree well within the listed uncertainties; and (ii) The calculated ages for sample 1767 are much younger than the other teeth. Regarding the second issue, samples 1767 and 1788 were collected from the top of sub-unit 3b within 1.3 m from one another, with the difference that sample 1767 lay directly on surface in the centre of the floor, whereas sample 1788 lay to the side under a cover of 2 cm of loose sediment (*Figure 4*). Given their position in the cave the teeth are expected to be of similar age. However, their ages as obtained by SCU vary by a factor of >2 (scenario 1: 104 ± 29 vs 247 ± 42 ka; scenario 2: 87 ± 22 ka vs 194 ± 34 ka; *Table 7*). The main difference between the two samples is that the internal dose rate of sample 1767 is ~20 times higher (*Table 7*), due to the high U concentration in the enamel (~2.5 ppm). Such high concentrations can lead to dose rate overestimations due to inappropriate alpha efficiency values (*Duval et al., 2012*). The high U values in sample 1767 are probably the result of its greater exposure to water, confirmed by its extremely weathered nature (*Figure 6a*). Given these factors, we do not trust the reliability of the age results for sample 1767 and have excluded this tooth from the final age estimates for the fossils.

The age estimates from the maximum and minimum age scenarios as calculated here are dependent on the amount of estimated Rn loss over time, that is, in assessing the best age estimate for the *H. naledi* fossils a major issue is whether $^{222}$Rn degassing was a process that operated continuously over the past 300 ka or not. $^{222}$Rn is a noble gas, radio-isotope that forms as part of the $^{238}$U decay chain, and its escape has a major effect on the total amount of gamma radiation generated by the sediments (e.g., *Guérin et al., 2011*), and consequently, the calculated US-ESR age (*Table 7*). The reason why $^{222}$Rn degassing occurs so readily probably relates to the crystallographic position occupied by U atoms. Uranium and Th are chiefly hosted in very thin Fe-Mn oxy-hydroxide coatings on grains within the sediments (*Dirks et al., 2015*; *Makhubela et al., 2017*), and given (alpha)-recoil in the $^{226}$Ra decay, escape of $^{222}$Rn (half-life 3.82 days) is likely. This process will have occurred throughout the history of the cave given that the formation of Fe-Mn oxy-hydroxides is an integral part of the chemical process that occurs within the unconsolidated cave sediments as a result of oxidation, auto-brecciation and weathering reactions (*Dirks et al., 2015*). It is, therefore, assumed that effective Rn degassing will have been an important process in the cave throughout the accumulation history of the Unit 3 sediments, although it seems unlikely that an extreme degassing environment similar to the 80% measured today was maintained during the entire history of burial. Therefore, we interpret the maximum US-ESR age estimates calculated for scenario 1 conditions to be the closest to the true age of the *H. naledi* fossils. If Rn degassing was somewhat less in the past, the older age bracket would move into the direction of 270 ka (i.e., the maximum age limit calculated under scenario 2 conditions).

The OSL results are more difficult to interpret than the US-ESR results, not only because we have to assume model parameters to estimate the environmental dose rate, but also because we have few constraints on the origin and provenance of the quartz grains that were sampled. OSL analyses were carried out for aliquots of ~30 grains and each analysis, therefore, is an average of the signals contained in the grains. There is significant within-sample scatter in measured total dose ($D_e$) values between different aliquots, meaning that age estimates are imprecise (uncertainties of 15–18%). Overdispersion in $D_e$ ranges from 50–70%, which is much higher than would be expected for a well-

**Table 6.** Summary table of model parameters used in ESR dating separated by sample number and laboratory. See text for detailed discussion.

| Sample: | 1767 | 1788 | | 1810 | | 1841 |
|---|---|---|---|---|---|---|
| Laboratory: | SCU | SCU | Cenieh-gu | SCU | Cenieh-gu | SCU |
| **Enamel:** | | | | | | |
| De (Gy) | 194 ± 4 | 231 ± 8 | 159 ± 10 | 296 ± 14 | 232 ± 30 | 1676 ± 127 |
| U (ppm) | 2.52 ± 0.53 | 0.38 ± 0.17 | 0.07 ± 0.07 | 0.32 ± 0.12 | 0.16 ± 0.16 | 2.28 ± 0.48 |
| $^{234}U/^{238}U$ | 6.21 ± 0.03 | 5.95 ± 0.32 | 6.258 ± 0.349 | 4.04 ± 0.18 | 4.773 ± 0.060 | 5.87 ± 0.03 |
| $^{230}Th/^{234}U$ | 0.37 ± 0.05 | 0.55 ± 0.52 | 0.598 ± 0.038 | 0.92 ± 0.05 | 0.950 ± 0.034 | 0.785 ± 0.038 |
| Alpha efficiency* | 0.13 ± 0.02 | 0.13 ± 0.02 | 0.13 ± 0.02 | 0.13 ± 0.02 | 0.13 ± 0.02 | 0.13 ± 0.02 |
| Initial thickness (μm) | 1027 ± 210 | 1049 ± 277 | 1486 ± 248 | 1150 ± 250 | 1527 ± 257 | 650 ± 145 |
| Water (%) | 0 | 0 | 0 | 0 | 0 | 0 |
| **Dentine:** | | | | | | |
| U (ppm) | 7.88 ± 0.66 | 5.76 ± 0.86 | 4.71 ± 0.27 | 9.08 ± 0.44 | 5.81 ± 0.37 | – |
| $^{234}U/^{238}U$ | 6.28 ± 0.09 | 6.40 ± 0.03 | 6.448 ± 0.046 | 5.93 ± 0.03 | 5.969 ± 0.035 | – |
| $^{230}Th/^{234}U$ | 0.35 ± 0.11 | 0.62 ± 0.02 | 0.608 ± 0.012 | 0.52 ± 0.09 | 0.572 ± 0.010 | – |
| Water (%) | 10 ± 5 | 10 ± 5 | 10 ± 5 | 10 ± 5 | 10 ± 5 | – |
| **Sediment:** | | | | | | |
| U (ppm) | 3.0 ± 0.3 | 2.9 ± 0.1 | 2.9 ± 0.1 | 3.2 ± 0.3 | 3.2 ± 0.3 | 0.64 ± 0.06[†] |
| Th (ppm) | 7.9 ± 0.4 | 8.3 ± 0.6 | 8.3 ± 0.6 | 8.6 ± 0.4 | 8.6 ± 0.4 | 4.72 ± 0.47[†] |
| K (%) | 1.17 ± 0.14 | 1.21 ± 0.14 | 1.21 ± 0.14 | 1.23 ± 0.14 | 1.23 ± 0.14 | 1.47 ± 0.15[†] |
| Water (%) | 25 ± 10 | 25 ± 10 | 25 ± 10 | 25 ± 10 | 25 ± 10 | 25 ± 10 |
| Depth below ground surface (cm) | 0 | 2 | 2 | 5 | 5 | 55 |
| Gamma Dose rate (μGy a$^{-1}$) | | | | | | |
| *25 ± 10% Water, 80% Rn degassing* <br> *25 ± 10% Water, no Rn degassing* | 534 ± 69 <br> 724 ± 116 | 534 ± 69 <br> 724 ± 116 | 534 ± 69 <br> 724 ± 116 | 534 ± 69 <br> 724 ± 116 | 534 ± 69 <br> 724 ± 116 | 534 ± 69 <br> 724 ± 116 |
| Cosmic dose rate (μGy a$^{-1}$) | 15 ± 1 | 15 ± 1 | 15 ± 1 | 15 ± 1 | 15 ± 1 | 15 ± 1 |

*After **Woodroffe et al. (1991)**;

[†]A relative error of ± 10% was assumed.

bleached sample (**Galbraith et al., 1999**), and is a reflection of the cave environment from which the samples were taken (**Duller, 2008**). We interpret this to mean that the quartz grains in each aliquot consist of a mixture of grains with some derived from outside the cave (and carried into the cave over a period of time), and others derived from quartz veins and chert beds within the cave that have only been partly bleached or have never been bleached at surface at all. Add to this the natural signal variability that can occur due to crystallographic orientation and lattice defects (**Olley et al., 2004**; **Galbraith et al., 1999**, **2005**) and a large overdispersion results. The Minimum Age Model (MAM) was chosen as the preferred statistical model to calculate age estimates for the sediments. This is a statistical averaging technique for aliquots that display a high degree of within-sample scatter (**Olley et al., 2004**; **Galbraith et al., 2005**), but still combines some grains that are well-bleached with others that are only partly bleached. Thus, the MAM age is likely to provide a maximum age estimate for the Unit 1 sediments, and must be viewed with caution. A more comprehensive sampling campaign will be required in future to fully establish the range of ages for Unit 1 across the chamber. Nevertheless, the preliminary results are consistent with the range of ages suggested for Unit 3 by ESR and U-Th analyses on capping flowstones (**Figure 14**).

## An updated stratigraphy for the Dinaledi Chamber

The importance of a deep understanding of the stratigraphic position of the fossils and the geological processes that led to their deposition cannot be overstated considering the extremely complex

**Table 7.** Summary of ESR dating results (2σ uncertainties) for two end-member scenarios: (i) complete burial of the samples, 80% Rn loss in the sediment and post Th-230 equilibrium in dental tissue (i.e., maximum age scenario); (ii) complete burial of the samples and post-Rn equilibrium in sediment (i.e., minimum age scenario). See text for detailed discussion.

| Sample: | 1767 | 1788 | | 1810 | | 1841 |
|---|---|---|---|---|---|---|
| Laboratory: | SCU | SCU | Cenieh-gu | SCU | Cenieh-gu | SCU |
| Scenario 1: 25 ± 10% Water, complete burial and 80% $^{222}$Rn degassing (maximum age scenario) | | | | | | |
| internal dose rate (µGy a$^{-1}$) | 1142 ± 515 | 190 ± 129 | 47 ± 47 | 323 ± 175 | 176 ± 176 | 1411 ± 596 |
| alpha (µGy a$^{-1}$)* | 0 | 0 | 8 ± 2 | 0 | 8 ± 2 | 0[†] |
| beta dose rate, dentine (µGy a$^{-1}$) | 73 ± 33 | 91 ± 62 | 64 ± 16 | 75 ± 41 | 51 ± 14 | –[‡] |
| beta dose rate, sediment (µGy a$^{-1}$) | 101 ± 24 | 105 ± 31 | 86 ± 17 | 95 ± 24 | 86 ± 18 | 358 ± 74 |
| gamma and cosmic (µGy a$^{-1}$) | 549 ± 69 | 549 ± 69 | 549 ± 69 | 549 ± 69 | 549 ± 69 | 549 ± 69 |
| total dose rate (µGy a$^{-1}$) | 1865 ± 521 | 935 ± 162 | 754 ± 87 | 1042 ± 194 | 870 ± 190 | 2318 ± 606 |
| p enamel | −0.03 | 0.49 | −0.02 | −0.70 | −0.77 | 0.91 |
| p dentine | 0.08 | 0.13 | −0.06 | 1.02 | 0.54 | – |
| Age (ka) | 104 ± 29 | 247 ± 42 | 211 ± 28 | 284 ± 51 | 267 ± 68 | 723 ± 181 |
| Combined SCU/CENIEH-GU age (ka) | | 229 + 60/–46 | | 276 + 59/–77 | | |
| Average age for 1788 & 1810 (ka) | | 253 + 82/–70 | | | | |
| Scenario 2: 25 ± 10% Water, complete burial and no $^{222}$Rn degassing (minimum age scenario) | | | | | | |
| internal dose rate (µGy a$^{-1}$) | 1277 ± 552 | 216 ± 165 | 51 ± 51 | 335 ± 193 | 184 ± 184 | 1520 ± 630 |
| alpha (µGy a$^{-1}$)* | 0 | 0 | 8 ± 2 | 0 | 8 ± 2 | 0 |
| beta dose rate, dentine (µGy a$^{-1}$) | 82 ± 35 | 102 ± 78 | 69 ± 18 | 87 ± 50 | 59 ± 16 | – |
| beta dose rate, sediment (µGy a$^{-1}$) | 132 ± 26 | 134 ± 33 | 111 ± 19 | 126 ± 26 | 112 ± 19 | 380 ± 81 |
| gamma and cosmic (µGy a$^{-1}$) | 739 ± 116 | 739 ± 116 | 739 ± 116 | 739 ± 116 | 739 ± 116 | 739 ± 116 |
| total dose rate (µGy a$^{-1}$) | 2230 ± 586 | 1191 ± 219 | 978 ± 129 | 1287 ± 232 | 1102 ± 219 | 2639 ± 647 |
| p enamel | −0.31 | 0.06 | −0.37 | −0.83 | −0.91 | 0.67 |
| p dentine | −0.22 | −0.22 | −0.40 | 0.54 | 0.10 | – |
| Age (ka) | 87 ± 22 | 194 ± 34 | 163 ± 24 | 230 ± 40 | 210 ± 50 | 635 ± 148 |
| Combined SCU/CENIEH-GU age (ka) | | 179 + 49/–40 | | 220 + 50/–60 | | |
| Average age for 1788 & 1810 (ka) | | 200 + 70/–61 | | | | |

*using alpha attenuation values of **Grün (1987)**.

[†]considered as negligible given the low radioelement concentrations in the sediment and the high total dose rate value.

[‡]for 1841, the beta dose rate on both sides of the enamel layer is derived from the sediment.

nature of sedimentary cave fill in many cave systems (e.g., **Wilkinson, 1985**; **Brain, 1993**; **Sasowsky, 1998**; **Stock et al., 2005**; **Stratford et al., 2014**; **Sutikna et al., 2016**) involving repeated cycles of deposition, erosion and reworking, leading to complex and sometimes contradictory age results (e.g., **Granger et al., 2015**; **Kramers and Dirks, 2017**). This problem is well illustrated with the ongoing debate on the age of Stw 573 ('little foot') in the nearby Sterkfontein Cave, where after 20 years of dating efforts no definitive age is yet established (see **Partridge et al., 1999**, **2003**; **Berger et al., 2002**; **Walker et al., 2006**; **Herries and Shaw, 2011**; **Pickering and Kramers, 2010**; **Granger et al., 2015**; **Kramers and Dirks, 2017**). Another good example illustrating the difficulties of linking cave stratigraphy to a definitive age for the hominin fossils they contain is presented by the *H. floresiensis* remains in the Liang Bua cave, Indonesia (**Morwood et al., 2004**; **Roberts et al., 2009**; **Sutikna et al., 2016**).

The stratigraphy within the Dinaledi Chamber has been previously described by **Dirks et al. (2015)**. The ages presented here help to resolve outstanding questions about the stratigraphy in the Dinaledi Chamber, and allow us to more closely define the distribution of correlative stratigraphic units (**Figure 14**), and thus constrain the age of the *H. naledi* fossils.

**Table 8.** Summary of OSL results obtained by the University of the Witwatersrand for samples of Unit 1 from the Dinaledi Chamber (samples OSL3, 4 and 5). The ages were calculated using effective U concentration values (taking disequilibrium into account; see text for details). CAM = Central Age Model; MAM = Minimum Age Model.

| Sample ID | $H_2O$ (%) | Th (ppm) | U (ppm) pre-Rn | U (ppm) post-Rn | K (%) | Total dr (Gy/ka) | $2\sigma$ | Total de (Gy) CAM | $2\sigma$ | Total de (Gy) MAM | $2\sigma$ | CAM Age (ka) | CAM $2\sigma$ | MAM age (ka) | MAM $2\sigma$ | Over dispersion (%) |
|---|---|---|---|---|---|---|---|---|---|---|---|---|---|---|---|---|
| OSL3 | 18.9 ± 5 | 3.71 ± 1.60 | 0.75 ± 0.177 | 0.193 ± 0.044 | 0.45 ± 0.12 | 0.76 | 0.07 | 428.59 | 68.92 | 176.4 | 27.7 | 560 | 103 | 231 | 41 | 63 |
| OSL4 | 25.8 ± 5 | 3.38 ± 1.60 | 0.485 ± 0.177 | 0.097 ± 0.044 | 0.47 ± 0.12 | 0.70 | 0.06 | 379.89 | 43.58 | 168.0 | 20.7 | 546 | 79 | 241 | 37 | 55 |
| OSL5 | 22.7 ± 5 | 5.11 ± 1.60 | 0.692 ± 0.177 | 0.138 ± 0.044 | 0.56 ± 0.12 | 0.90 | 0.07 | 759.54 | 102.33 | 315.67 | 48.68 | 849 | 132 | 353 | 61 | 68 |

The oldest ages returned from the Dinaledi Chamber are from the baboon tooth that is embedded in sediment attributed to sub-unit 3a, followed by U-Th ages for Flowstone 1a, which directly covers erosional remnants of Unit 2 (*Figure 14*). Flowstone 1a consists of at least three generations of flowstone growth, named, from oldest to youngest, phase A, B and C (*Figures 2b*, *13a,b* and *14*). We dated phases B and C via U-Th, at 502 ka (RS23; *Table 1*) and 478 ka (RS22; *Table 1*) respectively, but the uncertainties are large because the ages are close to the upper dating limit of the U-Th technique. The oldest, phase A layer at the base of Flowstone 1a records reversed magnetic polarity indicating that it formed before 780 ka (*Singer, 2014*). The three age estimates are consistent with the stratigraphic position of the three phases in Flowstone 1a, and they indicate that the erosion remnant of Unit 2 encrusted by Flowstone 1a is also older than 780 ka. The fact that the erosion remnants of Flowstone 1a dip into the chamber suggests that at the time of formation of all three phases of Flowstone 1a, the debris cone of Unit 2 sediment was still in place; that is, erosion of Unit 2 sediment from below Flowstone 1a would have occurred sometime after 585 ka (i.e., the older age limit for phase C in Flowstone 1a) and possibly as late as 437 ka (i.e., the younger age limit of phase C in Flowstone 1a). Unit 2 contains rare fossils of macrofauna, including a long bone in the erosion remnant, which must be older than 780 ka as well. We interpret the floor sediments of sub-unit 3a that contain the baboon tooth (sample 1841) to represent, at last in part, the reworked remains of the debris cone that once existed below Flowstone 1a (*Figure 8*) as material was removed from the chamber via floor drains, thereby undercutting the debris cone, which responded by slowly slumping into the chamber – a process ongoing today. The baboon tooth could be part of the original, Unit 2 debris cone and, therefore, older than 780 ka, although the US-ESR age indicates that it could also be younger (with a mid point age of ~ 679 ka between a maximum age of 723 ± 181 ka and the minimum age of 635 ± 148 ka, with a possible age range of 487 ka to 904 ka). If the tooth is younger than 780 ka, it would suggest that it was derived from different sedimentary deposits not tested in this study, or that it may have entered the chamber separately during erosion of the Unit 2 debris cone, and its presence may reflect more direct entry points from surface into the chamber now sealed by flowstone. This possibility was tested with sample RS9, a flowstone sample filling a thin (<14 cm wide) fracture in the dolomite on the surface that occurs above the Dinaledi Chamber. This flowstone yielded equilibrium U-Th results meaning that it formed before ~ 600 ka (*Table 1*), which is consistent with the interpretation that the Dinaledi Chamber was closed to direct entry of coarser-grained sediment from the surface prior to the entry of *H. naledi* into the cave system and remained closed until the present (*Dirks et al., 2016a*).

Below Flowstone 1a are five other flowstones (Flowstones 1b-e and Flowstone Group 2; *Figure 2b*), which each cover erosional remnants of sediments that we originally grouped as Unit 2 (*Dirks et al., 2015*). The geochronology results presented here (*Table 1*) now permit a better evaluation of the flowstone stratigraphy in the chamber, and it is evident that Unit 2 represents a significantly older stratigraphic unit that is restricted to deposits directly below Flowstone 1a, but not to the sediment deposits below Flowstones 1b-e. Flowstone 1c returns an age of ~243 ka (*Table 1*), suggesting that the sediments below Flowstones 1b-e are significantly younger than Unit 2 sediments below Flowstone 1a (*Figure 14*). It was noted before that the sediments below Flowstones 1b-e are less indurated and less weathered than the sediments below Flowstone 1a, and that they contain *H. naledi* material (*Dirks et al., 2015*). Their distinct appearance and fossil content is now

**Table 9.** Final mean palaeomagnetic data for all subsamples analysed from each phase of Flowstone 1a as shown in *Figure 13*. MAD = mean maximum angular deviation for individual samples; K = precision/sample dispersal parameter; Plat = palaeolatitude).

| Flowstone 1a | Declination (°) | Inclination (°) | MAD | K | Plat. | Polarity |
|---|---|---|---|---|---|---|
| Phase C | 15.5 | −39.7 | 3 | 70.8 | 75.4 | N |
| Phase B | 26 | −28.1 | 7.4 | 156.2 | 63.3 | N |
| Phase A | 156.4 | 15.9 | 5.7 | 30.2 | −60.0 | R |

confirmed with the dating. Therefore, we reinterpret the *H. naledi*-bearing sediments below Flowstones 1b-e as part of Unit 3 (sub-unit 3b), which means that all *H. naledi*-bearing sediments in the chamber are now part of sub-unit 3b (*Figure 8*).

Apart from the poorly consolidated erosional remnants below Flowstones 1b-e, hominin-bearing Unit 3 sediments also cover most of the floor of the Dinaledi Chamber. The layer of sub-unit 3b sediment has been interpreted as a relatively thin sheet (~20 cm) of rubbly mud clast breccia material mixed with *H. naledi* fossils based on outcrops in the excavation pit and preliminary ground penetrating radar results (*Naidoo, 2016*; *Figures 2* and *8*). Age brackets for sub-unit 3b were obtained by dating underlying outcrops of sub-units 1a and 1b via OSL, and by dating overlying flowstone units. In this context sample OSL5 is the most relevant for obtaining a maximum age estimate for sub-unit 3b, because it was taken from an outcrop of Unit 1 that is overlain by Unit 3. Here, a maximum age limit of 414 ka (the upper error limit of OSL5) can be assigned to sub-unit 3b if the OSL ages are taken at face value. Note however, that the maximum age limits for sub-unit 1b as determined from samples OSL3 and OSL4 are significantly less at 272 ka and 278 ka (upper error limits), respectively (*Table 8*), which supports the interpretation that sub-unit 1b formed due to erosion and redeposition on top of sub-unit 1a. The minimum age limit of sub-unit 3b can be more confidently constrained as it is overlain by flowstones with age ranges between 97 ka and 24 ka (*Table 1*). More importantly, a hanging remnant of sub-unit 3b with hominin material is covered by Flowstone 1c with a lower age limit of 236 ka obtained from the core of a stalactite overlying the rim of the flowstone remnant. This age provides the best minimum age estimate for sub-unit 3b, and by extension a minimum age for the *H. naledi* fossils.

U-Th dating of an erosional remnant of Flowstone 1 that occurs directly above an outcrop of sub-unit 1b from which sample OSL4 was taken, and appears to cover it, provides an age of 290 ± 6 ka, suggesting that Unit 1 in this location must be older than 284 ka. The OSL results (using the MAM model), however, suggest that the Unit 1 sediments in this location must be younger than 278 ka (*Table 8*; using MAM age models). This apparent paradox may indicate that the OSL ages are unreliable, but could mean that sub-unit 1b was deposited in an inverse stratigraphic order in relation to the flowstone. This is supported by physical evidence in the chamber in which a gap between the flowstone drape and underlying sediment widens towards the back of the outcrop (*Figure 2c*), implying that sub-unit 1b was deposited below an erosional remnant of Flowstone 1, and is, therefore, younger than this flowstone. Recrystallization-dissolution textures, and anomalously high $^{234}U/^{238}U$ ratios (>3) suggest that the age reported for this flowstone sample (RS5) should be treated as a minimum age.

Although the age constraints for sub-units 1a and 1b are imprecise they do suggest that Unit 1 in this part of the chamber is younger than Unit 2, and that the red mud clasts forming Unit 2 sediment were derived from source material matching our description of sub-unit 1a, but positioned higher up in the cave. This source material was possibly part of sub-unit 1c, or it could represent part of an older and as yet undefined sub-unit of Unit 1. No age assessment for sub-unit 1c deposits were done, because accumulations are too small to be tested with OSL or too difficult to access.

The U-Th ages from flowstones and teeth place constraints on the changing physical environment experienced in the cave chamber over time. Flowstone deposition in the Dinaledi Chamber occurred during discrete periods including 24–32 ka, 50 ka, 88–105 ka, and during older events around 242 ka, 290 ka, between ~437 ka and 683 ka, and >780 ka. The flowstones are associated with relatively low initial $^{234}U/^{238}U$ ratios of 1.8–2.4 (*Tables 2* and *3*). Flowstones formed in different parts of the chamber as drip points shifted, but no clear pattern in age distribution is apparent (*Figure 1b*) other

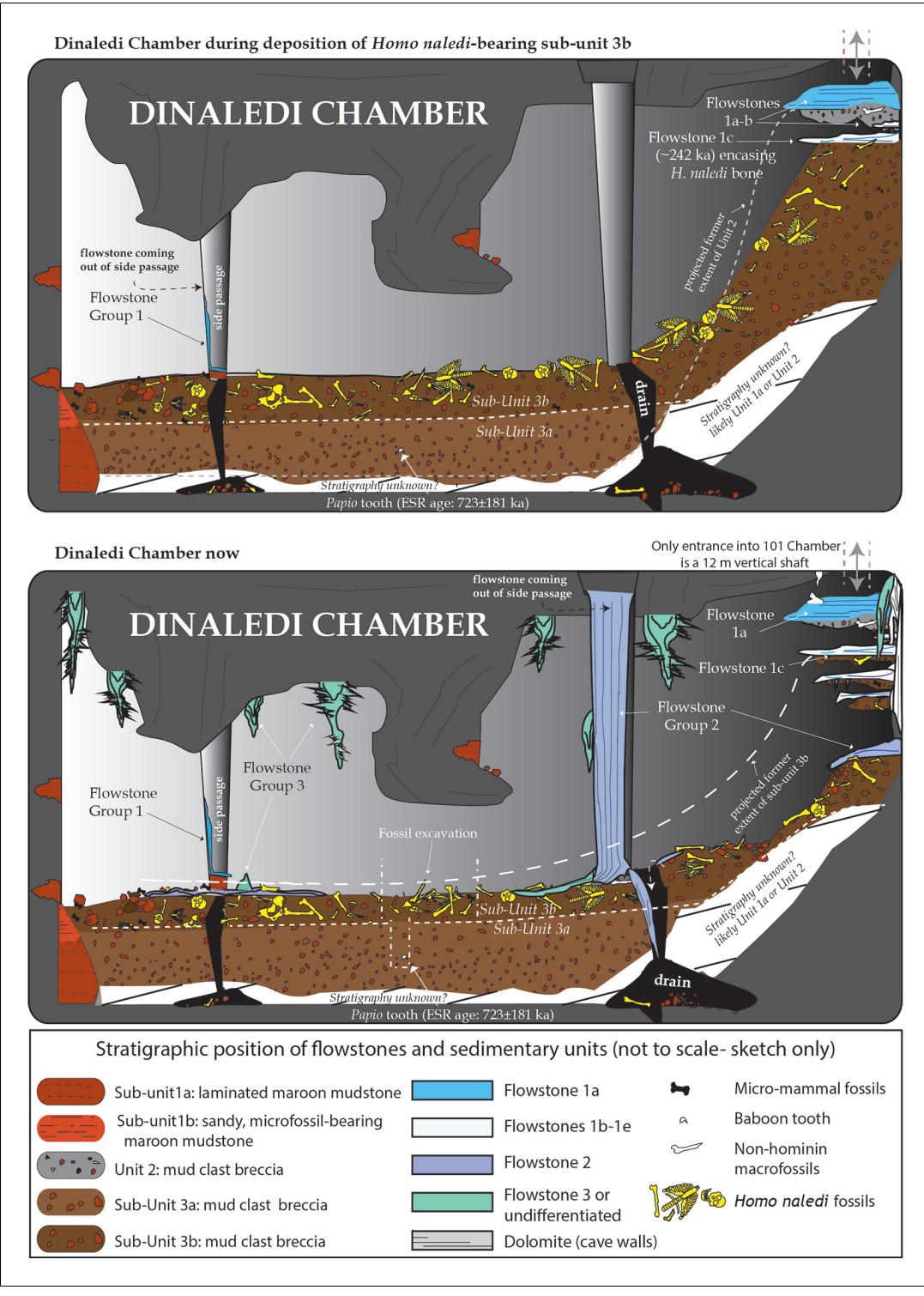

**Figure 8.** Cartoon illustrating the sedimentary history resulting in the deposition and redistribution of sediment of Units 2 and 3, and Flowstone Groups 1 to 3 in the Dinaledi Chamber. Note that all hominin fossils are contained in sub-unit 3b, but that this sub-unit has been repeatedly reworked after its initial deposition. Fossil entry occurred during the initial stages of deposition of Unit 3 below the entry shaft and predated deposition of Flowstone 1c. *H. naledi* fossils may have continued to enter the Dinaledi Chamber as older parts of Unit 3 were eroded from below Flowstone 1c, and as remnants of all older units were reworked to be incorporated into Unit 3 sediments that accumulated along the floor of the Dinaledi Chamber.

than that the flowstones deeper in the cave generally appear to be younger. The U-Th ages for the teeth also define a number of apparent U uptake events at around 43–48 ka, 70–75 ka, 80–90 ka, 100–120 ka and ~200 ka (*Tables 4* and *5*). The initial $^{234}$U/$^{238}$U ratios associated with each of these events is similar and anomalously high (6.9–8.1).

The periods of time during which flowstones formed in the cave, by and large, do not overlap with the periods of time during which U-uptake appears to have occurred in the teeth, although both types of events were probably associated with wet periods in the chamber. The systematic difference in the initial $^{234}$U/$^{238}$U ratios obtained from flowstones, as compared to teeth, indicates that an isotopically distinct water source led to U-uptake in the teeth, such as groundwater in the dolomitic aquifer of the Malmani Group with reported anomalously high initial $^{234}$U/$^{238}$U ratios (e.g., *Kronfeld et al., 1994*). Working on the assumption that the high initial $^{234}$U/$^{238}$U ratios are derived from the groundwater reservoir, the observed age groupings suggest that U uptake events in teeth represent (partial) inundation events of the Dinaledi Chamber, whilst the flowstone formation events reflect periods during which the groundwater table had dropped below floor level, but extensive drip still occurred within the Dinaledi Chamber caused by water derived from the surface with generally lower $^{234}$U excess. The age groupings also indicate that episodic wet periods in the Dinaledi Chamber alternated with periods during which no flowstone was deposited.

## Age estimates for *H. naledi* and implications for hominin evolution

*Figures 1b* and *14* summarize the results of all dating methods applied to the Dinaledi Chamber during the course of this study. It is clear from these results that the *H. naledi* assemblage in Unit 3 is of mid- to late-Middle Pleistocene age. The best age estimates for the *H. naledi* fossils come from the averaged US-ESR ages for samples 1788 and 1810: 229 +60/–46 ka (maximum) and 179 +49/–40 ka (minimum) for sample 1788, and 276 +59/–77 ka (maximum) and 220 +50/–60 ka (minimum) for sample 1810, with an age range of 139 ka to 335 ka. The maximum age scenario provides an average age for both teeth of 253 +82/–70 ka, and the minimum age scenario provides an average age of 200 +70/–61 ka. Considering the observed Rn loss in the cave sediments, the maximum age estimate is considered to be closer to the true age.

The lower age limit of 139 ka must be disregarded and shifted to ~200 ka considering a U-Th minimum age estimate for parts of enamel on tooth sample 1810 (*Table 5*). This minimum age limit can be constrained further to 236 ka based on U-Th age estimates for Flowstone 1c (242.0 ± 5.0 ka; 242.9 ± 6.6 ka; *Table 1*), which directly covers fossil material of *H. naledi,* noting that these ages may represent minimum age estimates for the flowstone as a result of possible U uptake during a period of raised groundwater levels (*Figure 14*).

The maximum age limit of 335 ka for the *H. naledi* fossils relies on 80% Rn loss throughout the burial history of the fossils and is probably an over-estimate. A separate maximum age estimate for sub-unit 3b can be obtained from the baboon tooth (sample 1841) in sub-unit 3a sediments devoid of hominin fossils, underlying the partly articulated remains of *H. naledi*, with US-ESR age estimates varying from 635 ± 148 ka (minimum) to 723 ± 181 ka (maximum) with an age range of 487 ka to 904 ka and a mid-point age of 679 ka. The older age estimate is considered to be more likely (considering the measured Rn loss). These maximum age estimates are consistent with the direct age estimates for the *H. naledi* fossils, but do not further constrain the upper age limit (*Figure 14*).

A further maximum age estimate for the *H. naledi* fossils can be obtained from age estimates for sub-unit 1a using sample OSL5. This sample provides a maximum age limit for sub-unit 3b, of ~414 ka (*Table 8*; using MAM). Although OSL ages are poorly constrained, and do not provide very precise age limits, this estimate is broadly consistent with the estimated age for reworking of Unit 2 sediments from below Flowstone 1a (437–585 ka) and the US-ESR age estimates for the teeth.

Considering all age results presented here the most parsimonious age estimate for the *H. naledi* fossils is sometime between 236 ka and 335 ka. More work will be needed in future to constrain these ages further (*Figure 14*).

Until now, it has been generally assumed that morphologically primitive hominins like *H. naledi* (*Berger et al., 2015*) did not survive into the later parts of the Pleistocene in Africa. This general assumption has commonly guided the interpretation of fossil discoveries with poor geological or stratigraphic context, including initial estimates for the age of the *H. naledi* fossils (*Thackeray, 2016*; *Dembo et al., 2016*). The new age estimates for *H. naledi* show that an approximate age for the hominin fossil fragments cannot be simply deduced from their morphology (*Thackeray, 2016*;

*Dembo et al., 2016*). Detailed geological investigations are critical before any attempt to ascribe an age to the fossils is made, and even then great care must be taken in interpreting results, which may not always be conclusive (e.g. see famous examples such as Sterkfontein or Liang Bua; *Partridge et al., 2003*; *Wilkinson, 1985*; *Walker et al., 2006*; *Roberts et al., 2009*; *Pickering and Kramers, 2010*; *Granger et al. 2015*; *Sutikna et al., 2016*; *Kramers and Dirks, 2017*).

It is generally assumed that all African fossil hominins producing Middle Stone Age archaeological industries in the past 300 ka were part of a single variable species of early *H. sapiens* or an immediate precursor (e.g., *Mcbrearty and Brooks, 2000*; *Lahr and Foley, 2001*; *Stringer, 2002*). The new ages now show that *H. naledi* existed at the same time as the first Middle Stone Age tools were produced in southern and eastern Africa, whilst skeletal evidence shows that *H. naledi* was probably capable of tool use (*Berger et al., 2015*; *Hawks et al., 2017*). This raises the possibility of *H. naledi* being responsible for some of the MSA traditions. The implications of the new ages for *H. naledi* are discussed in detail in *Berger et al., 2017*.

## Material and methods

### Flowstone samples for U-Th dating

A total of seventeen flowstone samples (RS1, RS5-6, RS8, RS10-11, and RS 13–23) from the Dinaledi Chamber were dated via U-Th geochronology, including one set of blind duplicates (RS1 and RS15; *Table 1*, *Figures 1b* and *3*). In addition, one flowstone sample (RS9) was taken on the surface (WGS84 571240–7121866) from a shallow pit about 11 m SW of the projected surface position of the excavation pit in the Dinaledi Chamber (*Figure 3d*). For each sample a powder was prepared and then split, with one half being dated at JCU and the other half being dated at UoM; that is, for each sample both JCU and UoM dated the same material.

In the Dinaledi Chamber samples of Flowstone Groups 1, 2 and 3 were collected from a variety of stratigraphic positions (*Figures 1b*, *2* and *3*). All the sampled flowstones formed as sheets, crusts, or drapes overlying older sediment units, with the exception of samples RS13 and RS18, which were taken from a small stalactite that formed along the lip of an erosional remnant of Flowstone 1c. Additionally, samples RS14, RS16 and RS17 were taken from a flowstone drape along a dolostone side-wall of the chamber. In all instances the flowstones have a free upper surface, that is, they are not overlain or covered by sediment, and all flowstones are interpreted to be younger than the sediment units they cover. The one exception to this rule could be RS5, which comes from a partly resorbed erosion remnant of Flowstone 1 that appears to overlie Unit 1, but is separated from the top surface of Unit 1 by a small opening that widens with depth (*Figure 2c*). This leaves the possibility that this flowstone is an erosional remnant, and that Unit 1 (sub-unit 1a) sediments built up below it after it had been deposited. The location of each sample within the Dinaledi Chamber is shown in *Figure 1b*, and outcrop and close-up photos of the sampled flowstones are shown in *Figure 3*.

**RS1 and RS15** are blind duplicate samples taken from a hanging remnant of Flowstone Group 2 that occurs in a N-trending fracture, ~10 m N of the excavation pit and ~3 m N of a major outcrop of sub-unit 1a sediment from which sample OSL3 was collected. The flowstone overlies largely unconsolidated floor sediments of Unit 3, which in this locality have eroded from underneath the flowstone to leave a hanging remnant ~8 cm above the current floor level (*Figure 3i*). The flowstone consists of a 15–18 mm-thick layer of calcite overlying an irregular surface of mud clast breccia, locally incorporating and growing around large mud clasts that were lying on the palaeo-surface. The flowstone is grey-white in colour and preserves 3–6 mm scale laminations visible due to subtle colour variations. The flowstone is recrystallized with elongated, acicular crystals of calcite growing from the base to the top of the layer across all internal laminations. The upper surface of the flowstone has a rough, pitted appearance as a result of partial resorption or dissolution of calcite along the grain boundaries of the needle-like crystals. Samples RS1 and RS15 were taken from a 3 mm-thick zone, 3 mm above the basal contact of the flowstone layer (*Figure 3i*).

**RS5** was sampled from a thin sheet of Flowstone Group 1 in a WNW-trending fracture, ~4 m W of the excavation pit (*Figure 1b*). The flowstone overlies orange sandy mudstone belonging to sub-unit 1b from which sample OSL4 was taken (*Figure 3a*), but appears to be separated from this unit by a narrow opening that widens to the back of the outcrop (*Figure 2c*). The flowstone consists of an 8–22 mm thick, cream white layer of carbonate with a sponge-like, porous, sugary texture that

largely masks (due to recrystallization of the primary calcite) underlying mm-scale laminations. The flowstone layer appears to be partly dissolved along its stratigraphic top with the external surface of the layer truncating internal laminations. Sample RS5 was taken from a 5 mm-thick horizon, 3 mm below the top of the flowstone layer (*Figure 3a*).

**RS6** was sampled from a thin sheet of Flowstone Group 2 in a WNW-trending fracture, ~6 m W of the entry zone (*Figure 1b*). The flowstone overlies erosion remnants of Unit 1 and Unit 3. At the sample site, sediments of Unit 3 have been partly eroded from underneath the flowstone leaving a hanging remnant, 5–10 cm above the current floor level (*Figure 3b*). The flowstone consists of an 8–12 mm thick crust overlying an irregular sediment surface, incorporating fine (<3 mm) mud clasts in the base of the layer. It is white-grey in colour and semi-transparent, with limited evidence of internal layering except for a slightly lighter coloured basal layer that is several mm thick. The entire layer is recrystallized with fine, radiating acicular crystals of calcite growing upward from the basal contact. The upper surface has a rough pitted appearance as a result of partial resorption/dissolution along grain boundaries of the acicular crystals. Sample RS6 was taken from a 3 mm-thick horizon at the bottom of the flowstone (*Figure 3b*).

**RS8** was sampled from a thin sheet of Flowstone 2 on the floor of the entry zone below the stack of hanging remnants of Flowstone 1a-e. The flowstone overlies sediments of Unit 3 that are partly eroded from underneath the flowstone leaving a 5–10 cm gap between the flowstone sheet and the current floor (*Figure 3c*). The flowstone consists of a 5–22 mm thick layer overlying an irregular surface of mud clast breccia. The grey-white flowstone preserves 3–5 mm thick layering visible as subtle colour variations, despite partial recrystallization and replacement by elongated acicular crystals of calcite. The basal 10 mm is composed of a mesh of fine aragonite needles. Above this zone acicular calcite replaces the aragonite. The upper surface of the flowstone has a rough, pitted appearance as a result of partial resorption/dissolution along grain boundaries of the acicular crystals. Sample RS8 was taken from the bottom 3–5 mm of the flowstone layer (*Figure 3c*).

**RS9** was sampled from a surface outcrop in a shallow mine pit that occurs about 11 m SW of the projected surface position of the excavation pit in the Dinaledi Chamber (*Figure 3d*). The sample site occurs along strike of the same fracture along which the Dinaledi Chamber was formed at depth, and it is therefore possible that this surface outcrop in the past could have linked to the cave system below. The flowstone consists of a 12–25 mm thick layer of carbonate overlying an irregular surface of consolidated mud clast breccia similar to sediments of Unit 2 in the Dinaledi Chamber. The grey-white flowstone preserves 1–5 mm scale laminations visible due to colour variations and the presence of several thin brown marker surfaces. The laminations are locally recrystallized and overgrown by radiating, elongated crystals of calcite growing from the base to the top of the flowstone layer. Additionally, fibrous aragonite needles are widely distributed in sheaf-like patterns. The upper surface of the flowstone has a rough, pitted appearance as a result of partial resorption/dissolution along the grain boundaries of the acicular crystals. Sample RS9 was taken from a 4 mm-thick white horizon, 2 mm above the basal contact of the flowstone layer (*Figure 3d*).

**RS10** was sampled from a thin crust of Flowstone Group 2, about 2 m W of the excavation pit and 0.2 m W of the location where tooth sample 1810 was found (*Figure 3e*). The flowstone directly overlies mud clast breccia of Unit 3 and hominin bone fragments. It consists of a 4–8 mm-thick laminated flowstone crust overlying an irregular sediment surface and incorporates fine (<3 mm) mudstone clasts within its basal laminae. This flowstone is white to grey in colour and is finely laminated and partly recrystallized, with recrystallization visible as white, fine, radiating needles of aragonite growing upward from the basal contact, along an irregular alteration front into grey-white, laminated calcite near the top of the layer. RS10 was taken from the laminated bottom 3 mm of the flowstone layer (*Figure 3e*).

**RS11** and **RS21** represent two samples taken from the top and bottom, respectively, of the same flowstone unit developed along the floor below an active drip point, 1.5 m E of the excavation pit. This flowstone consists of a basal layer of grey flowstone interpreted as Flowstone Group 2 overlying Unit 3 sediments, covered by a white flowstone layer containing small stalagmites interpreted as Flowstone Group 3 (*Figure 3f*). The sample collected for dating consists of a 20–28 mm thick, finely laminated flowstone including the two distinct layers described above (*Figure 3f*). The basal layer (RS21; *Figure 3f*) is 3–12 mm thick, and consists of brown-grey calcite resting on Unit 3 sediment. This layer is strongly recrystallized with radial, acicular crystals of calcite growing upward from the lower contact, which locally appears to have replaced an earlier generation of acicular aragonite.

These crystals overgrow mm-scale laminations defined by white to brown colour variations. RS21 was sampled from the basal-to-central segment of this layer. The basal layer is overlain by an 8–16 mm thick layer of finely laminated (sub-mm scale) white speleothem. The basal 2–3 mm of this layer is recrystallized with fine aragonite needles radiating out from the base. Above that the flowstone consists of fine lamellae defined by subtle grey to white colour variations. Each lamina is composed of botryoidal aggregates of acicular aragonite. RS11 was taken from the top 3 mm of this white flowstone (*Figure 3f*).

RS13 and RS18 are two samples taken from the same stalactite developed along the lip of an erosional remnant of Flowstone 1c near the entrance into the Dinaledi Chamber (*Figure 1b*). In this location, Flowstone 1c covers the erosional remains of a mud clast breccia containing a long bone consistent with the *H. naledi* assemblage (*Figure 3g*), and interpreted as Unit 3. The stalactite connects to part of the speleothem layers that cover the bone.

The stalactite preserves well-developed internal layering, with layers asymmetrically developed around a core, in which layers thicken towards the outward facing side of the stalactite. From core to rim the stalactite reaches a maximum thickness of 53 mm and includes three separate zones that can be distinguished based on internal texture, layering, and colour (*Figure 3g*). The innermost zone forms an 8 mm-thick, finely laminated (sub-mm scale) core centred on a small mud clast and is terminated by a thin brown, mud-rich rim. This core is surrounded by a ~23 mm-thick central zone that consists of more coarsely layered (3–10 mm-thick), cream to grey-white coloured calcite with layering preserved as subtle colour variations, which at its base shows replacement by radiating sheaves of aragonite needles. This zone is mantled by a ~22 mm-thick outer zone of white calcite preserving only remnants of internal layering. This outermost zone is characterized by extensive replacement of abundant, older aragonite needles by coarse-grained calcite, creating a patchy texture. The sample has been affected by recrystallization resulting in the formation of radiating acicular crystals of calcite that grow from the core outward. The white outer layer also shows evidence of further recrystallization, with the formation of coarse (2–4 mm), equant calcite grains, many with highly irregular grain boundaries that overgrow the acicular grains. The outer surface of the stalactite has a rough, pitted appearance as a result of partial resorption/dissolution along the surface with dissolution along the grain boundaries of the acicular crystals. RS18 was sampled from a 3 mm-thick horizon near the base of the central zone. RS13 was sampled from the outermost part of the outer zone (*Figure 3g*).

RS14 was sampled from an irregularly shaped, cascade-like crust of Flowstone Group 2, along the side-wall of the main floor drain, between the entry shaft and excavation pit (*Figures 1b* and *3h*). The sample of flowstone crust consists of an 8–11 mm thick layer of cream coloured carbonate displaying mm-scale laminations and a sugary, recrystallized texture with numerous, fine pore spaces along laminar surfaces. Sample RS14 was taken from a 3 mm-thick zone encompassing several fine laminations, ~5 mm above the base of the flowstone layer (*Figure 3h*).

RS16 and RS17 are two separate samples from a cascade-like crust of Flowstone Group 2 that formed within the main floor drain between the entry shaft and excavation pit (*Figure 1b*). This flowstone developed on top of the dolomite back-wall of the drain (*Figure 3h*). The sample consists of a mostly cream coloured flowstone that is up to 73 mm thick, with a massive sugary, recrystallized texture, preserving mm-scale laminations visible due to subtle colour variations. Many of the equant calcite grains contain remnant aragonite needles, reflecting an earlier phase of aragonite growth. Laminations along the basal 3–4 mm of this flowstone are brown in colour, whereas the top 10–12 mm of this flowstone layer consists of what is possibly a separate, younger unit of laminated grey-brown carbonate with small pore spaces developed along some of the laminar surfaces. Sample RS16 was taken from a 3–4 mm-thick layer directly above the basal zone with brown laminations. Sample RS17 was taken from a 4 mm-thick zone at the top of the cream-coloured laminated flowstone, immediately below the darker coloured top unit (*Figure 3h*).

RS19 was taken from a wedge-shaped sample of Flowstone Group 2 that formed along the lip of a drip pool directly overlying Unit 3 sediments, 1.2 m S of the excavation pit (*Figure 1b*). The flowstone is cream-coloured and up to 30 mm thick, and overlies a flat sediment surface. It preserves complex internal layering and a variety of textures (*Figure 3j*). A basal 2–4 mm-thick layer characterised by fine calcite needles growing at right angles to layering is overlain by an irregular mass of darker cream coloured, unstructured calcite with a sugary texture and numerous fine voids. Towards its top this unstructured mass is interlayered with and covered by several 1–4 mm-thick lamellae characterised by fine calcite needles growing at right angles to layering similar to the basal layer,

but containing pore spaces. These in turn are overlain by an 8 mm-thick zone of finely laminated flowstone with a fine-grained (<0.5 mm) sugary texture and abundant fine pore spaces. RS19 was taken from the basal flowstone layer with radiating calcite crystals (*Figure 3j*).

RS20 was sampled from a thin sheet of Flowstone Group 2 in a WNW-trending fracture, ~2 m W of the excavation pit (*Figure 1b*). The flowstone forms a cascade-like drape that directly overlies orange mudstone of sub-unit 1a from which sample OSL5 was taken (*Figure 3k*). RS20 occurs directly above sample RS10, which was taken at the foot of the cascade (*Figure 1b*). The sampled flowstone layer is irregular in shape as it covers sediments with topography. It consists of a finely laminated 15–22 mm thick crust overlying sub-unit 1a sediment and includes a ~6 mm thick, laminated basal unit of white flowstone that incorporates small (3–10 mm) mud clasts, overlain by more coarsely laminated (mm-scale) grey brown carbonate free of inclusions. The flowstone is internally locally recrystallized with acicular aragonite crystals growing within ~3 mm thick zones at right angles to layering. RS20 was sampled from the basal layer of the sample (*Figure 3k*).

RS22 and RS23 are two samples from an eroded rim of Flowstone 1a near the entry shaft into the Dinaledi Chamber (*Figure 1b*). This flowstone overlies erosional remnants of well-indurated mud clast breccia assigned to Unit 2 (*Figure 3l*), and mostly consists of coarsely recrystallized white carbonate. The flowstone sample from which RS22 and RS23 were taken occurs down dip from the palaeo-magnetic sample taken from Flowstone 1a as described below (*Figures 2b* and *13*). The flowstone layer is ~25 mm thick and comprises a 6 mm thick, basal unit of grey to brown calcite that is recrystallized into fine (sub-mm scale), equigranular grains of calcite overgrowing an older acicular texture. The basal layer corresponds to phase B carbonate described for the palaeo-magnetic sample, and is overlain by a clean white calcite unit (i.e., phase C in the palaeo-magnetic sample) that is coarsely recrystallized with 2–4 mm equidimensional calcite grains overgrowing (and partly destroying) an older texture formed by acicular grains. Small voids occur in the interstitial spaces between the coarse calcite grains. RS23 was taken from the grey basal unit (i.e., phase B) of the flowstone layer and RS22 was sampled from the central part of the upper white recrystallized unit (i.e., phase C, *Figures 3l* and *13b*). In this sample, flowstone belonging to the older phase A carbonate is no longer present, as this layer pinches out along dip between the point where the palaeomagnetic sample was taken and the point where the U-Pb sample was taken as can be seen in *Figures 2b* and *13*.

## Tooth samples for ESR dating

Three *H. naledi* teeth (samples 1810, 1767 and 1788) and one baboon tooth (sample 1841) were collected from Unit 3, in the Dinaledi Chamber (*Figures 1b*, *4*, *6* and *7*) for ESR dating and U-Th analysis. Sample 1767 (full catalogue number U.W.101–1767) is an extremely worn upper premolar crown (*Figure 6a*) obtained from approximately 1 m SW of the excavation pit (*Figure 4*), and occurred on surface surrounded by mud clast fragments of Unit 3. This tooth is deeply weathered, and preserved only a small rim of enamel on the buccal margin with a maximum height of 4.5 mm. Otherwise the crown is a concave dentine surface worn to the cervix, with bright white, highly bleached dentine. The tooth is brittle and appears strongly affected by water action. Sample 1788 (full catalogue number U.W.101–1788) is a lower right second molar (*Figure 6b*) obtained from approximately 2 m WSW of the excavation pit (*Figure 4*), embedded within loosely packed, mud clast breccia of Unit 3, ~ 2 cm below the ground surface level. This tooth is partly broken at the root, but has an otherwise well preserved crown with thick, light-grey enamel. The distal root of the tooth is present and complete, but the lingual root is broken off just below the cervix. The tooth is morphologically consistent as an antimere of U.W.101–284. The dentine is highly bleached and brittle and appears affected by water action. Sample 1810 (full catalogue number U.W.101–1810) is a lower left third premolar or possibly lower left fourth premolar (*Figure 6c*) obtained from the SE corner of the excavation pit (*Figure 4*), embedded within sediments of Unit 3, ~5 cm below the original ground surface level. This tooth is well-preserved with thick, light-blue-grey enamel, and shows little evidence of bleaching or weathering, with only a very slight polishing wear on the distal crest of the protoconid. The morphology of the crown is similar to other lower third premolars in the collection, however, it is slightly shorter than many of those. Sample 1841 (full catalogue number U.W.101–1841) is a well-preserved tooth crown, morphologically consistent with a lower left second molar of *Papio sp*. This is a complete enamel crown of an unerupted tooth with no wear facets or interproximal facets in evidence, and the roots had not formed (*Figure 7*). The specimen was recovered from a sediment

sample taken at a depth of 55–60 cm below the original ground surface of the cave floor near the base of the sondage dug in the centre of the excavation pit (*Figures 1b*, *2d* and *4*). This tooth occurs in sub-unit 3a, ~40 cm below the stratigraphically lowest occurrence of partly articulated remains of *H. naledi*, and represents the only non-hominin macrofossil recovered from Unit 3.

## Sediment samples for OSL dating

Three OSL samples (OSL3, OSL4 and OSL5) were collected in the Dinaledi Chamber from erosion remnants of sub-unit 1a, in which sandy, laminated mudstones are exposed (*Figures 1b* and *5*). These sandy intercalations were targeted because they contain fine-grained quartz and feldspar grains that can be extracted for analysis. For each sample a 30 cm length of aluminium piping with a diameter of 5 cm was hammered into the sediments in a horizontal direction or parallel to sedimentary laminations visible within the units. A core sample within the pipe was extracted for OSL analyses together with a sediment sample from the same unit to determine background radiation from measured values of U, Th and K.

Sample OSL3 comes from an erosion remnant of sediments of sub-unit 1b collected near the intersection point of two fractures trending N and E respectively, ~6 m N of the excavation pit (*Figure 1b*). Samples OSL4 and OSL5 come from an erosion remnant of Unit 1 sediment along an ENE-trending fracture, ~3 m W of the excavation pit. Sample OSL4 is obtained from sub-unit 1b directly below a thin, partly resorbed flowstone sheet attributed to Flowstone Group 1 (*Figure 3a*). Sample OSL5 occurs as an erosional remnant of sub-unit 1a, 1 m E, and stratigraphically 10–20 cm below sample OSL4. The Unit 1 sediments in this location are partly covered by Unit 3 sediments and a cascade of Flowstone Group 2 (*Figure 3k*).

## Flowstone sample for palaeomagnetic analysis

The speleothem sampled for palaeomagnetic analysis comes from Flowstone 1a near the entry zone into the Dinaledi Chamber (*Figure 2b*). The sample is layered and comprises three distinct phases (from base to top: A-C) separated by thin clastic horizons that mark disconformities (*Figure 13a,b*).

The lower phase (phase A) is interstratified with visible clastic laminations. The internal laminations are truncated along the lower surface of the sample indicating that phase A flowstone was partly dissolved during a phreatic event after the flowstone had been deposited. The middle phase (phase B) consists of intercalated flowstone speleothem and detrital sediment layers, with the detrital layering concentrated towards the downslope part of the sample. Detrital material is generally less than in phase A, indicating decreasing amounts of clastic contamination. Like phase A, the internal layering of phase B is truncated along the lower surface of the sample, indicating that the phreatic dissolution event occurred after deposition of phase B. There is no apparent truncation surface between phase A and B flowstone, indicating that phase B formed on top of phase A after a period of non-deposition during which detrital material accumulated on top of phase A. The upper surface of phase B truncates internal layering reflecting dissolution during a phreatic event. This was followed by a period of non-deposition of flowstone during which a thin layer of detrital clastic material accumulated, before deposition of the upper layer (phase C) of flowstone occurred. Phase C comprises younger, cleaner flowstone which displays extensive recrystallization along the base of the unit with the formation of elongated calcite crystals. This zone of recrystallization could potentially also indicate a younger infill of a cavity that had formed between phase B and C flowstone, and has been avoided during sampling. Phase C flowstone contains little to no detrital inclusions, and suggests that sediment influx into the chamber was not occurring during its deposition.

Phase B and phase C flowstone correlate with samples RS23 and RS22 respectively, that were collected for U-Th dating. The top of the sample was oriented in the cave to magnetic N (−18.2° degrees from true N at this location with a −62.9° inclination). The inclination was accounted for by marking the sample on a completely flat surface of the block.

## Bone samples for radiocarbon dating

Three weathered bone fragments of *H. naledi* were collected for radiocarbon dating including: (i) a tibia shaft fragment, 53 mm in length (U.W. 101–567); (ii) a femur shaft fragment comprising the whole circumference of the shaft, 79 mm in length (U.W. 101–857), and; (iii) a metatarsal or

metacarpal shaft with no articular morphology, 48 mm in length (U.W. 101–065). All samples were collected from the surface of Unit 3 as isolated fragments near the excavation pit.

## Analytical and computational dating methods

### U-Th dating

#### Background to U-Th dating

Cave carbonates are proven archives of sediment chronology and terrestrial climate variation (e.g., *Pickering and Kramers, 2010*; *Pickering et al., 2010*). Cave carbonates less than ~ 600 ka in age can be precisely dated using the U-Th disequilibrium dating method (e.g., *Hellstrom and Pickering, 2015*). The disequilibrium technique differs from other radiogenic techniques like U-Pb or Rb-Sr in which the daughter product is stable and accumulates indefinitely. In the $^{238}$U decay chain, $^{234}$U (half-life of 245.5 ka) decays to $^{230}$Th (half-life of 75.4 ka), which itself decays to $^{226}$Ra. This means that rather than accumulating indefinitely with time, the concentration of $^{230}$Th in a sample gradually moves to a point where the number of decays of $^{230}$Th atoms equals the number of $^{230}$Th atoms produced via the decay of $^{234}$U ('secular equilibrium'). Because Th is nearly insoluble in surface waters, flowstone does not usually contain initial $^{230}$Th, in contrast to U, which is soluble in oxidizing waters and is incorporated into flowstone at low concentrations (generally <5 ppm; e.g., *Pickering et al., 2010*). In U-Th dating, the accumulation of $^{230}$Th towards secular equilibrium and the decay of its immediate parent ($^{234}$U) yields the age. In calculating a U-Th age, the ratio of $^{234}$U to its parent $^{238}$U must also be determined, as this ratio is generally elevated above secular equilibrium in natural waters. The assumption that initial $^{230}$Th concentrations in flowstone are close to zero may be violated by varying amounts of 'detrital matter', carrying initial $^{230}$Th. However, many cave carbonates form with essentially no initial Th. Where corrections for initial Th are small, potential variability in the isotopic composition of this Th is generally incorporated in the age error. The U-Th dating technique has an upper age limit of ~600 ka, determined by the half-life of $^{230}$Th, and by the precision with which the various isotopes can be measured in the laboratory (*Cheng et al., 2013*).

After initial tests for U-Th disequilibrium dating at JCU proved successful, a suite of flowstone samples was collected from the Dinaledi Chamber and prepared as duplicate samples for double blind dating analyses at James Cook University (JCU) and at the University of Melbourne (UoM).

#### Methodology: James Cook University (JCU)

**Sample preparation** – For each sample a 100–200 mg flowstone fragment was removed with a hand-held Dremel tool. The purest flowstone layers were targeted for sampling, and dark layers containing detrital material were avoided. Samples were taken parallel to layering, typically across a 3 mm thick zone and any attached sediment or impurities were removed.

Prior to it being dissolved, each flowstone sample was cleaned. Samples were first loaded into centrifuge tubes filled with ethanol and agitated in an ultrasonic bath for 10 min. This procedure was performed twice for each sample, with clean ethanol for each bath. The centrifuge tubes containing the sampled fragments were then filled with ultrapure water (18.2 MΩ) and agitated in an ultrasonic bath for approximately 2 min. This procedure was performed three times for each sample, changing the ultrapure water each time, after which samples were dried under a fume hood.

**Dating protocol** – Chemical procedures largely follow *Horwitz et al. (1992)* and *(1993)*. Dried samples were transferred to Teflon containers and spiked with a mixed $^{229}$Th-$^{233}$U-$^{236}$U tracer prior to dissolution in 3M HNO$_3$. Chemical separation of U from Th was performed using either a chromographic extraction method using Eichrom UTEVA resin (*Horwitz et al., 1992*, *1993*) or Eichrom TRU resin (*Lou et al., 1997*). The UTEVA procedure produced higher U yields and better separation of U from Th, and has since been adopted as the standard procedure at JCU. Analytical blanks from both procedures were indistinguishable and low (<10 picograms U). Prior to analysis samples were treated with 32% H$_2$O$_2$ to destroy any particles of resin.

Uranium and Th isotope measurements were performed with a ThermoScientific Neptune MC-ICP-MS instrument at the Advanced Analytical Centre of JCU. The instrument is equipped with a single central secondary electron multiplier (SEM) and 9 Faraday detectors and amplifiers with $10^{11}$ Ω resistors. Measurements were performed using a Cetac Aridus combined with a 100 µl PFA nebulizer for sample introduction. An exponential mass fractionation law was used to correct isotopic ratios. For U isotopes we normalized ratios to a $^{235}$U/$^{238}$U ratio of 137.794 (*Goldmann et al., 2015*). For Th

isotopes, samples were bracketed with standard solutions of CRM-145 and corrected using the mass bias determined for the U standard. For isotopes measured on the SEM, we corrected for tails using the log mean of the relevant half masses. For isotopes measured on the Faradays detectors, the shape of the tail and tailing parameters were defined daily using a >25 V of a solution of U005A; with these parameters a correction for the family of U tails was made.

We represent age uncertainties that include propagated 2σ envelopes on isotope ratios and decay constants, as well as an error envelope on the assumed initial $^{230}$Th/$^{232}$Th. The assumed value for initial $^{230}$Th/$^{232}$Th is 0.83 ± 0.50 (2σ). These are the main sources of uncertainty for the age estimates, and we treat them as three sources of uncorrelated uncertainty. Uncertainty on measured isotope ratios reflects counting errors and variability within a mass spectrometric analysis. Decay constant uncertainty envelopes are after *Cheng et al. (2013)*, and we propagate uncertainty on tracer (spike) isotope concentrations. Uncertainties are represented as symmetric in both the older and younger direction and are expressed in ka.

## Methodology: The University of Melbourne (UoM)

**Sample preparation –** Samples were prepared at JCU as described above, and sent to UoM for analysis.

**Dating protocol –** Analyses at UoM were done by multi-collector inductively coupled plasma mass spectrometry (MC-ICP-MS), using the analytical methods of *Hellstrom (2003)*. An assumed value for the initial $^{230}$Th/$^{232}$Th activity ratio of 1.5 ± 1.5 (2σ) was used to calculate corrected ages in conjunction with the decay constants of *Cheng et al. (2013)*. All ages are expressed in ka and corrected for initial $^{230}$Th using *Equation 1* of *Hellstrom (2006)*. $^{234}$U/$^{238}$U activity ratios were determined after *Hellstrom (2003)* and *Drysdale et al. (2012)*. All uncertainties are presented as 2σ.

## ESR dating

### Background to ESR dating

Electron spin resonance (ESR) dating in combination with U-series data offer a largely non-destructive approach for direct dating of human fossil remains (e.g., *Grün, 1989*, *1997*, *2009*; *Joannes-Boyau et al., 2010*; *Joannes-Boyau and Grün, 2011*; *Joannes-Boyau, 2013*). The basic principles of ESR dating can be found in *Grün (1989)*, *(1997)*, and are briefly summarized here with reference to hominin teeth.

When the highly crystalline material (hydroxyl-apatite) that forms tooth enamel is exposed to ionizing radiation, resulting from the radioactive decay of naturally occurring radiogenic isotopes (mainly U, Th and K) and cosmogenic rays, unpaired electrons in the crystal lattice can be moved from their normal valence band to a higher energy level or excitation state. Some of these excited electrons become trapped in charge deficit sites within the crystal structure to form paramagnetic centres that can be measured with ESR spectroscopy. The number of trapped electrons builds up over time as a function of the strength of the gamma ray intensity (or dose rate) in the surrounding environment. For teeth buried in cave sediment deep underground, radiation is mainly derived from radioisotopes contained in the surrounding sediments and the tooth itself. Thus, a tooth acts as a natural dosimeter in which the total accumulated dose and dose rate can provide an age estimate (*Grün, 1989*, *1997*). The ESR signal contained within a tooth and the dose rate can be measured directly. However, the dose rate can vary over time as U can be highly mobile in wet cave environments (e.g., *Grün, 2009*), and can move in and out of teeth (*Grün et al., 2008a*). The environmental dose rate can be measured using in situ gamma ray spectrometry, or can be determined from measured values of natural isotopes (mainly U, Th and K) and calculations of the cosmic ray contributions following models. The internal dose rate (inside the tooth) can be determined by measuring the present day $^{230}$Th/$^{234}$U-ratios from which an uptake history for the tooth can be modelled (as opposed to applying assumed uptake models).

Using the combined results of U-Th analyses and ESR measurements, more accurate age estimates can be obtained than would be possible with ESR dating alone. To overcome the problem of not accurately knowing the complex U uptake history of a tooth, the U-Th data is used to establish the relationship between the ESR equivalent dose and apparent U-Th age as defined by the equation:

$$U(t) = U_m (t/T)^{p+1} \tag{1}$$

In which U(t) is the U concentration at t, $U_m$ is the measured U concentration, T is the age of the sample and p the uptake parameter linked to the uptake model determined from U-Th measurements in the tooth (closed system or early uptake: p= −1; linear uptake: p=0, late uptake: p>0 (**Grün, 2009**).

The methodologies used to obtain U uptake models and ESR dates are explained separately for each of the laboratories. For all reported ages, decay constants for $^{234}$U and $^{230}$Th are from **Cheng et al. (2000)**.

## Methodology for combined U-series and ESR dating: Southern Cross University (SCU)

**Sample preparation** – Small fragments of enamel were removed from each of the four teeth with the help of a hand-held diamond saw following protocols in Grün et al. (2008b) and **Joannes-Boyau, 2013**. Any dentine attached to the enamel fragment was removed (1841 had no observable dentine), after which a layer of 100 μm was removed from the outer surface of each fragment with a rotary tool. For each tooth, the enamel fragments and a section of dentine directly underlying each enamel fragment were analysed for U and Th concentrations. U-series analyses were also performed on enamel and dentine from the remaining tooth in the immediate vicinity of the fragment to assess variations in U/Th isotopic ratios and calculate ESR ages.

**Dating protocols: ESR dose evaluation** – ESR dating was performed at room temperature on a Freiberg MS5000 ESR X-band spectrometer at a 0.1mT modulation amplitude, 10 scans, 2 mW power, 100G sweep, and 100 KHz modulation frequency. At SCU, the samples were irradiated with X-rays in a Freiberg X-ray irradiation chamber, with a Varian VF50 X-ray gun at a voltage of 40KV and 0.5mA current, with dose rate calibrations depending on the output value of the X-ray gun. ESR intensities were extracted from T1-B2 peak-to-peak amplitudes of the ESR signal of enamel (**Grün, 2000a**).

Each enamel fragment was mounted into a para-film mould within a Teflon sample holder to record the angular dependency in the ESR response. Irradiation was performed by exposure to the X-ray gun with no shielding of the source. To estimate the equivalent dose ($D_E$), each fragment was irradiated in steps with exponentially increasing irradiation times (for samples 1767, 1788 and 1810, these steps were: 90s, 380s, 1080s, 1800s, 3600s, 7200s, 14400s, 2,8800s and 7,9200s, and for sample 1841: 90s, 380s, 1080s, 1800s, 3690s, 7200s, 1,4400s, 2,8800s, 4,3200s and 9,9000s). During each irradiation step, the output of the X-ray gun was recorded to calculate the dose rate received by the sample (for samples 1767, 1788, 1810 and 1841 these values were 0.178Gy/s, 0.169Gy/s, 0.188Gy/s and 0.240Gy/s respectively. For each irradiation step the fragment was measured over 180° in x, y and z-configurations (**Joannes-Boyau and Grün, 2011**; **Joannes-Boyau, 2013**). Isotropic and baseline corrections were applied uniformly across the measured spectra. The amount of NOCOR's in the natural signal was estimated using the protocol of **Joannes-Boyau (2013)**. The ESR dose response curves were obtained by using mean ESR intensities and associated standard deviations from the repeated measurements.

Fitting procedures were carried out with MCDOSE 2.0 software using a Markov Chain Monte Carlo (MCMC) approach based on the Metropolis-Hastings algorithm. The program uses a Bayesian framework, where the solution is presented as a full probability distribution of the equivalent dose (**Metropolis et al., 1953**). $D_E$ values were obtained by fitting a Single Saturating Exponential (SSE) at the appropriate maximum irradiation dose ($D_{max}$) following **Duval and Grün (2016)** (using 1767 $D_{max}$ = 1264 Gy, 1788 $D_{max}$ = 2465 Gy, 1810 $D_{max}$ = 2735 Gy and 1841 $D_{max}$ = 3526 Gy).

**Dating protocols: U series analysis** – The U and Th concentrations in each of the enamel fragments and surrounding dentine were measured by laser ablation, using an ESI NW193 ArF Excimer laser coupled to a MC-ICPMS Neptune Plus at the UoW. Sections of enamel and dentine were mapped using small rasters to document compositional variability in the teeth and constrain diffusion processes (**Figure 9a**).

A total of 86 separate U-series analytical runs were performed across the dentine and enamel surface for the three *H. naledi* teeth. Each individual run consists of an average value obtained across a raster or ablation track measuring 200 μm x 700 μm in size; that is, a succession of short measurements was taken along a raster and then averaged into one value. Measurements were performed at an ablation rate of 20 Hz and a translation speed of 50 μm/s.

The rasters were positioned in a series of transects following the growth axis of the dental tissue in locations immediately adjacent to the area from which the ESR fragment was taken, and on the ESR fragments themselves (*Figure 9a*). To evaluate the ESR internal dosimetry of each tooth, three zones were analysed: (i) one along the enamel growth axis; (ii) one perpendicular to the Dentine Enamel Junction (DEJ), and; (iii) one on the dentine (*Figure 9a*). Additional measurements were taken across each tooth in areas away from the ESR fragment to assess U variability and diffusion gradients. The U-series age estimates (*Table 4*) are based on the average U-series isotope values for each ablation run. No age calculations were done for areas of the tooth where the U concentrations were below 0.5 ppm or where the U/Th ratio was below 250 (marked in red in *Table 4*). Ages were calculated with Isoplot 3.75 (*Ludwig, 2012*), and uncertainties are reported as 2 σ.

Baseline and drift were corrected using analysis of the NIST 612 glass standard, while two coral standards (the MIS7 Faviid and MIS5 Porites corals from the Southern Cook Islands; *Woodroffe et al., 1991*) were used to correct $^{234}$U/$^{238}$U and $^{230}$U/$^{238}$U ratios and assess the accuracy of measurements. Each coral standard was analysed by solution MC-ICPMS at UoW and used for reference. To account for potential matrix effects, a bovid tooth fragment from South Africa with known isotope concentrations (U-series at equilibrium) was used to verify measurements. To account for tailing effects, measurements were carried out at half-masses of 229.5 and 230.5 for $^{230}$Th and 233.5 and 234.5 for $^{234}$U.

## Methodologies for combined U-series and ESR dating: CENIEH, Griffith University and Australian National University

**Sample preparation –** One enamel fragment was extracted from each of the two *H. naledi* teeth 1788 and 1810, and analyzed.

**Dating protocols: ESR dose evaluation –** The two enamel fragments (samples 1788 and 1810) were measured by ESR following the analytical procedure developed by *Grün et al. (2008b)*. Dose evaluations were carried out at the CENIEH using a sub-exponential dose step distribution (*Grün and Rhodes, 1992*) for the following gamma doses (6.705 Gy/min): 0, 13.4, 40.2, 93.9, 201.2, 415.7, 844.8, 1649, 3058, 5471, 8690, 12713, 17943, 25989, 34861 and 61412 Gy.

ESR measurements were performed at room temperature with a EMXmicro 6/1 Bruker ESR spectrometer coupled to a standard rectangular ER 4102 ST cavity, with the following acquisition parameters: 1–10 scans, 2 mW microwave power, 1024 points resolution, 15 mT sweep width, 100 kHz modulation frequency, 0.1 mT modulation amplitude, 20 ms conversion time and 5 ms time constant. To ensure reproducible measurements, each fragment was mounted in a para-film mould within a Teflon holder in a Z-configuration only (*Grün et al., 2008b*), which can be inserted into a Bruker ER 218PG1 programmable goniometer. Because the ESR signals in fragments show very strong angular dependences, the ESR spectra of each dose step were recorded every 10° over 360°. ESR intensities were extracted from T1-B2 peak-to-peak amplitudes of the ESR signal of enamel (after *Grün, 2000a*).

Fitting procedures were carried out with Microcal OriginPro 9.1. software using a Levenberg-Marquardt algorithm by chi-square minimisation. Data were weighted by the inverse of the squared ESR intensity $(1/I^2)$ (*Grün and Brumby, 1994a*). The dose response curves (DRC's) were obtained by averaging the T1-B2 ESR intensities recorded for all the angles at a given irradiation dose. The final $D_E$ values were obtained by fitting a Single Saturating Exponential (SSE) through the ESR intensities and by selecting the appropriate maximum irradiation dose ($D_{max}$) in order to avoid dose overestimation (*Duval and Grün, 2016*; *Table 10*).

**Dating protocols: U series analysis –** Laser ablation U-series analyses were carried out at the Australian National University (ANU), using a custom-built laser sampling system interfaced between an ArF Excimer laser and a Finnigan Neptune MC-ICP-MS (*Eggins et al., 2003*, *2005*), following principles and procedures described in *Grün et al. (2014)*. To account for tailing of $^{238}$U into the ion counters for $^{234}$U and $^{230}$Th, measurements were carried out at half-masses of 229.5 and 230.5 for $^{230}$Th and 233.5 and 234.5 for $^{234}$U. The tailing corrections were 7 cts/V($^{238}$U) and 17 cts/V($^{238}$U) for $^{230}$Th and $^{234}$U, respectively.

U-series data were obtained from distinctive spot analyses on polished surfaces. Firstly, all spot locations were cleaned with the laser set to 263 μm for 10 s, followed by the analytical run with the laser set to 203 μm for 60 s. The rim from the cleaning run is clearly visible in *Figure 9*. The enamel of sample 1788 (*Figure 9b*) was analysed along two transects at an oblique angle to the surface to

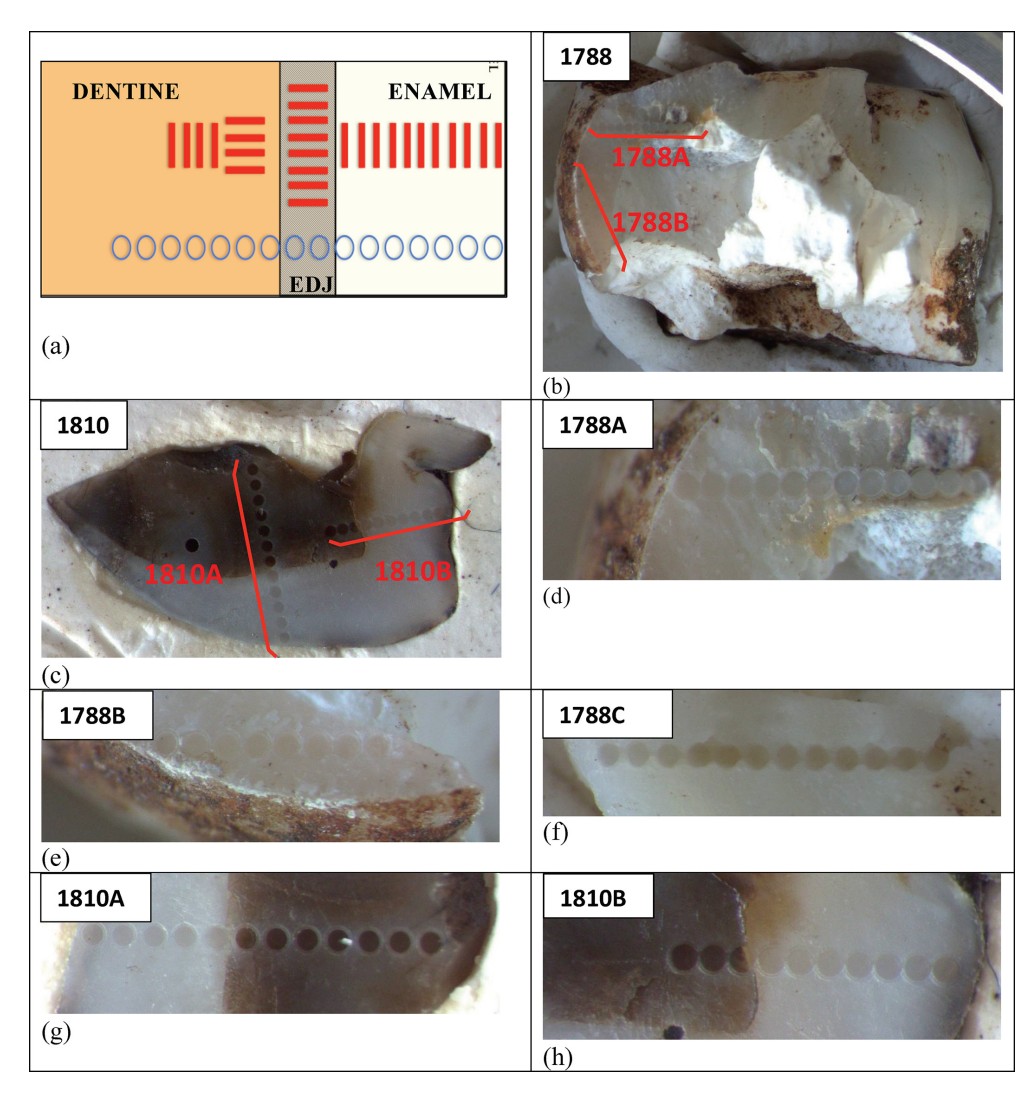

**Figure 9.** Photographs illustrating the sampling approaches taken by SCU-UoW and GU-ANU in obtaining the U-Th results presented in *Tables 4* and *5*. (a) Comparison of sampling grids across the enamel-dentine boundary measured by SCU-UoW (red lines) vs. GU-ANU (blue circles). SCU-UoW (red lines) measured a series of parallel, shallow (<5 µm) pits along grid lines across the teeth and averaged U concentrations across each grid. GU-ANU (blue circles) measured the average composition of the tooth in single spots that were laser-bored along profiles across the teeth, and report results for each spot. (b, c) Locations of LA-ICP-MS spot analyses for teeth samples 1788 (b) and 1810 (c) conducted by GU-ANU. The detailed transects are shown in panels (d) to (h).

increase the number of analyses across the enamel layer. The dentine was analysed on a piece that separated just below the two enamel tracks. The results from the enamel spots in transect A for sample 1788 are biased towards the domain close to the dentine. It was further noted that the last two analyses contained some dentine material, and these analyses were, therefore, neglected (*Figure 9d*). Sample 1810 was analysed along two tracks perpendicular to the enamel surface (*Figure 9c*). For sample 1810, analyses of enamel spots only show higher U concentrations in transect B (*Figure 9h*) where the spots occur close to the dentine boundary, while in transect A (*Figure 9g*) all enamel spots have uniformly low U concentrations. The average enamel concentrations were derived from the averages of the two transects.

The analytical data of the enamel and dentine sections (*Table 5*) were combined to provide the data input for the ESR age calculations. Because the enamel layer analyzed by ESR was not cleaned

on both sides, an average external alpha dose rate component of 8 ± 2 µGy a$^{-1}$ was used based on the alpha attenuation values of *Grün (1987)*.

Apparent U-Th ages were calculated with Isoplot 3.75 (*Ludwig, 2012*), and uncertainties are reported as 2σ. No age calculations were done for areas of the tooth where the U concentrations were below 0.5 ppm or where the U/Th ratio was below 250 (marked in red in *Table 5*). Ages were calculated assuming closed system behaviour (CS), and compared with age results assuming continuous diffusion models (after *Sambridge et al., 2012*).

## Determining the environmental dose rate

Apart from radiation derived from U and Th contained within the dental tissue of the teeth, age calculations must take the environmental dose rate into account. The environmental dose rate mainly results from gamma and beta radiation derived from the immediate surroundings of the teeth with an additional component from cosmic radiation, which in a deep cave environment is usually very small. In calculating the ages we have used a cosmic dose rate of 15 ± 1 µGy a$^{-1}$ for all samples assuming that the fossils were overlain by a 20 m thick roof of dolomite with a density of 2.80 ± 0.05 g/cm$^3$ (*Barbouti and Rastin, 1983*).

Gamma and beta radiation from the immediate surroundings of the teeth comes from U, Th and K in the sediments in which the teeth are embedded, and is partly attenuated by the water content of the sediments. Thus, the U, Th, K and water content of the sediments needs to be measured (*Table 6*). The measured water content in sediment surrounding sample 1841 is 27.7%. This value is expected to vary over time, but not by much given that the Dinaledi Chamber occurs deep inside the cave, close to the water table, where sediments are expected to have always been close to water saturated. Therefore, in calculating the ages we have assumed the water content to be 25 ± 10%. Uranium, Th, K concentrations were measured by ICP-MS at SCU and at the UoW using both leaching (in a 1:3 mixture of nitric and hydrochloric acid) and total dissolution (in a 1:3 mixture of nitric and hydrochloric acid with additional HF using an Agilent 7700 solution and a Thermo Icap) methods. Leaching results gave lower U concentrations reflecting incomplete dissolution of all U-bearing phases. Only the total dissolution results (*Table 6*) have been used to calculate the environmental dose rate.

If the teeth were not fully covered by sediment up to a depth of about 30 cm (which is the average attenuation length of gamma-rays in material with a density of ~2.5 g/cm$^3$) an estimate has to be made for the burial history of the teeth as well as the more general background radiation in the cave. The environmental dose rate in the Dinaledi Chamber was measured in situ with a portable gamma ray spectrometer on the surface of Unit 3, next to the places where samples 1767, 1788 and 1810 were collected. We measured dose rates for Th and K, that were equal to or somewhat higher than those derived from the chemical analysis of the sediments, which suggests that the background gamma radiation in the cave may be a little higher than in the sediments. However, the in situ gamma measurements varied considerably from one place to the next, whereas the measured concentrations of U, Th and K in sub-units 3a and 3b are consistent (*Table 6*). Samples 1767, 1788 and 1810 were collected from near surface where they may have resided for a long time, and, therefore, may have received between 25% and 50% of their gamma dose rate from the wider cave environment. However, the resulting gamma dose rates for this scenario are not significantly different from a situation in which the teeth are fully buried when considering the spectrometer readings. Given the uncertainties with the in situ gamma ray spectrometer measurements (see below), we are confident that the analytical data of the sediment provide more robust constraints for calculating the environmental dose rates. Therefore, the teeth have been modelled as if fully encased in sediment assuming infinite matrix dose rates for sediment and using the measured values of U, Th and K listed in *Table 6*.

The gamma dose rates calculated for sediment collected from the immediate vicinity of the sampled teeth (*Figure 10*) were compared with gamma dose rates calculated from U, Th and K concentrations for sediment samples collected at regular intervals along a 53 cm deep, vertical profile in the sondage at the center of the excavation pit (*Figure 10*). For this profile samples 1 to 4 were collected from a 13 cm thick layer of hominin-bearing sediment belonging to sub-unit 3b, and samples 5–8 were collected from underlying sediments belonging to sub-unit 3a, to a total depth of 53 cm. The variation in the gamma dose rate shows no clear trend with depth (*Figure 10*), and the samples along the vertical profile were used to obtain an average gamma dose rate. Sample 2 from the top

3 cm of Unit 3 was excluded because of anomalously high Th concentrations (this sample also contains many small bone fragments).

An additional problem in estimating the gamma radiation for the sediments may arise if $^{222}$Rn loss occurs as a result of degassing. This is of importance, because virtually all (98%) of the gamma radiation generated in the $^{238}$U decay chain is produced by $^{222}$Rn and its daughters (e.g., *Guérin et al., 2011*). If $^{222}$Rn loss is detected, then corrections have to be made. In situ gamma ray spectrometer measurements yielded K and Th values broadly in agreement with analysed concentrations, but did not detect U. Since U is determined via the (gamma) emission in the decay of $^{214}$Bi, which is a post-$^{222}$Rn nuclide in the $^{238}$U decay chain, we conclude that the absence of this signal is due to $^{222}$Rn loss (*Aitken, 1985*). The $^{222}$Rn loss was confirmed by analyzing a sediment sample that was collected next to sample 1841, for which high resolution, Ge-gamma spectrometry showed ~80% Rn loss.

It is difficult to estimate whether Rn degassing occurred during the entire burial history of the teeth. Therefore, age calculations were performed for two different scenarios: (i) a situation where it is assumed that the present day situation of 80% Rn loss persisted during the entire burial history, and (ii) a situation in which no Rn loss occurred (*Table 7*). Assuming water content of 25 ± 10%, the average gamma dose rate for the sediments has been calculated at 724 ± 116 µGy.a$^{-1}$ assuming no radon loss, and 534 ± 69 µGy.a$^{-1}$ assuming 80% Rn loss (*Figure 10*).

## Combined US-ESR age calculations

All age calculations are provided with 2σ uncertainties, and were carried out using the US-ESR program of *Shao et al. (2014)*, which utilizes the dose rate conversion factors of *Guérin et al. (2011)*. In doing the calculations, the input parameters and criteria listed in *Table 6* were used, and age results are given in *Table 7*. In all calculations we have made the following assumptions: (i) post-$^{230}$Th daughter elements are in equilibrium in dental tissues, which is the standard assumption in ESR dating; and (ii) complete (effective) burial of the samples occurred (i.e., infinite matrix assumption for the gamma dose rate measured in sediment).

Results have been calculated for the two scenarios (*Table 7*): (1) 80% Rn loss in the sediment; and (2) post-Rn equilibrium in dental tissue and sediment (i.e., no radon degassing). In both scenarios the water content has been taken as 25 ± 10%. Scenario 1 is based on measured, current-day values and probably provides a maximum age estimate, because it is unlikely that the high degree of Rn

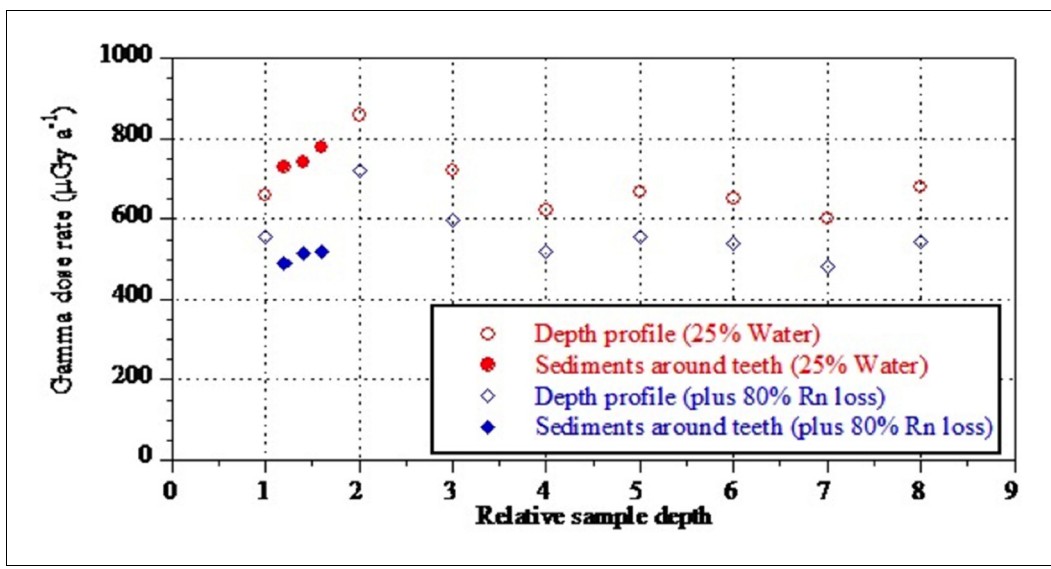

**Figure 10.** Gamma dose rate reconstructions derived from analytical data of sediment samples collected around ESR samples 1767, 1788 and 1810 (closed circles and diamonds), combined with samples from a vertical profile in the excavation pit and sondage (open circles and diamonds). The data show little variation in dose rate with depth (see text for explanation).

loss observed today would have occurred throughout the history of the cave. Scenario 2 will provide a minimum age estimate, because Rn loss has been observed. These two scenarios provide the most reasonable range for the age of the fossil teeth.

Additional modelling was carried out to assess the effects of early U uptake on the ESR age, in a closed system setting (CSUS-ESR, *Grün, 2000b*). This showed that the differences in US-ESR and CSUS-ESR model ages are less than 4%, thats is, which is well within the uncertainty ranges of the listed US-ESR ages (*Table 7*). Therefore, CSUS-ESR ages are not further considered.

## ESR dose evaluation: inter-laboratory comparison

Each laboratory (SCU and CENIEH) independently measured the $D_E$ values following the methods outlined above. To ensure reliable $D_E$ values, the maximum irradiation dose ($D_{max}$) was selected in accordance with the recommendations made by *Duval and Grün (2016)*, in order to avoid $D_E$ over-estimation (i.e., for $D_E$ values between 100–500 Gy, the $D_{max}/D_E$ ratio has to be kept between 5–10; for $D_e$ values between 1000–2000 Gy the $D_{max}/D_E$ ratio has to be kept between 1–2; *Table 10*). Nor-malised dose response curves (DRC's) are shown in *Figure 11*.

Samples 1767, 1788 and 1810 have $D_E$ values that vary within a narrow range between 200 and 300 Gy, while the $D_E$ value of sample 1841 is significantly higher (>1500 Gy). For the two samples that were measured by both laboratories, the SCU procedure systematically provides higher $D_E$ results. These differences partly result from different fitting procedures: when plotting the SCU experimental data points using the CENIEH-GU fitting procedure, the calculated $D_E$ values shift by < 2% for samples 1767, 1788 and 1841, and −6% for sample 1810 (*Table 10*). Both laboratories also used different irradiation sources (gamma-rays at CENIEH, and X-rays at SCU), which may have had some influence on results. A third factor that may have most fundamentally influenced the differ-ence in measured $D_E$ values is the radiation sensitivity observed for each fragment (*Figure 11*); for sample 1810, the DRC's from CENIEH and SCU are similar, but for sample 1788 they are different, indicating that the response of the two fragments from sample 1788 to the radiation source was dis-tinctly different. Each laboratory had sampled its own fragment from different domains of the tooth, with different radical concentrations and crystallinity, and the U-concentrations in the fragment used by SCU were somewhat higher than those used by GU-CENIEH (*Tables 4* and *5*). However, in spite of these differences, the ages agree within error (*Table 7*).

The use of enamel fragments in measuring the dose allows us to differentiate between the rela-tive contributions of non-oriented $CO_2^-$ radicals (NOCOR's) versus anisotropic radicals (AICOR's) (e. g., *Grün et al., 2008b*). *Joannes-Boyau and Grün (2011)* recently showed that gamma irradiation in the laboratory may produce additional unstable NOCOR's when compared to gamma irradiation in a natural environment, which could lead to an overestimation of the measured dose if this contribu-tion is not removed. To evaluate this possible bias, we followed Method 3 described in *Joannes-Boyau (2013)* for the extraction of the NOCOR's from the main radiation-induced ESR signal (*Fig-ure 12*). When a fragment is rotated over 360°, the ESR intensity varies. The ratio of the maximum ($I_{max}$) and minimum ($I_{min}$) intensities ($I_{max}/I_{min}$) is used as a proxy to quantify the increased anisotropy potentially induced by either gamma or X-ray irradiations. The evolution of this ratio with the irradia-tion dose is shown in *Figure 12*. When comparing the results from the two laboratories it is clear that both obtain similar results. For measurements conducted at CENIEH the $I_{max}/I_{min}$ ratios for

**Table 10.** ESR fitting results obtained by SCU and CENIEH-GU. Both laboratories employed a Single Saturating Exponential (SSE) fit-ting function. $D_{max}$ was selected in accordance with *Duval and Grün (2016)* to avoid $D_E$ overestimation. SCU results in brackets show $D_E$ values that were obtained by SCU using the CENIEH-GU procedure (see text for details).

| Sample | SCU | | | Cenieh-gu | | |
| --- | --- | --- | --- | --- | --- | --- |
| | $D_E$ (Gy) | $D_{max}$ (Gy) | $D_{max}/D_E$ | $D_E$ (Gy) | $D_{max}$ (Gy) | $D_{max}/D_E$ |
| 1767 | 194 ± 4 (*193 ± 6*) | 1264 | 7 | – | – | – |
| 1788 | 232 ± 8 (*232 ± 22*) | 1204 | 5 | 159 ± 11 | 1649 | 10 |
| 1810 | 296 ± 14 (*281 ± 34*) | 2735 | 9 | 232 ± 29 | 1649 | 7 |
| 1841 | 1676 ± 127 (*1648 ± 500*) | 3526 | 2 | – | – | – |

samples 1788 and 1810 remain near-constant (between 1.22–1.24) as the gamma dose varies. For measurements conducted at SCU the $I_{max}/I_{min}$ ratios remain near-constant as well (and vary within uncertainty, between 1.6–2.0 for samples 1767, 1788 and 1810, and remain around 1.34 for sample 1841; *Figure 12*) as the X-ray dose varies. The differences in the $I_{max}/I_{min}$ ratios for each tooth calculated by the two laboratories results from the orientation chosen, the spatial variations of the ESR signal within the enamel layer, and the different crystal orientation and fragment position during ESR measurements. The fact that the results from both laboratories show constant ratios indicates that no additional NOCOR's have been created by both gamma and X-ray irradiations. Consequently, no corrections had to be applied to the measured $D_E$ values.

## OSL dating
### Background to OSL dating
Optically stimulated luminescence (OSL) dating is a widely used method for estimating the last time that sediments were exposed to sunlight within the past 300–500 ka (e.g., *Aitken, 1985*; *Murray and Wintle, 2000*). Sand-sized grains of quartz and K-feldspar (two commonly occurring minerals) are preferred for OSL dating. These grains may be measured collectively in a single composite sample (an aliquot), which provides an average signal, or as individual grains, which provides greater insights into the depositional history of the sediments.

OSL dating is a radiation dosimetric dating technique based on the time-dependent accumulation of radiation damage in minerals (*Adamiec and Aitken, 1998*; *Murray and Wintle, 2000*), as a result of exposure to low levels of ionising radiation in the environment. The intensity of radiation damage reflects the total amount of energy (the 'equivalent dose') absorbed from background radiation by the mineral over time. The radiation damage can be removed from the mineral by exposure to heat or light, which is accompanied by the release of a small amount of light or luminescence. OSL dating

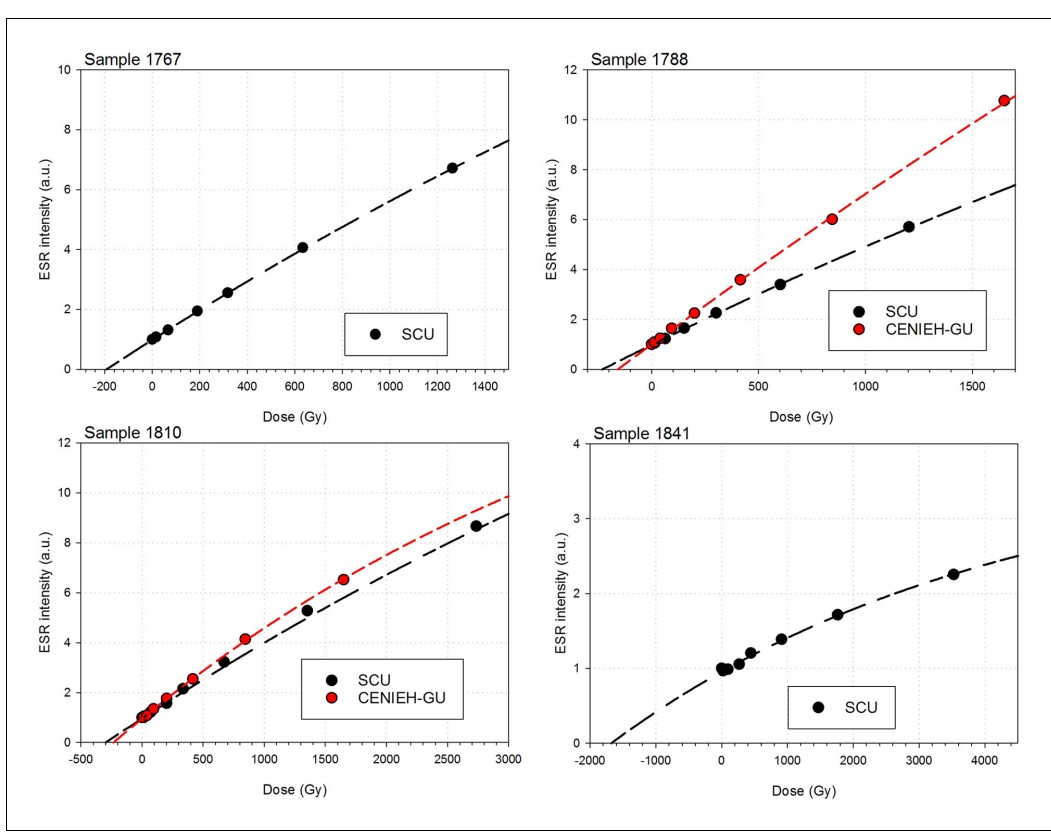

**Figure 11.** ESR dose response curves (DRC's) obtained for the samples 1767, 1788, 1810 and 1841. To facilitate comparison, all DRC's have been normalised to the intensity of the natural point (=1).

estimates the age when sediment was last exposed to light ('bleached') during erosion, transport and deposition of the mineral grains. Once sediments are buried and no longer exposed to sunlight, the luminescence signal can start to build up again (**Duller, 2007**).

The latent luminescence signal that has built up in a sediment sample can be released and recorded in the laboratory using light (the OSL technique). This luminescence signal is related to the environmental radiation dose the mineral has received since the last exposure to sunlight. If the environmental dose rate is determined together with the equivalent dose contained in the sample, it is possible to determine an age for the sediment, or rather a depositional age which measures the last time the sediment was exposed to sunlight or heat.

## Methodology for OSL dating: University of the Witwatersrand (Wits)

**Sample preparation** – For samples OSL3, 4 and 5, sediment was removed from the sample tubes under controlled, safe-light laboratory conditions. Material located within 2 cm of the ends of the tubes was removed to isolate any quartz grains potentially exposed to light during sampling. This material was used to measure water content, and to determine the dosimetry of the sample. The remaining sediment was treated with 33% hydrochloric acid and 20% hydrogen peroxide to remove carbonate and organic components. Quartz grains were isolated from denser minerals and feldspars by using solutions of sodium poly-tungstate with specific gravities of 2.70 g/cm$^3$ and 2.62 g/cm$^3$, respectively. After rinsing, drying and sieving, the fine sand (180–212 μm) fraction was etched for 40 min in 40% hydrofluoric acid to remove the outer layer (~10–15 μm-wide) affected by alpha radiation and any remaining feldspars. Subsequently, 33% hydrochloric acid was added to remove acid soluble fluorides. Each sample was dried and re-sieved in preparation for equivalent dose determination.

**Determining the Equivalent Dose ($D_e$)** – OSL measurements were carried out on the 180–212 μm grain-size fraction. A monolayer of grains was prepared on steel discs in aliquot sizes of ~30 grains. Luminescence measurements were performed on an automated Risø TL/OSL -DA-20 reader. $D_e$ values were obtained through calibrating the 'natural' optical signal acquired during burial, against 'regenerated' optical signals obtained by administering known amounts of laboratory dose.

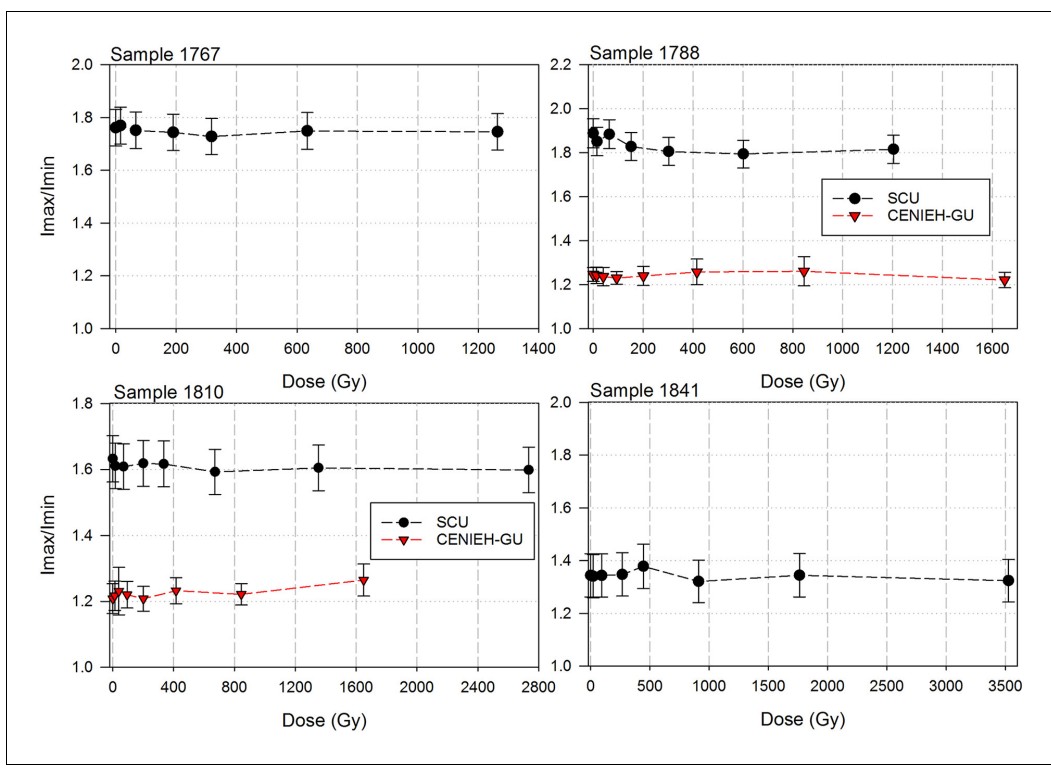

**Figure 12.** Evolution of the $I_{max}/I_{min}$ ratio vs the irradiation dose for the four tooth samples. (see text for explanation).

$D_e$ estimates were obtained using the Single-Aliquot Regenerative-dose (SAR) protocol of *Murray and Wintle (2000)*. An appropriate preheat temperature was determined using a 'preheat-dose recovery test'. A test dose of 27 Gy was used to monitor sensitivity change. Up to five different regenerative doses were given to define the linear portion of the dose response from which $D_e$ was determined. The OSL signals were stimulated at 125°C for 40 s with blue LED at 90% power and a final OSL measurement was used to clear out any signal that may be found in thermally unstable traps. The $D_e$ was evaluated using an exponential and linear fit to a single regenerative dose point. Errors on $D_e$ assume poisson statistics, with an additional 1% uncertainty added to each OSL measurement (*Galbraith, 2002*; *Duller, 2007*).

To obtain a luminescence age, a burial dose ($D_b$) is calculated from the aliquot measurements, in which individual values for $D_e$ are combined to obtain a single value for $D_b$. Given the often complex distribution of $D_e$ values, $D_b$ is calculated using complex statistical models including the central age model (CAM), and the minimum age model (MAM) (*Galbraith et al., 2005*).

Measured $D_e$ values for quartz grains show a degree of variability that depends in part on the crystal lattice structure of the grains and their burial history (e.g., complete vs incomplete bleaching). For well-bleached quartz where aliquots have been heated prior to irradiation the statistical variability, expressed as the standard deviation, in measured $D_e$ values will be less than 20% (*Olley et al., 2004*; *Galbraith et al., 2005*). This variability in the distribution is referred to as overdispersion (*Galbraith et al., 1999*). An overdispersion value of 20% is used to guide the use of appropriate statistical models to calculate a burial age: MAM is used to determine the burial age when overdispersion exceeds 20%; CAM is used when overdispersion is below 20%. However, this is only a rough guide and environmental conditions need to be considered in deciding the statistical model to be used. The CAM and MAM ages were calculated with software provided by Geoff Duller (University of Aberystwyth, UK).

The Central Age Model (CAM) computes an age value much like a weighted average, but it assumes a natural distribution of $D_e$ values rather than a single value from which to calculate the age. A CAM is used when it is assumed that samples experienced homogenous bleaching. The Minimum Age Model (MAM) is used if incomplete bleaching is expected to be the reason for an observed skewed dose distribution, or if (as in this case) the sample contains mineral grains that have never been at surface (*Galbraith et al., 1999*). The MAM uses a truncated normal distribution and fits it to the individual $D_e$ data points to determine the proportion of grains that were fully bleached before they were deposited (*Galbraith and Laslett, 1993*).

**Environmental dose rate determinations** – The concentrations of U, Th and K in the sediment samples of Unit 1 (sub-units 1a and 1b) were determined at the University of Johannesburg by isotope dilution on aqua regia leaching techniques. A mixed $^{236}$U-$^{229}$Th spike (allowing measurement of (U, Th) disequilibrium) was added before leaching. U and Th were separated from major elements using ion exchange columns with Dowex 1 × 8 anion exchange resin; samples were loaded with 2 ml of 80% ethanol + 20% 5M HNO$_3$, washed with 3 ml of the same solution and eluted with 0.5N HCl. Uranium and Th were analysed together on a Nu Instruments Plasma II multi collector ICP mass spectrometer, using an APEX-Q desolvating nebulizer. Strong excesses of $^{234}$U and $^{230}$Th ($^{234}$U/$^{238}$U activity ratios of 2.2–4.5, $^{230}$Th/$^{238}$U activity ratios of 3.1–4.1) necessitated significant upward corrections to the dose rate for U, yielding age-dependent 'effective U concentrations' to be used for the dose rate calculations.

Dose rate calculations (after *Aitken, 1985*) incorporated beta-attenuation factors (*Mejdahl, 1979*), dose rate conversion factors (*Adamiec and Aitken, 1998*) and an absorption coefficient using the present water content (*Zimmerman, 1971*) with a 5% relative uncertainty to reflect potential temporal variations in past moisture content. The water content may affect the validity of the ages as it is possible that a different, possibly wetter, climate regime may have prevailed. This is suggested by the disequilibrium in the U-series measurements. The cosmic dose rate was determined as a function of altitude, latitude, longitude and depth, according to *Prescott and Hutton (1994)* equations, plus the soft component from *Madsen et al. (2005)*.

**Luminescence age determinations** – The luminescence age was obtained by dividing the palaeodose with the total dose rate. The error on the luminescence age estimates represents the combined systematic and experimental error associated with both the $D_e$ and dose rate values. There is no datum for luminescence dates, therefore the age reported is taken from date of sampling (i.e., AD 2015).

## Palaeomagnetic dating
### Methodology for palaeomagnetic dating: La Trobe University (LTU)
The procedures for palaeomagnetic analysis of the speleothem sample from Rising Star Cave follow those outlined in *Herries and Shaw (2011)* and a comprehensive review of speleothem magnetism can be found in *Lascu and Feinberg (2011)*.

**Sample preparation** – The sample was drilled vertically, across the layering, using a non-magnetic rock drill to produce three 2.5 cm by ~5–6 cm cores from the upper, purer part of the flowstone. Care was taken to remove recent contamination. The surface of the flowstone has a calcified covering of fine clastic dust that is likely to be much younger. Therefore, the upper ends of the cores were removed to make sure that only primary flowstone was measured. The cores were then cut in half at their midpoint where more recent contamination has seemingly occurred within the sample (at the interface between different growth phases; *Figure 13*). This was done using an ASC Scientific non-magnetic saw with bronze saw blades to produce a total of six 2.5 by 2.5 cm subsample cores from the upper part of the flowstone; 3 from the uppermost 2.5 cm (phase C flowstone) and 3 from the lower 2.5 cm (phase B flowstone). Thinner samples could not be produced as the samples were weak. Only a small part of the lowest, clastic rich layer (phase A flowstone) was preserved on the base of the block sample (*Figure 13a*) and this layer was cut into two 2 × 2 cm cubes for analysis.

**Palaeomagnetic field determinations** – Because the samples all consist of flowstone speleothem, alternating field (AF) demagnetisation was the sole method of magnetic cleaning that was used on the samples. AF has been shown to be effective for recovering the primary palaeomagnetic signal formed within South African speleothems for both detrital inclusions deposited during phases of flooding or within the water forming the speleothem itself (detrital remanent magnetisation; DRM), and for chemical precipitation (Chemical Remanent Magnetisation; CRM) of iron phases within dripwater (*Herries and Shaw, 2011*; *Pickering et al., 2013*). Samples were demagnetised using an AGICO LDA5 AF demagnetiser and measured using an AGICO JR6 spinner magnetometer on the high speed setting. Samples were analysed using the program Plotcore to establish the primary remanence direction using principle component analysis of Zijderveld plots. Multiple samples were taken from each layer within the flowstone speleothem and analysed using Fisher statistics and the Program FISH2 to establish the Palaeolatitude (Plat.) for each layer. This was then used to assign the polarity of each layer, with Plat. Values > + 60°/−60° are considered to represent normal or reversed polarity.

## Radiocarbon dating
### Methodology for radiocarbon dating
Radiocarbon dating is a commonly used method for dating materials that contain carbon, by using the decay of the radioactive carbon isotope $^{14}$C with a half-life of ~5730 years. Because of the relatively short half-life of $^{14}$C, this technique generally only returns reliable results for ages less than 50 ka, and it is widely used in Archaeological applications (e.g., *Taylor and Bar-Yosef, 2014*).

**Sample preparation** – Two bone fragments collected along the floor of the Dinaledi Chamber near the top of sub-unit 3b were sent to Beta Analytic Inc. in Miami, Florida. Here, the outer surfaces of the bone fragments were acid etched with 10% HCl (at room temperature) after which samples were rinsed to a neutral pH and dried. Samples were then ground to a powder and pre-treated with 1N acetic acid for 24 hr, rinsed, dried and weighed.

The pre-treated bone powder was acidified in 85% phosphoric acid at 70°C in a closed chemistry line that had been purged of any $CO_2$ (to <$10^{-15}$ atoms). $CO_2$ produced from the sample was introduced into a reaction vessel containing an aliquot of cobalt metal catalyst. Hydrogen was introduced and the cocktail heated to 500°C, to crack $CO_2$ and form carbon (graphite). The graphite was pressed into a pellet for analysis.

**Analyses and data processing** – Analyses for radiocarbon dating were performed using accelerator mass spectrometry (AMS). Analyses were calibrated with graphite produced from the NIST-4990C modern reference standard. Reported $\delta^{13}$C was measured relative to the PDB-1 on the sample itself. Total fractionation using the AMS $\delta^{13}$C correction was done to derive at a 'conventional radiocarbon age:' Ages are calculated using BetaCal 3.17 provided by Beta Analytic following procedures outlined in *Bronk Ramsey (2009)* and using the SHCAL13 database (*Hogg et al., 2013*). Dates are reported as radiocarbon years before present ('BP' with 'present' taken as AD 1950).

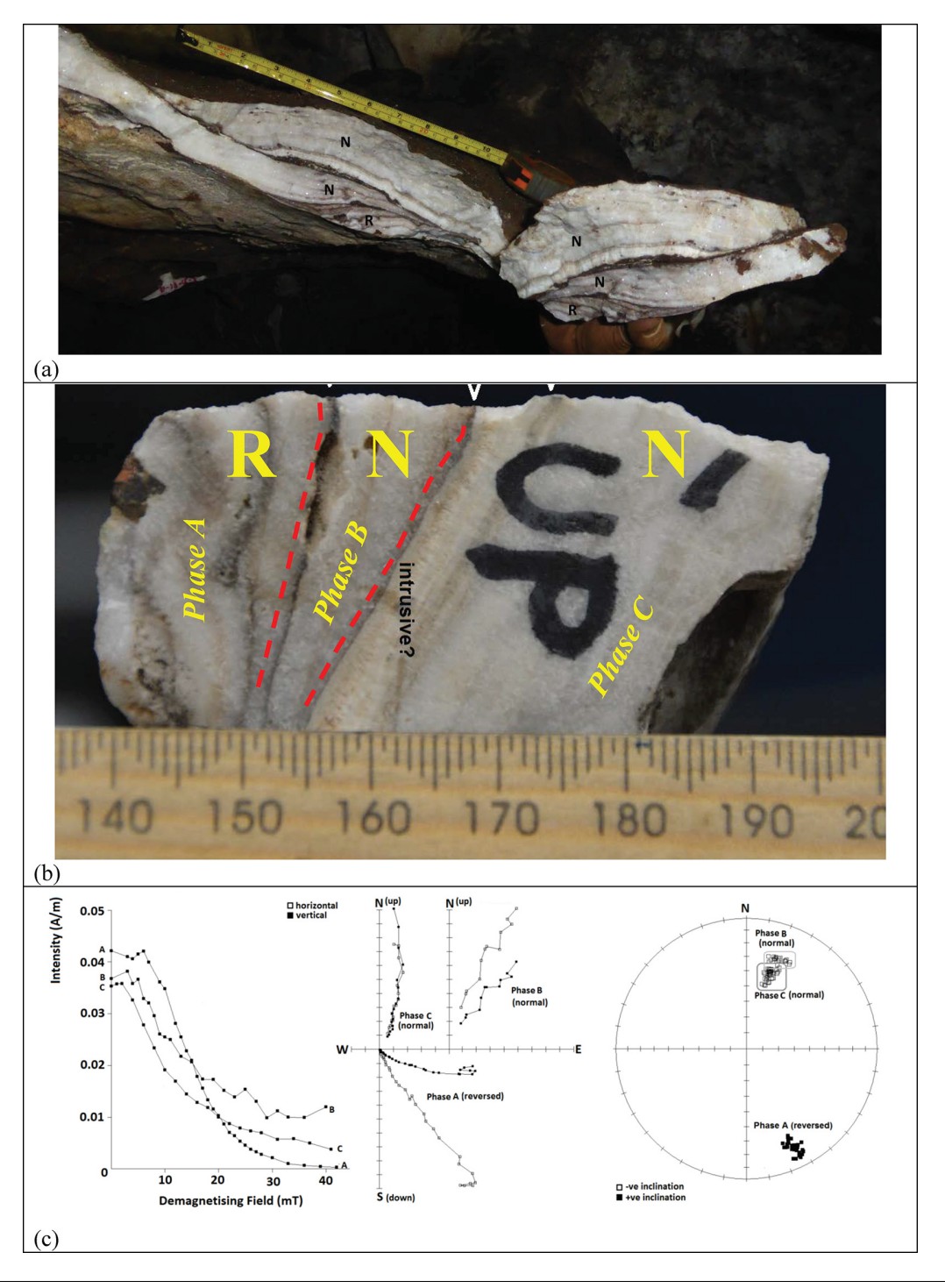

**Figure 13.** Samples and results of palaeomagnetic analyses forFlowstone 1a. (a) Outcrop photo of hanging erosion remnant of Flowstone 1a from which the palaeomagnetic sample was taken. The three flowstone phases separated by detrital horizons are clearly visible, and their magnetic polarity has been marked (N = normal; R = reverse). The stratigraphic top is towards the top of the photo; (b) close-up of a hand sample taken for palaeomagnetic analysis from Flowstone 1a in the Dinaledi Chamber. The sample is layered and comprises three distinct phases (from base to top: A-C marked in yellow) separated by thin clastic horizons that mark disconformities indicated with red dashed lines. The larger-scale extent of the three phases can be seen in (a); (c) intensity spectra, Zijderveld plots, and stereo plots for samples from phases A to C taken from (b). Phases B and C show normal polarity and phase A shows reversed and intermediate polarity directions.

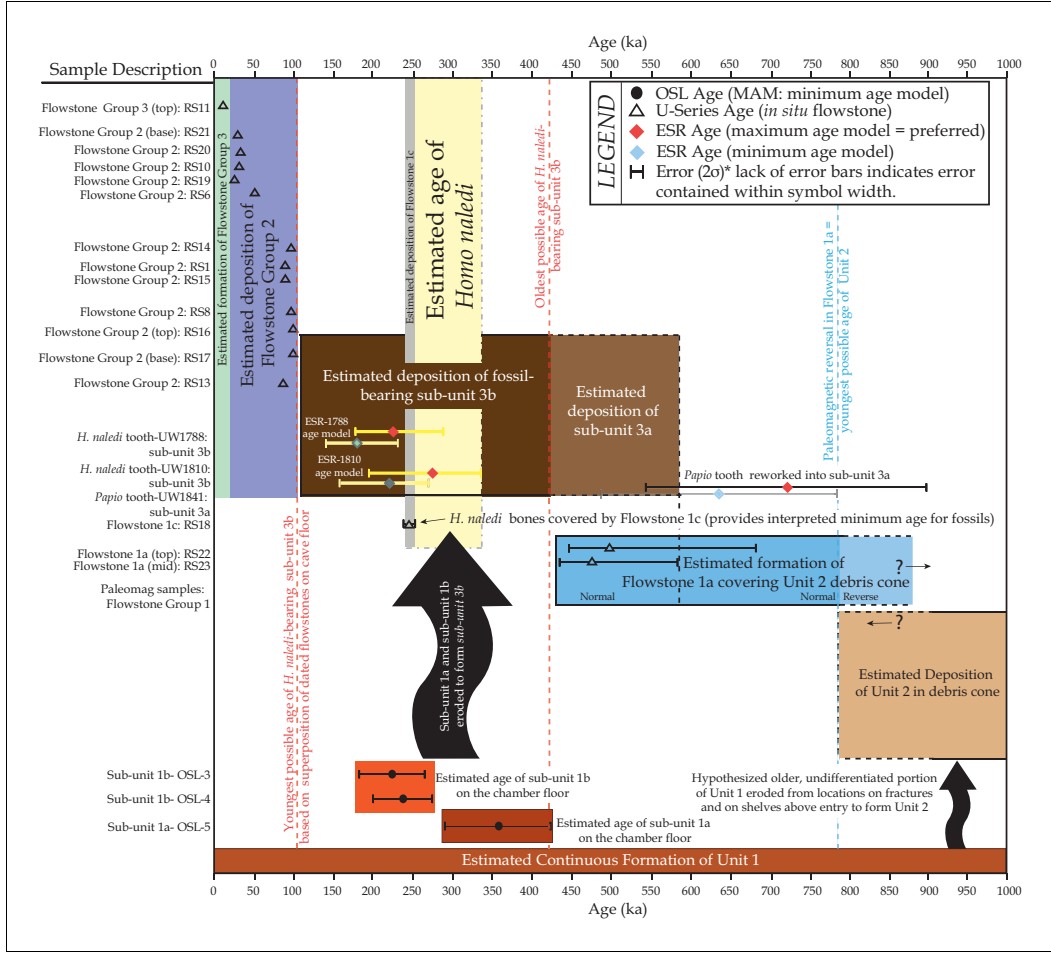

**Figure 14.** Chronostratigraphic summary of radio-isotopic dating results, and interpretation of the depositional ranges of stratigraphic units, flowstones and *H. naledi* fossils in the Dinaledi Chamber. Following the preferred US-ESR maximum age model and associated uncertainties for ESR samples 1788 and 1810, a maximum depositional age of 335 Ma was determined, while the minimum depositional age of 236 ka was constrained by Flowstone 1c (sample RS18), which covers *H. naledi* material in the entry zone.

# Acknowledgements

We would like to thank Chris Stringer and four anonymous reviewers for their constructive comments that have greatly helped improve this paper. We would also like to thank the many funding agencies that supported various aspects of this work. In particular we would like to thank the National Geographic Society, the National Research Foundation and the Lyda Hill Foundation for significant funding of the discovery, recovery and initial analysis of this material. Further support was provided by ARC (DP140104282: PHGMD, ER, JK, HHW; FT 120100399: AH). The ESR dosimetry study undertaken by CENIEH and Griffith University has been supported by a Marie Curie International Outgoing Fellowship (under REA Grant Agreement n° PIOF-GA-2013–626474) of the European Union's Seventh Framework Programme (FP7/2007-2013) and an Australian Research Council Future Fellowship (FT150100215). ESR and U-series dating undertaken at SCU were supported by ARC (DP140100919: RJB). We would also like to thank the University of the Witwatersrand, the Evolutionary Studies Institute and the South African National Centre of Excellence in PalaeoSciences for hosting many of the authors while studying the material, and allowing original material to be made available for dating. We would like to thank the South African Heritage Resource Agency for the necessary permits to work on the Rising Star site; the Jacobs family and later the Lee Berger Foundation for granting access. The assistance of members of the Speleological Exploration Club, in various

safety aspects within the cave during excavations is gratefully acknowledged. Zenobia Jacobs and Bert Roberts of the University of Wollongong are thanked for helping with initial OSL tests for single grain analyses and for helpful discussions. We would also like to thank Wilma Lawrence, Bonita De Klerk, Natasha Barbolini, Merrill van der Walt, Wayne Crichton and Justin Mukanku for their assistance during all phases of the project. RG and MD thank Les Kinsley, ANU, for his invaluable help to keep the Neptune in tune. ME would like to thank Stephan Woodborne (iThemba LABS, Gauteng) for his help in unravelling the spectrometer results related to OSL and ESR dating.

## Additional information

### Funding

| Funder | Grant reference number | Author |
| --- | --- | --- |
| Australian Research Council | DP140104282 | Paul H G M Dirks<br>Eric M Roberts<br>Hannah Hilbert-Wolf<br>Jan D Kramers<br>Carl Spandler<br>Lee R Berger |
| Australian Research Council | FT 120100399 | Andy IR Herries |
| Australian Research Council | DP140100919 | Renaud Joannes-Boyau |
| Marie Curie International Outgoing Fellowship | PIOF-GA-2013-626474 | Mathieu Duval |
| Australian Research Council | FT150100215 | Matthieu Duval |
| National Geographic Society | | Lee R Berger |
| National Research Foundation | | Lee R Berger |
| Lyda Hill Foundation | | Lee R Berger |

The funders had no role in study design, data collection and interpretation, or the decision to submit the work for publication.

### Author contributions

PHGMD, Conceptualization, Formal analysis, Supervision, Funding acquisition, Validation, Investigation, Visualization, Methodology, Writing—original draft, Project administration, Writing—review and editing; EMR, Conceptualization, Formal analysis, Supervision, Funding acquisition, Investigation, Visualization, Methodology, Project administration, Writing—review and editing; HH-W, Formal analysis, Validation, Investigation, Writing—review and editing, Visualisation (U-Th; geochemistry; sample preparation); JDK, Resources, Formal analysis, Validation, Investigation, Writing (U-Th, OSL, geochemistry); JHa, Conceptualization, Visualization, Writing—review and editing; AD, JHe, CJP, Formal analysis, Investigation (U-Th); MD, Resources, Formal analysis, Validation, Investigation, Writing—review and editing, Visualisation (ESR); MEl, Supervision, Investigation, Visualisation (archaeology, speleology); MEv, Formal analysis, Investigation (OSL); RG, Formal analysis, Validation, Investigation, Resources (ESR); AIRH, Formal analysis, Investigation, Writing (Palaeomagnetism); RJ-B, Formal analysis, Validation, Investigation, Writing—review and editing, Visualisation (ESR); TVM, Formal analysis, Investigation (geochemistry); JR, JWi, Investigation (field sedimentology); CS, Validation, Writing—review and editing; JWo, Formal analysis, Investigation, Resources (U-Th); LRB, Conceptualization, Resources, Funding acquisition, Project administration, Writing—review and editing

### Author ORCIDs

Paul HGM Dirks, http://orcid.org/0000-0002-1582-1405
Renaud Joannes-Boyau, http://orcid.org/0000-0002-0452-486X
Lee R Berger, http://orcid.org/0000-0002-0367-7629

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
