## [Decision Letter]

Thank you for submitting your article "The age of *Homo naledi* and associated sediments in the Rising Star Cave, South Africa" for consideration by *eLife*. Your article has been reviewed by five peer reviewers, and the evaluation has been overseen by a Reviewing Editor and Ian Baldwin as the Senior Editor. One of the five reviewers, Chris Stringer, has agreed to share his name. The other reviewers remain anonymous.

The reviewers have discussed the reviews with one another and the Reviewing Editor has drafted this decision to help you prepare a revised submission.

This paper provides the first dating results for the recently discovered and described *Homo naledi* taxon. As such, the potential importance of the results for our understanding of hominin evolution is very high. The reviewers were in agreement that the paper’s base is strong, with separate, generally consistent results obtained by the two excellent labs. There were, however, a large number of comments raised (on which the reviewers were not in disagreement) that need to be addressed before the manuscript can be considered for acceptance by *eLife*. These concerns are considered substantial but addressable. Please note that given the number of issues that need to be addressed and the technical nature of the paper, we will ask reviewers to evaluate the revised manuscript (as quickly as possible) prior to considering it for acceptance.

As the Reviewing Editor, I am choosing to take the unusual step (for *eLife*) of providing the full reviews as part of the decision letters for the co-submitted papers for two reasons. First is the interconnectedness among the multiple papers, which is reflected in the reviews (as some reviewers participated in the review of multiple papers). Seeing the more direct reader reactions to each paper in the context of others should be helpful (and should be taken into account) as you consider the larger revision process for the entire group of papers. Second, included within the reviews are many detailed comments that are too numerous to repeat as part of the decision letters, but that also should be addressed. Note that one reviewer provided their specific comments in an attached document.

An overview summary of the most critical issues that must be fully addressed (as above, this is only a small list of critical issues raised by reviewers) prior to further consideration:

1) Critical points concerning the relationships among the various stratigraphic units were raised by Reviewer 4 (in particular, see major comments 1 and 3 from that review) that must be addressed thoroughly in any revision. During the consultation process, other reviewers highlighted the importance of these points given the interdependence of most of the age/depositional models on those interpretations.

2) The overall synthesis/analysis of the various dating results is too limited at present, and choices on what results are highlighted at various points in the manuscript seem arbitrary at times. A more concerted and comprehensive effort to assimilate the different datasets in an evenly considered, summary results section (with a synthesis view agreed to by both dating labs) would help to address this concern. This issue also filters up to the abstract.

3) The manuscript currently reads as if various sections were composed by different authors with insufficient consistency in style and level of detail. Likewise, the tables need to be reworked with more care for consistency and interpretability.

4) The reviewers also felt that the discussion should also be expanded, and can be even without overtly duplicating the broader synthesis provided in the other papers.

5) Similarly detailed treatment for reporting the radiocarbon methods and results, as the other dating methods, should be provided.

6) The reviewers also had suggestions for the order of presentation that may help the reader more readily interpret the results.

*Reviewer #1:*

I will state up front that the dating methods used in this paper are not within my field of expertise. I was asked to review this paper as one component of the entire suite of submitted papers. Thus, I will leave the methodological critiques to those more qualified than me and comment more on the implications of the dates. The authors suggest that deposition of the hominin material was over a time period of about 100k between 200-300 ka. It does seem like the resolution and precision of the dates are not great and makes it difficult to properly interpret the context of the hominins. When the duration of deposition is considered, it then seems unusual that there would be so few hominins in the cave system, particularly if it is being used as a burial site (even if only for special individuals). Given the complexity of cave systems and their formation, I don't know that the authors have convincingly demonstrated that the chamber was completely inaccessible during the times of hominin deposition (especially with the wide range of dates and low resolution). That said, I sympathize with the inherent difficulties of dating fossils from cave systems and appreciate that the authors used multiple methods to try to pin down the dates and that this may be the best that can be done at this point.

*Reviewer #2:*

Although I have reservations about the way radiocarbon dating was attempted, as explained further below, and there is a need for more caution in some areas, the paper should be suitable for publication after revisions. I have marked up a text copy of the manuscript, which I attach, and summarise some of the points below.

The references for all four papers need work for consistency of formatting, journal abbreviations, accents etc.

"The results for RS18 (~242 ka) may provide a/provide a potential minimum age for the *H. naledi* fossils in this part of the cave." Insert some caution because some other age estimates are still younger than this figure.

"Dates acquired via U-Th and ESR techniques were obtained using a double blind approach for each technique to ensure robust, reproducible results, with each laboratory using their own analytical and computational approach. Methodologies and approaches taken by each laboratory that has contributed to this paper are described in detail in the methodology section." Why wasn't radiocarbon dating pursued in the same manner, using established research labs rather than a commercial lab? Why no details of the procedures: e.g., was ultrafiltration attempted?

"Some of the ESR ages (Table 4, Table 5) independently suggest that some of the fossils around the excavation pit may be younger than 237 ka, but confirm a probable age of >200 ka for the fossils." Well a few of the ESR ages *don’t* confirm an age of 200 ka+ so this needs rewording.

*Reviewer #2 Minor Comments:*

Other suggested edits are added in an attached text version of the paper.

*Reviewer #3:*

This manuscript, "The age of *Homo naledi* and associated sediments in the Rising Star Cave, South Africa" presents five different dating techniques applied to both the fossils and cave deposits found preserved in the Dinaledi Chamber of the Rising Star Cave and combines these methods to propose a depositional age for the Homo naledi fossils of between 200 and 300 ka. New stratigraphic data on the three sedimentary units and three flowstone units is provided, as well as details on the locations and descriptions of the dating samples. The geochronological data is provided in detail with a brief discussion on the depositional age (and inferred age) of the hominin fossils.

Overall this paper presents a large new geochronological data set for an exciting site home to a unique and huge collection of hominin fossils. Knowing the age of these fossils is critical in their interpretation and there has been much speculation as to their age. The authors certainly have thrown everything at dating the fossils and deposits, which given the nature of the fossil find is to be expected. However, there are many issues, the three major ones being:

1) there is no synthesis or analysis of the various geochronological data sets;

2) there is no discussion of how the new ages fit into the regional and global picture in terms of cave formation;

3) Exactly how the age of the fossils has been determined is not abundantly clear: this is certainly the most important message of the paper and needs to be clearly and carefully explained. The existing discussion on pages 20 and 21 is confusing and no clear story emerges.

The authors presume a huge prior knowledge of the region, caves and fossils, as well as the background and methods of the various dating techniques; I am not unfamiliar with the sites in this region and these dating techniques but struggled to follow a clear story with the manuscript. The confusing order of the sections, the disjointed writing styles from sections clearly contributed by separate co-authors, the lack of synthesis of the various dating techniques, the short and undeveloped discussion and the rather vague final age assignation all need serious work.

The *Homo floresiensis* (Hobbit) fossils found preserved in the limestone cave site of Liang Bua on the island of Flores in Indonesia are an example of how complex dating this type of site (fossils in a limestone/dolomite cave) and how long it takes to get good, robust data (see Morwood et al, 2004; Roberts et al., 2009 and Sutikna et al., 2016). There is scope for some comparison between this site and its multistranded chronology and the work presented here. Another example of using multiple strands of evidence to date cave deposits is the Marean et al (various years) work done at Pinnacle Points in South Africa. This is the kind of synthesis this paper is lacking.

So, as it stands this paper reads as a technical report of the different dating techniques, the type of document circulated between member of a large project to share information. Much work is needed to get this manuscript up to a standard ready for publication.

In detail:

1) There is no clear narrative. The paper reads like a technical report of the site, samples and results that would be circulated between co-authors – the data from the various methods is presented, often in excruciating detail (ESR) but there is no actual synthesis of the various data sets into a cohesive narrative.

2) The order of the paper is confusing and does not help in building a clear story – having the very detailed results section before the methods section does not make sense and makes reading the paper from beginning to end not easy. Some kind of discussion about what the different techniques used are all actually dating, which components of the cave system vs the fossils themselves etc would make a good introduction for the paper and segue into the more detailed methods section. Major rearrangement of the paper is needed, with a short clear methods section, then results.

3) Overall there are very few references to the literature – there is very little introduction to the Cradle and sites beyond repeated references to Dirks et al., 2015. The concept of the deposition of sediments in hominin caves being cyclic, with periods of erosion is by no means new and reference (and credit) needs to be given to Brain (1958, 1993, 1995) and Wilkinson (1983, 1985).

4) In the description of Unit 3, angular to sub-angular grains are classified as being reworked – on what basis? Such high angularity does not support reworking. Some micromorphology of these sediments (resin impregnated blocks then cut to make thin sections) would be very useful. How are the orange mudstones of Unit 1 and 3 differentiated from each other? It is also a little confusing and not entirely convincing to argue that Unit 2 is probably older than Unit 1 and was sourced from sediments that look like Unit 1 – there is very little actual evidence to back up this claim. A stratigraphic column summary diagram, plotted against age, would be useful.

5) The results section is very long and very detailed, all of the sections but especially the ESR/U-Th. The palaeomag results are the best presented and should be used as a model to redo the other sections. These different sections were clearly written by different people, as the long list of co-authors suggests, and need to be better synthesized together to produce a well-written document that flows well and is easier to read.

6) There is no discussion on the reproducibility of the various techniques – the authors are at pains to explain how the U-Th and ESR/U-Th analyses were both undertaken by two separate laboratories to produce blind duplications of the data, and yet there is no synthesis discussing this. Why was no blind duplicate done with the OSL samples?

7) There is no proper discussion. I realize there is a Berger et al submission alongside this manuscript which deals with the implications of the age proposed here, but there is still scope for a discussion of both the quality of the age data and what it all means in terms of the formation, development and filling of the Dinaledi Chamber, and of course the deposition and reworking of the hominin fossils. Given the long and illustrious list of co-authors this is surprising – among these people are world leaders in cave formation, dating and climate comparisons – suggesting that there has been little actual input from these authors in the discussion as it stands.

8) The big U-Th flowstone data set opens up the possibility for interesting comparisons with both local sites with U-Th dated flowstones (Gladysvale, Sterkfontein, Plovers Lake, Swartkrans); episodes of flowstone formation are strongly linked to climatic parameters (increased rainfall being the major one) and are expressed as similar aged flowstone deposits preserved in different caves. This comparison needs to be made here and would help build an argument as the when the deposition of the fossils took place.

9) There are also regional and more global records of known climate variability to which the data presented here can be compared, not just the U-Th flowstone data, but the OSL sediments data too. Seeing how the ages of flowstone and sediment deposition in the Dinaledi Chamber compare to other records is not only interesting but opens up the possibility to argue for glacial/interglacial deposition of the fossils and a more thorough picture of the changing landscapes outside the cave.

10) Following on from all these points about the lack of synthesis and discussion, as it stands the abstract is weak and once the paper is revised will need to be re-written into something more punchy. If some of the ages (OSL) are quoted directly in the abstract, then they all need to be. The abstract can also be a sentence of two longer and say something about the significance of these data – the reproducibility, the dating of difference aspects of the cave deposits and how these all fit together to produce a final age for the fossils.

Technique specific comments:

*1) OSL*a) The OSL section is weak. I understand a program was used to calculate the ages from the measured data but there is no discussion of these data relative to the literature as to which set of ages are more likely. OSL of cave sites is not easy, sites like Pinnacle Point in South Africa require significant method development, lead by Zenobia Jacobs, in order to date the sediments. It was certainly not as simple as crunching a single set of measurements through a program (written by someone else?). Some more detail as to why the Minimum Age model is most appropriate is needed.

b) Where does the maximum age for Unit 3 of ~400 ka come from? Is this a kind of midpoint age from the scatter (300-660 ka) of CAM ages? What evidence is there to suggest that this is the best approach?

c) To the best of my knowledge, this is the first use of OSL on Cradle cave sediments – this alone is worthy of a mention, as well as a caution that much work is still needed to thoroughly understand the OSL signal preserved in these grains.

d) Why was no blind test done for the OSL? Especially as these ages are heavily used to assign final ages to fossils.

e) Why do aliquots and not single grains? Especially in this complex cave setting where there has been much reworking.

f) Why not date the feldspar as well?

g) There is a very big difference between the two sets of ages calculated with no real discussion as to why this is.

*2) U-Th*a) Why were no thin sections made? For samples as important as these, every precaution to make sure the U-series system has not been disturbed should be taken. RS8 looks recrystalised?

b) Following on from this, some kind of pre-screening, laser ablation trace element profiles ideally, would have been very useful for imaging the location and concentration of U and Th.

c) There is no actual comparison between blind tests – why are JCU errors so much smaller?

d) Why are U ratios 234/238 and 230/232 ratios from JCU only reported in Table 1 as explained in caption – actual ratios are not shown in table. These data must be available from UoM and are available in Table 3. So what does this mean?

e) Methods and introduction section is very light, while Hellstrom and Pickering (2015) is a recent paper, this is a review paper and there needs some reference to original papers.

f) It is more accurate to say (page 17, line 533) that the later stage of flowstone formation was dated to 478,000 +107,000/-41,000 ka than give the full range of the error. That is not actually what the errors mean.

g) Why are the U-Th ages given as kyr before 1950 AD? They are not C-14 ages. This is not conventional with U-Th.

*3) ESR/U-Th*a) This section is heavy and hard going, even for a specialist reader. It needs to be shorter, clearer and have a better introduction – even just clearly explaining what ESR is measuring and what the advantage is to combining ESR with U-series.

b) The final ages are presented rather confusingly – again there needs to be some final synthesis: how old does the ESR suggest the fossils are?

*Reviewer #3 Minor Comments:*

1) The University of Melbourne is abbreviated (by itself) to UoM, not Melbourne University (MU).

2) A summary table of all the ages (not age data) will help pull everything together and allow for a comparison both between the blind duplicates of the same methods and between the different methods.

3) The text on a number of the figures (1, 3 and 10) is very small in places and will be tiny in the final typeset paper.

4) Figure 1 has the words 'Relative height (m), cave floor' twice.

5) It is useful to see all the field photos of the U-Th flowstone samples but these all need proper scale bars. So does panel C of Figure 3 and all of Figure 4.

6) Figure 8's caption is rather confusing, a little bit more introduction.

7) There appears to be some confusion over the caption for Figure 9 total of three captions are given, two different ones for Figure 9 and the caption from Figure 10 as well.

*Reviewer #4:*

This paper may be the definition of "throwing the kitchen sink" at something. I appreciate the exhaustive analyses of all techniques at their disposal, although I do have a few concerns regarding their interpretations. I'm glad there are finally some dates out for the chamber now so that speculation is no longer needed. I don't have any major issues with the general ages proposed, but I'm also not an ESR expert, which ultimately is what the *H. naledi* ages are drawn from. The results seem reasonable, although I do have some questions/issues regarding the interpretations. Descriptive components are done rather well and I appreciate the very detailed materials and methods section (even if it is obvious that each section has a different "voice" to it). My more detailed commentary is noted below, but I do believe there are some more major issues that need to be addressed, even if most have to do with dates not associated with the hominin-bearing Unit 3.

1) They need to provide more sufficient evidence for the "time-transgressive" nature of Unit 1. All Unit 1 ages are younger than Unit 2. Proposing that Unit 2 (and 3) is composed of reworked Unit 1 is a sedimentological hypothesis (not a geochronological result) that needs to be tested.

I think they also need to be more clear on the reworking of Unit 1 forming Units 2 and 3 and reworking of Unit 2 forming Unit 3. Is this supposed to be happening within the Dinaledi Chamber? Outside of the chamber then being transported in via the debris cone?

2) In the original Dirks et al. (2015) paper, they note that their flowstone samples were unsuitable for dating, yet now they have all sorts of dates on the flowstones. What changed?

3) The reassignment of material from under Flowstones 1c-e from Unit 2 to Unit 3 is one of the most important components of this paper. Without it, the majority of their age/depositional models fall apart. This needs to be addressed much more thoroughly, perhaps with additional sedimentological analyses. Simply that it is less indurated is not sufficiently convincing.

Also need some sort of explanation as why the description of the fossil from Flowstone 1A (Unit 2) went from hominin (Dirks et al., 2015), to not. It seems like interpretations are being changed to fit the dates as opposed to being based on the primary evidence/material (sedimentology, fossils) themselves.

4) Several critical errors in Table 1 (or the text) need to be fixed (see details below). The most important one is that sample RS18 is said to overly Unit 3 (hominin-bearing) in the text, but Unit 2 in the Table. This is a key date they use for anchoring the hominin fossils. Wondering if the change wasn't made in the table because they altered their unit assignments only after the dates were produced.

5) Table 2–Table 7 need substantial revisions. I know the work was done in different labs, but that doesn't mean the tables should be completely different. They need to be totally reformatted to match each other so that comparisons can be made between the two labs. As is, they have different fields, different units, etc.

6) The final age determination of the hominin fossils of 200-300 ka is not sufficiently explained. If based on ESR dates, then they need to more fully explain/justify why settled on 200-300 ka specifically as opposed to the ESR range of ~150-425 Ma? This appears to favor results of UW1810 and results from SCU/UOW (vs. GU/ANU). This seems rather arbitrary when in fact it is something that could be either modeled or calculated statistically. If it is just based of the younger age peak of 1767 (~200ka) and the older peaks of 1788 and 1810 (~275ka) in Figure 9, they should specifically state (and reference Figure 9).

Also, the yellow box in Figure 13 is not 200-300 ka, but more like 200-340 ka. Need to be consistent between text and figures. This is one of their main conclusions, yet not specifically supported.

*Reviewer #4 Minor Comments:*

- Line 88: Other papers format as refer to U.W. 101, not UW101. Change for consistency.

- Line 90: Lesedi Chamber papers says "some of which (facies and stratigraphic units) correlate to facies in the Dinaledi Chamber". Does not imply "sedimentologically distinct" as stated here.

- Lines 83 and 92: COH or CoH. Pick one.

- Line 167+: What specifically is used to correlate/group these discontinuous flowstone units across the chamber. You mention appearance is included (line 153), but no appearance is described. It seems that Group 2 is really just a group because it is not Group 1 or 3.

- Line 194: Should be noted that in Dirks et al. (2015) Figure 3 the fossil below Flowstone 1a is attributed to hominin. What changed the identification?

- Line 200: How can Unit 2 sediments be derived both from local erosion/reworking of Unit 1 *and* as a debris cone below a vertical fracture system? They can't be both autochtonous and allochthnous in relation to the chamber. I believe this needs to be clarified. Are the Unit 1 clasts derived from Unit 1 equivalents outside the chamber?

- Line 213/218. This also goes for Unit 3. Is the Unit 3 material derived from reworked Unit 1 and 2 material outside the chamber, and brought in via the debris fracture/debris cone? Please clarify.

- Line 202: Why is there no mention of the baboon tooth in Unit 2 here?

- Lines 203-260: This description doesn't seem to jive with what is depicted in Figure 7. The top cartoon should be more of a debris cone without a thick and even deposit running along the floor of the cave. Only AFTER the debris cone collapsed/eroded would Unit 2 material have spread/deposited along (evenly?) across the floor of the cavern. As currently depicted, if the debris cone in the top cartoon eroded, then the thickness of Unit 2 should be much, much thicker (not thinner) or you have to explain where all that collapse/eroded material went in the cavern (down the drains?).

- Line 211: It would be helpful to rephrase this sentence a bit to note how Unit 3 is differentiated from Unit 2. As is, they sound the exact same except for lithified (Unit 2) vs unlithified (Unit 3). Is that the only difference? Is that sufficient to re-assign the material below FS1c from Unit 2 to 3?

- Line 218: Along with Figure 7, is the interpretation then that the hominin remains were brought in along with the sediments from outside the chamber and deposited as part of the debris cone? Please clarify.

- Line 246: With other macro-fossils now identified in Unit 2, what are the implications for chamber accessibility in the past? (Especially as there is little Unit 2 exposed, but it has fossils.)

- Line 260-263. Need to provide some supporting evidence that Unit 1 was deposited during/after Unit 2 through modern times. Similarly, need to provide evidence that Unit 2 sediments are older than most of the Unit 1 deposits presently observed (no dating results presented at this point). What precludes it from being older? The only evidence provided (reworked into Unit 2) is the only concrete evidence provide for the sequence of the two units (and indicates older).

- Line 294: Should include or refer to a Figure 2 photo(?) showing FS2 directly overlying U3 and hominin fossils.

- Line 297: "fossils entered" suggests they walked in along with the deposition as opposed to being deposited/reworked along with Unit 3. Perhaps better to say they were deposited or transported.

- Line 317: Dirks et al. (2015) noted that flowstone samples included a detrital component derived from associated muds in all tested pilot samples which confounded U-Pb dating because of the high and isotopically variable background of common Pb it carries. But now they are apparently all suitable? Need an explanation as to why they were undatable in the last publication, but are now magically all datable.

- Line 332: Text says RS5 sample is from FS1, Table 1 says FS2. Which is it?

- Line 337+: Text uses Units, Table 1 uses Facies and they are *not* comparable! JCU/RS18 overlies Unit 3 (text) or Facies 2b (table)? This perhaps the most critical date as they argue it as the minimum for the fossils, but there is an error either in the text or the table.

- Line 368: Should state what the teeth are (e.g., upper premolar of *H. naledi*) either here or where each sample is discussed shortly later. Not just refer to figures that don't contain the information either (e.g., Figure 5, Figure 8). Shouldn't have to wait until materials section at the end to get some basic information.

- Line 375: Why are Table 6 presented before Table 4 (line 444)? Need to reorder.

- Line 379: Needs a brief explanation/justification as to why U-Th ages should be regarded as minimum.

- Line 426: There is no data on Sample 1841 in Table 7.

- Lines 427-435. Why are Sample 1767 and 1810 (and the age of Naledi in relation to it) being discussed under Sample 1841? This should have a different header.

- Line 493: Should provide radial plots for the samples for evaluation/visualization of the data.

- Lines 507-511: Even if one agrees with the time-transgressive argument of Unit 1 (and that's a big if), these are the ages you have at the moment. Similarly, they still need to be reconciled with the flowstone ages that cover them, which is brushed aside here.

- Line 572/573: This totally ignores the results of RS23 from the BASE of FS1a which has a date of 683-449 Ka and is not compatible with the pmag of older than 780ka. The authors focus on the top sample RS22 as it fits their pmag, but seem to selectively ignore the data that doesn't fit. This was not discussed in the paleomag section and it is totally omitted here. If RS23 is not from the base, or is from higher up and in the Normal section of FS1a, then that needs to be properly documented here. In Figure 13 they state RS23 as "mid", but in Table 1 it is "base" and in the description of the sample (Line 912) and Figure 2 it is at the base.

- Lines 568-595: It seems like the determination to reallocate the sediments associated to Flowstones 1c-e from Unit 2 to Unit 2 was based more on the geochronology results than sedimentological analysis. Different levels of induration could be the result of other processes besides being different depositional units. It seems like they are changing their interpretation post-hoc to fit their new ages rather than based on the actual sedimentology.

- Lines 604-607: Their argument for Unit 1 to be time-transgressive is not very convincing. All 3 of their samples from 2 different parts of the cave say Unit 1 is younger than Unit 2. It doesn't match their interpretation of Unit 2 being reworked material Unit 1 so they propose that it is time-transgressive, yet they provide no data to support this. Seems that this hypothesis could be tested by additional sampling to find pieces of reworked Unit 1 in Unit 2 (or 3) that are suitable for OSL analysis. If dramatically older, then I'd believe their conclusion. If not, then they need to seriously re-evaluate their current ages of either Unit 1 or Unit 2.

- Line 640/641: Again, Table 1 says RS18 overlies Facies 2b (=Unit2?), *not* Unit 3. Perhaps it was Unit 2 until they decided to change it Unit 3 because of the age (i.e., changing based on age, not sedimentology).

- Lines 644-647: This is confusing sediment age with depositional age. Reworking sediment doesn't change the age of the sediment, it simply reflects a different period of deposition. It would be more appropriate to say that Unit 3 is a reworked unit and that different deposits of it in the chamber may have been deposited at different times. However, it is also worth noting that although *H. naledi* may have been deposited over an extended period of time, they also may not have. There is not sufficient evidence presented to say one way or the other.

- Line 657: This is cherry picking the U-Th tooth data as it is only 1 (the oldest) of 14 results. Should just stick to the ESR data.

- Line 719: Why 200-300 ka? If based on ESR dates, then need to explain/justify why settled on 200-300 ka specifically as opposed to the ESR range of ~150-425 Ma? This appears to favor results of UW1810 and results from SCU/UOW (vs. GU/ANU). This seems rather arbitrary when in fact it is something that could be either modeled or calculated statistically. If it is just based off the younger age peak of 1767 (~200ka) and the older peaks of 1788 and 1810 (~275ka) in Figure 9. Then should specifically state (and reference Figure 9).

Also, the yellow box in Figure 13 is not 200-300 ka, but more like 200-340 ka. Need to be consistent between text and figures. This is one of their main conclusions, yet not specifically supported.

- Lines 729+: Appreciate the detailed materials and methods.

- Figure 1. Why not use the update surface map from the Elliot et al. submission that also includes the location of domal stromatolites, chert markers and mylonite markers? Should at least use consistent maps between papers.

- Figure 12'm assuming that this wasn't actually analyzed as Phase A and B are only a cm thick yet samples were 2-2.5cm cores.

- Table 1: It would be helpful to sort the table by flowstone group (or possible age), not HW sample number to compare proposed clusters of flowstone ages easier. Blind duplicates of the same sample or base/top of the same flow shouldn't be rows apart. Also, text only refers to Units, table refers to Facies, and they are *not* comparable! Why are RS10 and RS5 listed if there is not data?

Should delete (similar to no HW to pair with RS1).

Text says RS5 sample is from FS1, Table 1 says FS2. Which is it?

Text says RS18 overlies Unit 3, Table says RS18 overlies Facies 2b (=Unit2?). Which is it?

- Table 2: Need to be formatted exactly the same (same fields, units, etc.) for a proper comparison.

Data for RS1 is missing from Table 3. As it is a blind duplicate it *needs* to be included so the blind comparison can be compared.

Sorry, but the different formatting of Figure 2–Figure 7 is sloppy.

- Table 4 & 5. Again, why are these tables formatted differently and with different information.

Also, why are they presented before Table 5, but discussed/referenced after them in the text?

- Table 6 & 7. Again, why are these tables formatted differently and with different information?

- Table 6 is 4 tables, not 1. What do * represent? Needs to be explained.

- Table 7: Missing data for Sample 1841.

*Reviewer #5:*

At first reading, this manuscript raises more questions than answers. *H. naledi* unquestionably has primitive characteristics and originally was expected to date to at least 1 Ma. Dating that places these specimens at 200-300 ka suggests several scenarios.

1) The dating is correct. *H. naledi* originated at some unknown time, perhaps 1-3 Ma, and survived until the Middle Pleistocene, in a similar fashion to the proposed survival of *H. floriensis*. Note that *H. floriensis* was isolated from other species; *H. naledi* would have co-existed with possibly several other species of hominins. In itself, that is plausible; we are learning that the existence of a single hominin species is quite recent. However, this hypothesis conflicts with the evolutionary assumption that more primitive species cannot compete with more advanced, and thus to be convincing it would need to be bolstered by some evidence of an advantage that compensated for primitive features or an analysis that these features were irrelevant.

2) The dating is correct but the fossils are reworked from some other deposit. This hypothesis has been rejected on the grounds that there is no known entrance into the cave other than the one found by this group. It is further challenged by the sheer number of finds; survival of so many pieces is more plausible over 300 ka than 1 Ma.

3) The dating has been done with great precision and care, but there is a factor of which the analysts were unaware, or perhaps that could not be measured.

Reviewing a 96-page manuscript means that the reviewer must concentrate on a portion of it. Of the various dating methods used, only the combined ESR-U series method deals specifically with the fossil ages. It seems reasonable therefore to focus on the details of this method in commenting on the manuscript.

Let us consider the scenarios in reverse order. By definition, something that cannot be measured cannot play a role in calculations. Of the known factors, the only possible problem with the dating lies in the evaluation of the external dose rate. The cave has experienced significant transformations over time. The external dose rate is as much as 80-90% of the total dose rate for the hominin teeth. The text notes that sediment radioisotope contents are in Table 8; that table lists sediments from Unit 1 for OSL, not Unit 3. A reference is made to 'average' sediment values; it would be of interest to say how many samples were used. Also, the values for Unit 2 are very similar to those for Unit 3, reinforcing the suggestion (p. 8, line 213) that Unit 3 reflects reworking of Unit 2. On the whole, there seems little leeway for significant changes in external dose rate.

What about reworking? To say that 'there is no evidence that the present entrance has significantly changed' is not the same as saying that there is evidence it could not possible have changed, the latter being an unreasonable standard, of course. Internal reworking – where the fossils were originally deposited in a thicker Unit 2 and eroded onto a deflational surface in what is now Unit 3 – would not be likely to change the age, at least not from 200 ka to 1 Ma, since as noted in the preceding paragraph the enclosing sediment remains essentially the same.

That leaves the first scenario, which is one that is not dealt with in this manuscript. It will be interesting to see what sorts of interpretation are made along that line.

Therefore this manuscript needs only minor revision. The authors should explain more fully the number and locations of sediment samples, preferably adding yet one more table instead of referring to Table 8 (p. 38, line 1203).

*Reviewer #5 Minor Comments:*

Page 15, line 447: 'did' should be 'was'. Page 41, line 1296: 'enable one to differentiate'.

As noted in the review, the authors have not submitted sediment data for ESR dating that indicates how many sediment samples were used for each sample and where they came from.

[Further revisions were requested before acceptance.]

Thank you for resubmitting your work entitled "The age of *Homo naledi* and associated sediments in the Rising Star Cave, South Africa" for further consideration at eLife. Your revised article has been evaluated by a Reviewing Editor and five reviewers.

The reviewers were pleased with the improvements made to the manuscript, but several major remaining concerns were still noted. Many of the major concerns that need to be addressed are technical in nature, so they are provided in full detail below as provided by the reviewers in their comments, rather than summarized.

In addition, there was discussion among the reviewers during the consultation process concerning the last line of the Abstract and the penultimate paragraph of the Discussion (that begins on line 1009), which were viewed as unnecessarily critical of prior authors' work and of the challenges in accurately dating hominin fossils. In addition, at the end of the Abstract, it may be better to simply state that "These dating results indicate a much younger age for the *H. naledi* fossils than have previously been hypothesized based on their morphology."

Technical Points:

1) Lines 175-178: The addition is appreciated, but a sentence at least on how the material "accumulated inside fractures and along ledges" needs to be added. I would also drop the last sentence in this paragraph and move it to the discussion of Unit 1 (out of place here).

2) As a result of the revised units and stratigraphy (esp. "facies 1c" and "informal upper and lower members of Unit 3), the authors have now introduced inappropriate and inconsistent terminology. This is not just an issue of semantics.

Lines 217-229: The introduction of facies 1c is problematic, not as a subunit in general, but as a subfacies in particular. The confusion and misuse of facies happens often and here is a chance to correct it. Quoting Maill (facies guru), in sedimentology, "lithofacies are defined on the basis of its distinctive lithologic features, including composition, grain size, bedding characteristics, and sedimentary structures. Each lithofacies represents an individual depositional event. Lithofacies may be grouped into lithofacies associations or assemblages, which are characteristic of particular depositional environments". Thus, the basis of different subfacies here based on "relative topographic position in the cave" (i.e., facies 1c) is not valid criteria. Specifically, facies 1a is fine (non-lithified, horizontally laminated, orange mud). Facies 1b is fine (laminated mud with fine sand and rodent remains) and distinct from 1a. Facies 1c is "similar in appearance and composition to the laminated, muddy sediments of facies 1a, but they occur "in different places". This is not a properly differentiated as a separate facies. Facies 1a in a different location in the cave topography is still facies 1a unless it can be differentiated appropriately. There are many examples in sed/strat where the same/similar facies are repeated multiple/different times in a succession. This is again an issue at the end of the paragraph where the facies differ in "relative age and position within the cave chamber", which is not a characteristic of facies. Facies 1c needs to be differentiated from 1a on the basis of composition, grain size, bedding, or sedimentary structures to be valid.

Lines 270-276: Unit 1 is divided (incorrectly) into 3 subfacies. Unit 3 is divided into 2 informal members. This inconsistency is confusing and unnecessary. Also, members are formal lithostratigraphic units of formations. It is OK to provisionally have informal members, but this is almost always in the context of a sequence where there are other formal members (as part of a formation), which this sequence does not have, nor is it part of a defined formation. For example, you could have formal Members A, B, C, D and then informal members E and F, but randomly throwing in informal members into this context is exceptionally odd. Ironically, the authors could subdivide Unit 3 into facies 3a and 3b as the presence/absence of fossils is a characteristic that can be used to differentiate facies. Also, what happens if they do ultimately find *H. naledi* fossils in the "lower member"? Something that would easily fix all of this is if they just used informal sub-units for all (e.g., 1a, 1b, 1c, 2, 3a, 3b), that way they can be consistent and define them as they wish as units have no formal definition while facies and member do. The fact that all of the sedimentary material in the cave appears to multiple phases of reworking the same material over and over suggests everything is genetically related and easily dealt with as a series of related units.

Lines 304-315 (and elsewhere): Again, until other ages are generated for facies 1c subfacies or its stratigraphic relationship/contacts with other units can be observed/demonstrated, the time-transgressive explanation for Unit 1, and the relationship between Units 1 and 2 are just a hypothesis. Similarly, their assertion that facies 1c (as opposed to 1a or 1b) must have been the unit that provided the source material to Unit 2 (now 3?) is not confirmed by evidence and, again, is just a hypothesis. The authors are very definitive about their interpretation: "it is *clear* that older laminated orange mud deposits ascribed to facies 1c in Unit 1 occurred above Unit 2 and *must have* provided the source material now found as much clasts within Unit 2". As there are no dates for facies 1c, and as facies 1c is lithologically the same as 1a, this is a hypothesis, not an unquestionable fact. I have no problem if this is presented as a hypothesis: it might even be likely, but it is not definitive and I think the authors need to back away from such a claim. The authors note the challenges of OSL dating these deposits, yet rely almost exclusively on the OSL ages for a "young" facies 1a/b argument. An alternative explanation is that Unit 1a/b is older than Unit 2 and the OSL dates are incorrect (aliquots as opposed to single grain). Or perhaps the 1a looking material was brought into the chamber from elsewhere where similar depositional environments also produced laminated orange mud. This all needs to be demonstrated or proposed as a hypothesis (which can be tested by more/future dating even if far down the line).

3) Somewhere, the authors should to be more specific of the possible age of *H. naledi* from this chamber, versus the best fit age of *H. naledi* specimens dated here. For example, flowstone group 1c does provide a minimum age for the specimen that is stuck underneath it, and the ESR ages do provide a maximum age for the 2 teeth. The authors suggest that the fossils entered the cave over a period of time and may have continued during the deposition of upper Unit 3 (lines 299-303). As such, I have no problem with the authors saying those 3 particular fossils have a best fit of 236-335 ka, but does anything preclude the other hominid specimens from being associated with older or younger parts of upper unit 3 with the potential to span approximately 90ka (min of above RS1/15 above U3) to 414 ka (max of OSL5)? At minimum, a red dashed vertical line should be added to Figure 14 at ~414 ka for the oldest possible age to match the minimum line at ~100 ka.

4) Line 465: This sentence is confusing as presently constructed. It seems to report that something is both younger than unit 1 and older than unit 1. It would be better to write "younger than ages derived from facies 1a, but slightly older than ages derived from facies 1b in this location."

5) The U-Th ages must be quoted as just ka, using kyr BP is a radiocarbon convention and does not apply to U-Th - there is no industrial carbon adjustment to U-Th ages, so to quote them as kyr BP is misleading and incorrect.

6) In Table 4, "Dose" should be "De" to prevent confusion with subsequent discussion of artificial doses given to samples. Thicknesses are μm, not mm and according to the table, 100 μm was removed from the dentine and the sediment, not enamel and dentine, due to misplaced headers. The removal from dentine was not mentioned in the methods section.

7) There was no dentine on the baboon tooth (line 618) but would a modern analogue suggest that there should have been dentine on at least the inside? It probably doesn't matter since the dentine survival time would be a relatively small portion of the entire burial time, but the authors might comment on this.

8) Figure 8: With the addition of another unit, it is very challenging to differentiate the little sliver of Unit 2 (below Flowstone 1a) and Unit 3 upper because of the similarity in color, especially in the upper panel. It would be helpful to address this. Also, given the revisions, what now is the explanation/model for the deposition of Unit 2? Virtually all of Unit 2 is now assigned to lower Unit 3. In the previous submission, Unit 2 was under the flowstone and underneath the hominid fossils on the cave floor. Now it is stuck up under the uppermost flowstone and nowhere modeled to be anywhere on the cave floor, just completely washed away(?). While the authors do touch upon this in the text, they should consider adding back in a revised panel to explain Unit 2 a bit better.

[Further revisions were requested before acceptance.]

We appreciate that you have been extremely responsive to the various reviews, including your recent detailed comments. However, the reviewers collectively do not feel that the manuscript can be published without revisions that fully address the remaining outstanding concerns on the sedimentology. We also want to clarify that while many of the most detailed comments on these topics in previous rounds of review were from one reviewer who has consistently provided their expertise and extensive time to help improve the paper (Reviewer 4 from the original round of review), all of the reviewers have expressed agreement and consensus on these points during consultation sessions.

Perhaps one of the major continuing complications and challenges in these revisions is the effort to maintain consistency with the earlier descriptions in the Dirks et al. 2015 paper. Simply stated, the reviewers feel strongly that the use of facies as units in Dirks et al. 2015 was incorrect; updated stratigraphy needs to be used in the present paper alongside a simple, clear statement of change from the original interpretations. Editorially, we agree that the present paper should standalone and use the most up-to-date, appropriate descriptions and approach. Doing otherwise is introducing complication and would be perpetuating confusion.

---

## [Author Response]

In revising the original manuscript we have made some major changes and have re-written sections, mostly following the recommendations made by the reviewers. We also made some significant changes that were not mentioned by the reviewers, but that needed to be done with respect the ESR data.

For ESR we have now standardized the modelling input parameters for the ESR models. Originally the results provided by the two labs varied significantly, largely because each lab had made different à priori model assumptions (as a result of this being a double blind experiment), which would have been hard to spot by any non-specialist. To allow for more objective comparisons of the two different data sets, we have now based all ESR modelling on mutually agreed to in-put parameters for the environmental dose rates (presented in Table 4), and we have introduced two modelling scenarios; one resulting in maximum age estimates; the other in minimum age estimates with the true age somewhere in between.

We have explained this approach in detail in the revised manuscript. These changes also go a long way in addressing some of the concerns raised by the reviewers regarding consistent bracketing of age ranges for the fossils.

In response to the main editorial issues:

*1) Critical points concerning the relationships among the various stratigraphic units were raised by Reviewer 4 (in particular, see major comments 1 and 3 from that review) that must be addressed thoroughly in any revision. During the consultation process, other reviewers highlighted the importance of these points given the interdependence of most of the age/depositional models on those interpretations.*

We believe that this has now been addressed; see our detailed comments in response to reviewers 3 and 4 in particular. In brief:

A) We have introduced an extra facies (facies 1c) with Unit 1, to explain that components of Unit 1 occur in different topographic positions within the cave, and that some could have contributed to the sediment provenance of Unit 2.

B) We have removed Unit 2 as underlying Unit 3 from the floor sediments of the Dinaledi Chamber, and restricted Unit 2 outcrops to occur below flowstone remnants near the entrance (as originally defined by Dirks et al., 2015). Instead we have subdivided Unit 3 into a hominin-bearing upper Unit 3 member and a hominin-free lower Unit 3 member, with the upper and lower members occurring along the floor of the chamber and the upper member also occurring below flowstone remnants at the entrance into the chamber. The reason for changing this back to Dirks et al. (2015) is that all floor sediments are probably significantly younger than the Debris cone and are probably better grouped with Unit 3 than with Unit 2. Unit 2 near the entrance is >780 ka old (from paleomag), and started eroding after formation of Flowstone 1a came to an end, which could be as late as 437 ka (the lower age limit of Flowstone 1a). It is this erosional process of the debris cone, probably driven by sediment moving down the drains, which causes the material to spread out along the floor.

C) To illustrate these relationships more clearly we have added an extra figure (Figure 2), which shows cross sections through the cave. We have also completely redrawn Figure 14 to more clearly reflect these relative timing relationships as well.

D) We have expanded our discussion around how the ages help re-define the stratigraphic units.

*2) The overall synthesis/analysis of the various dating results is too limited at present, and choices on what results are highlighted at various points in the manuscript seem arbitrary at times. A more concerted and comprehensive effort to assimilate the different datasets in a evenly considered, summary results section (with a synthesis view agreed to by both dating labs) would help to address this concern. This issue also filters up to the abstract.*

This has been addressed in several ways:

A) We have extensively rewritten the discussion section, expanded it and introduced a section that looks specifically at the quality of results obtained by the different methods and how these results should be compared

B) We have introduced a new Figure 14 that more clearly summarizes how the age results should be interpreted and how they constrain the age of various geological units in the cave chamber

C) Part of the problem with the age results was that the ESR results from the two labs, which are central to the age estimates for the fossils, had not been standardized; i.e. the two labs were using different model assumptions in calculating their environmental dose rates, and in consequence, their ages. We have now standardized the modelling approach for ESR with fixed model parameters that have been varied between two scenarios resulting in minimum and maximum age estimates. As a result the paper is easier to read, and ages are now better and more consistently constrained.

D) We have defined the age constraints in a clear and objective manner (using the listed uncertainties), and we have applied these age estimates in a consistent manner throughout the paper.

*3) The manuscript currently reads as if various sections were composed by different authors with insufficient consistency in style and level of detail. Likewise, the tables need to be reworked with more care for consistency and interpretability.*

The core of the entire paper was written by one person (PD), with input and edits from experts for various sections. For the methodology sections we have had to rely extensively on input from participating labs, so some style differences in this part are inevitable, but even here we have tried to homogenise the ‘tone’, and we included introduction sections written by the same person (PD). Please note that this paper discusses 6 different dating techniques, provides detailed sedimentology of complex stratigraphy, talks about paleoanthropological and archaeological constraints and even includes some anatomy. All these fields have their own vocabulary and ways of describing things (e.g., provenance in sedimentology is referred to as provenience in archaeology; should we refer to the baboon tooth as baboon or *cf. Papio* etc.).

For a long, complex, multi-disciplinary paper like this, I am afraid that some style differences will be inevitable, however, like I said, we have tried to homogenize the style as best as possible.

*4) The reviewers also felt that the discussion should also be expanded, and can be even without overtly duplicating the broader synthesis provided in the other papers.*

This has now been done as explained under point (2) above.

*5) Similarly detailed treatment for reporting the radiocarbon methods and results, as the other dating methods, should be provided.*

We have now treated the radiocarbon dating in the same way as the other techniques; but we have not explained things to the same amount of detail, because we only did some preliminary testing with 14C, and then decided not to pursue things further. Although this was already explained in the paper, we have now expanded the dating strategy section to make this more explicit.

*6) The reviewers also had suggestions for the order of presentation that may help the reader more readily interpret the results.*

The reviewers asked for a methodology section up front. In our original submission we followed the format suggested by *eLife*; i.e., results are presented before the methodologies. After consultation with the journal editors we have now included a general section entitled “Dating the *H. naledi* fossils”, which explains the dating strategies and approaches taken, and acts as an easy to read introduction of methodologies. This section has now been placed directly before the results.

Responses to the reviewers’ comments follow:

Reviewer #1:

*I will state up front that the dating methods used in this paper are not within my field of expertise. I was asked to review this paper as one component of the entire suite of submitted papers. Thus, I will leave the methodological critiques to those more qualified than me and comment more on the implications of the dates. The authors suggest that deposition of the hominin material was over a time period of about 100k between 200-300 ka. It does seem like the resolution and precision of the dates are not great and makes it difficult to properly interpret the context of the hominins. When the duration of deposition is considered, it then seems unusual that there would be so few hominins in the cave system, particularly if it is being used as a burial site (even if only for special individuals).*

Point taken, but the dating techniques have their limitations. We have now clearly explained what the age range is for the fossils, and for the sediment unit that contains the fossils. This still allows for almost 100ka of time, which we cannot constrain much further, even though we have given indications of our preferred interpretation.

This paper is about geochronology. I think we have presented the data in an objective, detailed and transparent manner. How the age constraints should be interpreted with respect the time period over which *H. naledi* entered the cave, or its behavioural patterns is largely speculation and has been dealt with elsewhere (e.g., accompanying Berger et al., co-submission).

*Given the complexity of cave systems and their formation, I don't know that the authors have convincingly demonstrated that the chamber was completely inaccessible during the times of hominin deposition (especially with the wide range of dates and low resolution). That said, I sympathize with the inherent difficulties of dating fossils from cave systems and appreciate that the authors used multiple methods to try to pin down the dates and that this may be the best that can be done at this point.*

First, in terms of inaccessibility, we have performed surface geophysics, laser scanning of the chamber, and extensive caving surveys, all of which have yielded zero evidence for alternative entry routes into the Dinaledi Chamber. Additionally, we have shown that major sedimentological differences exist between the Dinaledi Chamber and other chambers in the cave system. Geochemical analysis shows a distinct difference between sediment of the Dragons Back Chamber and the Dinaledi Chamber. We appreciate that this is difficult to accept, but we can find no evidence to refute this claim.

Second, because we are dealing with an active sedimentary system, obtaining ages for material that is participating in the active processes of continuous reworking is exceedingly hard.

Reviewer #2:

*Although I have reservations about the way radiocarbon dating was attempted, as explained further below, and there is a need for more caution in some areas, the paper should be suitable for publication after revisions. I have marked up a text copy of the manuscript, which I attach, and summarise some of the points below. The references for all four papers need work for consistency of formatting, journal abbreviations, accents etc.*

Done, but only up to a point. Regarding references, the *eLife* website says:

"Please note, authors do not need to spend time formatting their references and can submit manuscripts formatted in a variety of reference styles, including Harvard, Vancouver, and Chicago. Wherever possible, please do not truncate the number of authors in the references list, but please do provide a DOI if possible." We have made sure that each reference has all of its information.

*"The results for RS18 (~242 ka) may provide a potential minimum age for the* H. naledi *fossils in this part of the cave." Insert some caution because some other age estimates are still younger than this figure.*

This is a valuable comment and issues are now discussed extensively in the discussion section.

*"Dates acquired via U-Th and ESR techniques were obtained using a double blind approach for each technique to ensure robust, reproducible results, with each laboratory using their own analytical and computational approach. Methodologies and approaches taken by each laboratory that has contributed to this paper are described in detail in the methodology section." Why wasn't radiocarbon dating pursued in the same manner, using established research labs rather than a commercial lab? Why no details of the procedures: e.g., was ultrafiltration attempted?*

As explained in the text carbon dating was done “as due diligence to obtain an indication if carbon dating should be further pursued”. This was most easily done by sending samples to a commercial lab to see if it works. Given these initial results and the destructive nature of the technique, carbon dating was not further pursued. We have clearly explained this in the text (and have now expanded on this explanation in the new version).

The initial results from the commercial lab indicated that carbon dating was not going to work. Subsequent U-Th and ESR work made it clear that carbon dating should not be further pursued, and attempting blind dates at research labs would be pointless as far as the purpose of this study is concerned, and we did not want to unnecessarily impose on people’s time and expenses. We are only reporting the carbon dates here for completeness sake, and the fact that it corroborates flowstone formation around 30ka.

However, we have added a more detailed methods section on 14C dating, and we have brought the description of the samples and technique in line with the other methods.

*"Some of the ESR ages (Table 4, Table 5) independently suggest that some of the fossils around the excavation pit may be younger than 237 ka, but confirm a probable age of >200 ka for the fossils." Well a few of the ESR ages* don’t *confirm an age of 200 ka+ so this needs rewording.*

Part of the problem with some of the dating results was that the ESR results from the two labs, which are central to the age estimates for the fossils, had not been standardized because of the double blind strategy employed; i.e. the two labs were using different model assumptions in calculating their ESR ages (i.e. their environmental dose rates). We have now identified the best input parameters and standardized the modelling approach for ESR with fixed model parameters that have been varied between two scenarios resulting in minimum and maximum age estimates. As a result the paper is easier to read, ages are now better and more consistently constrained, and it has been easier to explain how the age results place constraints on the ages of the sediments and fossils.

Reviewer #2 Minor Comments:

*Other suggested edits are added in an attached text version of the paper.*

Done.

Reviewer #3:

*[…] However, there are many issues, the three major ones being:*

*1) there is no synthesis or analysis of the various geochronological data sets;*

Although we did have quite an extensive analysis of the new stratigraphy in the cave and what the ages meant for the age of the fossils, we agree that more work was required to improve clarity and readability.

We have now expanded the discussion section significantly and have included an entire section on the quality of the age data and how the various age estimates should be understood and compared, which should go a long way in addressing this issue.

*2) there is no discussion of how the new ages fit into the regional and global picture in terms of cave formation;*

Although true, this is well outside the scope of this particular paper, and it would be inappropriate to discuss this here without more work. Indeed, we have two PhD students (both coauthors on this paper) who are conducting significant additional analyses and specifically focusing on the “global picture” of cave formation in the Rising Star Cave. We (they) have much work still to do before this comes into focus. Rather, this paper is about dating the *H. naledi* fossils. All methods have been designed with that aim in mind.

3) Exactly how the age of the fossils has been determined is not abundantly clear: this is certainly the most important message of the paper and needs to be clearly and carefully explained. The existing discussion on pages 20 and 21 is confusing and no clear story emerges.

This is a good point. We have now extensively rewritten the discussion section, and have included a more standardized approach to generating the ESR results. See further comments on this issue in our various responses below.

*The authors presume a huge prior knowledge of the region, caves and fossils, as well as the background and methods of the various dating techniques;*

Extensive background has been provided for each of the dating techniques, much more than is usual in dating papers of this kind. This has been done to make the techniques more accessible for a wider audience. To provide more, would transform the paper into a review paper, which is not the intent here.

However, we have included some more info on the regional geology as requested.

I am not unfamiliar with the sites in this region and these dating techniques but struggled to follow a clear story with the manuscript. The confusing order of the sections…

The sections follow the prescribed format of the journal in which results are presented before methods. We have tried to overcome some of this problem by including an abbreviated discussion of methods up front.

*…the disjointed writing styles from sections clearly contributed by separate co-authors, the lack of synthesis of the various dating techniques, the short and undeveloped discussion and the rather vague final age assignation all need serious work.*

We have expanded this and clarified the discussion as requested, however, we have also intentionally left out discussions related to paleoclimate and cave formation, which form the subject matter of two ongoing PhD student projects.

Nevertheless, the comment about a lack of synthesis is true, and we have completely rewritten the discussion section and expanded it with a section comparing the various techniques. We have also more clearly defined the age brackets, and standardized the ESR approach, and included a new summary figure and extra sections, all of which should make it much more explicit why certain age brackets were chosen.

*The* Homo floresiensis *(Hobbit) fossils found preserved in the limestone cave site of Liang Bua on the island of Flores in Indonesia are an example of how complex dating this type of site (fossils in a limestone/dolomite cave) and how long it takes to get good, robust data (see Morwood et al, 2004; Roberts et al., 2009 and Sutikna et al., 2016). There is scope for some comparison between this site and its multistranded chronology and the work presented here. Another example of using multiple strands of evidence to date cave deposits is the Marean et al (various years) work done at Pinnacle Points in South Africa. This is the kind of synthesis this paper is lacking.*

This is a valid point and we have included some new references and discussion with respect to Liang Bua and the difficulty of obtaining good results. We have also applied similar strategies to those used by Marean et al. and others for Pinnacle Points to summarise our various results (e.g., see Figure 14). However, we also wish to highlight that the caves mentioned by the reviewer are mature excavation sites with extensive excavation histories and multiple dating and stratigraphic studies. To date, only a single excavation pit is available for comparison in the Dinaledi Chamber and this paper presents the first attempt at dating this complex cave system. Simply put, we have no previous results or data to compare this to, but we do not doubt that this study will provide the framework for future work as more of the chamber is excavated and more focused studies are developed.

*So, as it stands this paper reads as a technical report of the different dating techniques, the type of document circulated between member of a large project to share information. Much work is needed to get this manuscript up to a standard ready for publication.*

*In detail: 1) There is no clear narrative. The paper reads like a technical report of the site, samples and results that would be circulated between co-authors – the data from the various methods is presented, often in excruciating detail (ESR) but there is no actual synthesis of the various data sets into a cohesive narrative.*

Here we have to cordially disagree, although we have certainly made efforts to make this clearer. Specifically, to address this concern, we have added a new, highly detailed figure (Figure 2), which we hope better explains the relationships between the geology in the cave and the dates presented here and we have modified Figure 14 to help synthesise the various results. We have expanded the discussion. We wish that there was a simple, narrative to tell, however the results are not all entirely straightforward and we have intentionally tried not to interpret the data.

*2) The order of the paper is confusing and does not help in building a clear story – having the very detailed results section before the methods section does not make sense and makes reading the paper from beginning to end not easy.*

This is the prescribed format of the paper, but we have now addressed this to some degree with permission from the editors.

*Some kind of discussion about what the different techniques used are all actually dating, which components of the cave system vs the fossils themselves etc would make a good introduction for the paper and segue into the more detailed methods section. Major rearrangement of the paper is needed, with a short clear methods section, then results.*

With agreement from the editors we have now done this. Immediately before the results section we have included a section entitled “Dating the *H. naledi* fossils’. In this section we provide a general background to the problems with dating fossils in the cradle, with sideways references to other fossil sites in caves as well (esp. the H. floresiensis site) as suggested by this reviewer. We have further extended this section, with a more detailed explanation of how the dating strategy was designed, why certain techniques (U-Th, ESR and palaeomagnetics) were pursued, whilst other techniques (14C and OSL) were not continued beyond preliminary studies.

*3) Overall there are very few references to the literature – there is very little introduction to the Cradle and sites beyond repeated references to Dirks et al., 2015. The concept of the deposition of sediments in hominin caves being cyclic, with periods of erosion is by no means new and reference (and credit) needs to be given to Brain (1958, 1993, 1995) and Wilkinson (1983, 1985).*

No disrespect was intended. Given that the original submission was already over 90 pages long, we did omit much of the background literature. We have now included and expanded introduction to the caves and fossil deposits in the Cradle with additional references as suggested by this reviewer.

*4) In the description of Unit 3, angular to sub-angular grains are classified as being reworked – on what basis?*

Please note that for the most part, we interpret the reworking to result from gravitational driven collapse and slumping processes in the cave due to wetting and drying cycles. This is distinct from reworking processes in alluvial settings that involve considerable transport in fluids that would indeed cause considerably greater textural maturity as the reviewer implies. This is a well-supported phenomenon in Karst sedimentology. As explained in the text, material rolls in from fractures and collapses from below flowstones 1b-e; material occurs as loosely packed clumps of clay, in places on top of flowstone 2 remnants. In addition we have described flowstone 2 skirtings along the side wall with the floor-level of Unit 3 dropping below that, we have described drains, cross-cutting Mn-oxide tide marks on bone, etc. This is now clearly and repeatedly described in the text and is what forms the basis for our interpretation of repeated reworking.

*Such high angularity does not support reworking.*

Again, this is a concept that is quite true in alluvial settings, but cannot be applied equally to karst systems. The fact that grains are angular does not mean they are not reworked, it means that transportation routes have been short and there has not been much physical abrasive activity during the transportation process, nor has there been significant involvement of water transport. The primary agent of reworking and sediment movement is mass-wasting (gravitational forces).

*Some micromorphology of these sediments (resin impregnated blocks then cut to make thin sections) would be very useful. How are the orange mudstones of Unit 1 and 3 differentiated from each other?*

This was all presented in Dirks et al. (2015), and this work is clearly cited throughout.

*It is also a little confusing and not entirely convincing to argue that Unit 2 is probably older than Unit 1 and was sourced from sediments that look like Unit 1 – there is very little actual evidence to back up this claim.*

Not at all; the sediments of Unit 1 are near-identical in appearance and geochemistry to sediment blocks of Unit 1 inside Unit 2 (as presented in Dirks et al. (2015)), with the one big difference that Unit 2 and Unit 3 sediments have been oxidized and affected by Fe-Mn oxy-hydroxide infiltration. Again, all of this is well cited in Dirks et al. (2015); hence, we have not repeated this here.

*A stratigraphic column summary diagram, plotted against age, would be useful.*

This is an extremely helpful comment and we have added three new cross-sections to better illustrate the stratigraphic relationships of the units and dates presented. We have also updated our summary diagram (Figure 14) to make the various relationships clearer. We have also discussed these complex relationships in greater detail in the text and figure captions.

Based on this comment, we have also decided to introduce a further facies in Unit 1 (i.e. facies 1c) which is similar in appearance (laminated mudstone) to facies 1a (from Dirks et al., 2015), but which forms on ledges and in cracks away from the cave floor. In contrast facies 1a forms on the cave floor. All this has now been better (we hope) explained in the text.

5) The results section is very long and very detailed, all of the sections but especially the ESR/U-Th. The palaeomag results are the best presented and should be used as a model to redo the other sections. These different sections were clearly written by different people, as the long list of co-authors suggests, and need to be better synthesized together to produce a well-written document that flows well and is easier to read.

We have worked hard to make these sections more readable. However, the results sections were all written by the same person (PD) with input from individual contributors. All contributors also edited and approved the written sections. I disagree that there is much of a difference in writing style across the results sections; we are dealing with a range of very different techniques that require a different approach when being explained. Most of the results section is simply a summary of the tabulated results highlighting the main points.

Where differences do occur is in the methodology section, because these are based on the lab and methodology descriptions provided by each of the individual labs. The technical details are lab specific and therefore not easily ‘homoginized’. To overcome some of this issue and make the principles of the dating techniques more accessible to a general reader, we added a general description of each technique at the start of each methodology dating section, These were all written by the same person (PD) and edited and approved by the individual labs.

*6) There is no discussion on the reproducibility of the various techniques – the authors are at pains to explain how the U-Th and ESR/U-Th analyses were both undertaken by two separate laboratories to produce blind duplications of the data, and yet there is no synthesis discussing this. Why was no blind duplicate done with the OSL samples?*

This is a valid and useful point. A detailed synthesis of this has now been included in the discussion section under the heading “Reliability of the age estimates”.

*7) There is no proper discussion. I realize there is a Berger et al submission alongside this manuscript which deals with the implications of the age proposed here, but there is still scope for a discussion of both the quality of the age data and what it all means in terms of the formation, development and filling of the Dinaledi Chamber, and of course the deposition and reworking of the hominin fossils.*

Some of this was actually discussed at length (we included an entire section on discussing the revised stratigraphy based on the new ages including the deposition and reworking of the fossils). Nevertheless, we have expanded the discussion, especially by focusing more on the quality of the data presented as recommended by the reviewer, and we have added a summary of what the ages mean for hominin evolution.

*Given the long and illustrious list of co-authors this is surprising – among these people are world leaders in cave formation, dating and climate comparisons – suggesting that there has been little actual input from these authors in the discussion as it stands.*

Indeed we have been very lucky to have the help of many World-class collaborators on this project. However, the suggestion that there has been little actual input from these authors is certainly not true. Some have contributed more than others, however the main point that the reviewer makes about dating and climate comparisons is well and truly outside of the focus of this paper. Climate and cave formation studies will come, however the focus of the manuscript is solely intended to be around the age for *H. naledi.*

*8) The big U-Th flowstone data set opens up the possibility for interesting comparisons with both local sites with U-Th dated flowstones (Gladysvale, Sterkfontein, Plovers Lake, Swartkrans); episodes of flowstone formation are strongly linked to climatic parameters (increased rainfall being the major one) and are expressed as similar aged flowstone deposits preserved in different caves. This comparison needs to be made here and would help build an argument as the when the deposition of the fossils took place.*

This is indeed very interesting, but again it is very much outside the scope of this paper. As one reviewer points out, we effectively throw the kitchen sink at dating the fossils and this is the sole focus of the study. Comparisons of flowstone formations with other caves are the focus of a number of ongoing student and coauthor projects. We hope that the reviewers and editors will understand that these results will follow in time, but that they require a great deal of additional work still.

*9) There are also regional and more global records of known climate variability to which the data presented here can be compared, not just the U-Th flowstone data, but the OSL sediments data too. Seeing how the ages of flowstone and sediment deposition in the Dinaledi Chamber compare to other records is not only interesting but opens up the possibility to argue for glacial/interglacial deposition of the fossils and a more thorough picture of the changing landscapes outside the cave.*

We know and agree, but this is well outside the scope of this paper as indicated above. In particular the OSL results are exciting, but they necessarily are going to require a great deal of focused work in the cave to fully understand and interpret.

*10) Following on from all these points about the lack of synthesis and discussion, as it stands the abstract is weak and once the paper is revised will need to be re-written into something more punchy. If some of the ages (OSL) are quoted directly in the abstract, then they all need to be. The abstract can also be a sentence of two longer and say something about the significance of these data - the reproducibility, the dating of difference aspects of the cave deposits and how these all fit together to produce a final age for the fossils.*

Thank you. We agree that the abstract would benefit from a bit more punch. The abstract has now been changed to reflect some of this. However, the paper simply contains too much information, and we are not able to provide all of it in the abstract. So we have tried to keep the core message in the abstract simple, especially because many of the readers of this paper will not be experts.

*Technique specific comments:*

*1) OSL a) The OSL section is weak. I understand a program was used to calculate the ages from the measured data but there is no discussion of these data relative to the literature as to which set of ages are more likely. OSL of cave sites is not easy, sites like Pinnacle Point in South Africa require significant method development, lead by Zenobia Jacobs, in order to date the sediments. It was certainly not as simple as crunching a single set of measurements through a program (written by someone else?). Some more detail as to why the Minimum Age model is most appropriate is needed.*

We have expanded the explanations in this regard, both in the data section and in the discussion section. A full OSL study of the cave system is being planned.

*b) Where does the maximum age for Unit 3 of ~400 ka come from? Is this a kind of midpoint age from the scatter (300-660 ka) of CAM ages? What evidence is there to suggest that this is the best approach?*

We have expanded the explanations in this regard, both in the data section and in the discussion section.

*c) To the best of my knowledge, this is the first use of OSL on Cradle cave sediments – this alone is worthy of a mention, as well as a caution that much work is still needed to thoroughly understand the OSL signal preserved in these grains.*

Good points; we have expanded discussion on the OSL somewhat to reflect the preliminary nature of the work, and the challenges with obtaining good dates by OSL. Note however, that the OSL dates are of modest value here to defining the age of *H. naledi* because the absolute relationship between Unit 1 and 3 is not yet clear, which is why an in depth study was not pursued beyond the initial work (this has now been explained in the new methodology section preceding presentation of the results). We have shown that the technique holds promise, but future work will require single grain analysis and analyses of feldspars and this will require considerable work in future.

Note also that we did do trials of single grain analyses and feldspar analyses with U. Wollongong (Zenobia Jacobs and Bert Roberts), but due to the complexity of the system and the fact that the initial results did not contribute significantly towards gaining a deeper understanding of the age of the fossils, we have limited our data presentation to reporting the initial results from Wits, following the advice of Jacobs and Roberts.

*d) Why was no blind test done for the OSL? Especially as these ages are heavily used to assign final ages to fossils.*

See above; note that these ages are not ‘heavily used’ to assign ages to the fossils; only ESR and U- Th are significant in that regard. However, the OSL ages do provide insight in the age of Unit 1; confirm the general stratigraphic models proposed; and are consistent with the general age range during which the deposits, including the fossils, accumulated. For those reasons we thought it appropriate to include the results, with ample caution.

Because the OSL ages are not vital in determining the age of the fossils, there was no need to do a double blind test. Note that few studies on fossils in the CoH report double blind dates. Double blinds are time consuming, hard to organise and difficult to coordinate. Most labs are very busy and it would not be correct to unnecessarily impose on people’s time and resources to conduct the work unless absolutely necessary.

*e) Why do aliquots and not single grains? Especially in this complex cave setting where there has been much reworking.*

See above. Because we made use of the University of the Witwatersrand lab, which only has the ability to date aliquots. Given that this project is coordinated through Wits, it was important to use that lab. Also, preliminary single grain analyses were undertaken at the University of Wollongong as discussed above, but this work will require a fully focused (and funded) project in future.

*f) Why not date the feldspar as well?*

As above.

g) There is a very big difference between the two sets of ages calculated with no real discussion as to why this is.

This has been corrected; a mix up with sample numbers occurred in the lab, leading to a wrong (much younger) age being reported for OSL 5 in the original manuscript. This has been corrected and more discussion has been added. This was a simple, but confounding mistake that has happily been resolved.

*2) U-Th a) Why were no thin sections made? For samples as important as these, every precaution to make sure the U-series system has not been disturbed should be taken. RS8 looks recrystalised?*

All the detailed descriptions of the U-Th samples in the paper are based on thin-section work and extensively discuss recrystallization textures, sample by sample. The reviewer may have missed this section?

*b) Following on from this, some kind of pre-screening, laser ablation trace element profiles ideally, would have been very useful for imaging the location and concentration of U and Th.*

This is not as critical for U-Th as it would have been for U-Pb (the reviewer appears to have U-Pb in mind in making this comment.). As explained, we did do preliminary test work to determine whether to apply U-Pb vs. U-Th dating. In the end it was clear that U-Th was the best option and we obtained excellent results, reproducible by two independent labs.

*c) There is no actual comparison between blind tests – why are JCU errors so much smaller?*

The JCU ages are not that much smaller; rather, they are very similar and for a number of flowstones they are bigger (see Table 1 for the direct comparison). Note that we have re-edited the tables so that they all appear in a similar format, which makes comparison easier.

*d) Why are U ratios 234/238 and 230/232 ratios from JCU only reported in Table 1 as explained in caption – actual ratios are not shown in table. These data must be available from UoM and are available in Table 3. So what does this mean?*

Table 1 is a summary table that compares results from JCU and UM. Table 2 presents the actual data from JCU (including the ratios the reviewer refers to) Table 3 presents the actual data from UM. Note that we have re-edited the tables so that they all appear in a similar format, which makes comparison easier.

*e) Methods and introduction section is very light, while Hellstrom and Pickering (2015) is a recent paper, this is a review paper and there needs some reference to original papers.*

See our explanation under (4), why we have chosen for a relatively light introduction to the methods section. Note that our descriptions of methods are significantly longer than in any paper that deals with dating fossils in the CoH, which is not specifically focused on method development. We made choices as to how long and how detailed we should make the descriptions. Some more literature references have been added.

*f) It is more accurate to say (page 17, line 533) that the later stage of flowstone formation was dated to 478,000 +107,000/-41,000 ka than give the full range of the error. That is not actually what the errors mean.*

Okay.

*g) Why are the U-Th ages given as kyr before 1950 AD? They are not C-14 ages. This is not conventional with U-Th.*

In Archaeology it is not uncommon to quote both U-Th and C ages in kyr before 1950 AD to facilitate easy comparison. We have chosen to follow this approach given that both techniques were used in this paper, and some of the dated flowstones are less than 10 kyr old. In addition, this approach was chosen because younger sediments also exist in other parts of the cave system, which we are dating by 14C. This will facilitate comparisons in future studies too.

*3) ESR/U-Th a) This section is heavy and hard going, even for a specialist reader. It needs to be shorter, clearer and have a better introduction – even just clearly explaining what ESR is measuring and what the advantage is to combining ESR with U-series.*

We have rewritten this section to better explain how the ages have been obtained, and also to explain how common model parameters for the environmental dose rate were established.

*b) The final ages are presented rather confusingly – again there needs to be some final synthesis: how old does the ESR suggest the fossils are?*

This has been rewritten and made much more explicit.

Reviewer #3 Minor Comments:

*1) The University of Melbourne is abbreviated (by itself) to UoM, not Melbourne University (MU).*

Corrected.

2) A summary table of all the ages (not age data) will help pull everything together and allow for a comparison both between the blind duplicates of the same methods and between the different methods.

Such a table was included as Figure 14.

*3) The text on a number of the figures (1, 3 and 10) is very small in places and will be tiny in the final typeset paper.*

All figures have been checked and corrected to make sure font sizes are of sufficient size.

*4) Figure 1 has the words 'Relative height (m), cave floor' twice.*

Corrected.

*5) It is useful to see all the field photos of the U-Th flowstone samples but these all need proper scale bars. So does panel C of Figure 3 and all of Figure 4.*

Corrected.

6) Figure 8's caption is rather confusing, a little bit more introduction.

Corrected; more explanation has been added to the text.

*7) There appears to be some confusion over the caption for Figure 9 total of three captions are given, two different ones for Figure 9 and the caption from Figure 10 as well.*

Corrected.

Reviewer #4:

*[…] My more detailed commentary is noted below, but I do believe there are some more major issues that need to be addressed, even if most have to do with dates not associated with the hominin-bearing Unit 3.*

*1) They need to provide more sufficient evidence for the "time-transgressive" nature of Unit 1. All Unit 1 ages are younger than Unit 2. Proposing that Unit 2 (and 3) is composed of reworked Unit 1 is a sedimentological hypothesis (not a geochronological result) that needs to be tested.*

*I think they also need to be more clear on the reworking of Unit 1 forming Units 2 and 3 and reworking of Unit 2 forming Unit 3. Is this supposed to be happening within the Dinaledi Chamber? Outside of the chamber then being transported in via the debris cone?*

These are very valid concerns and we have tried to address this as follows:

A) We realise that the time-transgressive nature of Unit 1 can be somewhat confusing, more so because it includes different facies and erosional remnants that have been preserved at different topographic levels in the cave. To make things simpler to understand we have subdivided unit 1 into three facies:

- facies 1a & facies 1b which is the laminated orange mudstone found as erosional remnants along the cave floor; this unit was deposited as the combined result of mud settling in stagnant water (facies 1a) and the occasional influx of coarser material due to water flow and water pooring in from above (facies 1b), presumably during flood events (as described in Dirks et al., 2015, 2016).

- facies 1c, which is a new facies that we have introduced to explain why some Unit 1 sediments are probably older than Unit 2. Facies 1c also consists of orange laminated mudstone, similar to facies 1a, with the one important distinction that this unit accumulates on ledges and in cracks higher up in the cave and that it has been interpreted to result from mud carried in through cracks and fractures by water that slowly seeps along cave walls.

In the paper we have treated the three facies as if they are all part of the same stratigraphic unit (they have the same composition and share a high content of characteristic, orange cave mud; Dirks et al., 2015). However, this clearly causes confusion with the reviewers who question how Unit 1 sediments that appear to be younger than Unit 2 (and physically lower down in the cave stratigraphy) can possibly be incorporated in Unit 2. By defining facies 1c, we can make it explicit that this component of Unit 1 occurs as erosional remnants in fractures and on ledges throughout the cave including physical positions above Unit 2 at the entry into the Dinaledi chamber; they can therefore contribute to unit 2.

B) Note that the explanation would have probably been easier to read if the oldest unit was called Unit 1, the second oldest Unit 2 etc. However, to prevent inconsistency, we follow the Unit numbering/descriptions of Dirks et al. (2015), which were based on lithostratigraphic arguments and were done before any geochronological constraints were available.

C) The text and figures been reworked to reflect the above changes

D) Extra sections have been added as Figure 2, to further illustrate some of the complex stratigraphic relationships that exist in the cave.

*2) In the original Dirks et al. (2015) paper, they note that their flowstone samples were unsuitable for dating, yet now they have all sorts of dates on the flowstones. What changed?*

At the time of submission of the Dirks et al. (2015) paper, it was assumed (based on the descriptions of the fossils) that the deposit was older then the limit of c. 500 ka for U/Th disequilibrium dating. Therefore, preliminary work was carried out to assess the possibility of U-Pb dating, which allows dating of much older flowstone material. This did not look hopeful because of the presence of detrital material (and therefore abundant common Pb, which interferes with the signal) in most flowstones, as mentioned in Dirks et al. (2015). However, subsequent preliminary U/Th disequilibrium dating, which is much less critically affected by detrital material than U-Pb dating (because use is made of U and Th ratios and not Pb directly) revealed a much younger age range than originally anticipated, mostly well within the range of U/Th disequilibrium dating. Since this method is both more robust and more precise in the >500 ka range than U-Pb dating, no further U-Pb dating efforts were made.

We have now expanded the section in which we explain our dating approach and this has been made clear.

This reassignment of Unit 2 sediments to Unit 3 is not simply based on induration as is now more clearly stated in the text and discussed at length in the discussion under the heading “An updated stratigraphy for the Dinaledi Chamber”. We open the discussion by stating: “The ages presented here help to resolve outstanding questions about the stratigraphy in the Dinaledi Chamber, and allow us to define more closely the distribution of correlative stratigraphic units (Figure 13).” In other words, the reassignment of lithological units to their appropriate chronostratigraphic position was firstly based on the new age results.

It is normal practice in stratigraphy to re-define units based on chronostratigraphic constraints once these constraints have been obtained. We have obtained new ages that have allowed us to refine the lithostratigraphy presented in Dirks et al (2015) into a more precise subdivision of Units 2 and 3 as presented here.

3) The reassignment of material from under Flowstones 1c-e from Unit 2 to Unit 3 is one of the most important components of this paper. Without it, the majority of their age/depositional models fall apart. This needs to be addressed much more thoroughly, perhaps with additional sedimentological analyses. Simply that it is less indurated is not sufficiently convincing.

We also clearly state in the discussion section that: “However, the geochronology results now permit a better evaluation of the stratigraphy in the chamber, and it is evident that Unit 2 represents a significantly older stratigraphic unit that is restricted to deposits directly below Flowstone 1a, but not to those below Flowstones 1b-e. Flowstone 1c returns an age of ~242 ka (Table 1), indicating that the Unit 3 sediments below Flowstones 1b-e are probably significantly younger than sediments below Flowstones 1a (Figure 14). In addition, these sediments are less indurated and less weathered than the sediments below Flowstone 1a, and they contain *H. naledi* material (Dirks et al., 2015).”

In other words, unit 2 is not simply redefined on the basis of induration, but more importantly on the basis of new chronology and the observed absence of hominin bone material. We appreciate this comment and have explained all of this in greater detail, because it is of great importance to the paper.

*Also need some sort of explanation as why the description of the fossil from Flowstone 1A (Unit 2) went from hominin (Dirks et al., 2015), to not. It seems like interpretations are being changed to fit the dates as opposed to being based on the primary evidence/material (sedimentology, fossils) themselves.*

Nowhere in the text of Dirks et al (2015) was it stated that the fossil material below flowstone 1a was hominin even though Figure 3 in Dirks et al. (2015) has caused confusion because it shows a bone with the caption assigning all bones to Hominin, including a bone drawn under flowstone 1a. The assignment of the bone as hominin in this cartoon figure was unintentional (a mistake), and we are grateful to the reviewer to point this out to us. However, the relevant text on the matter from Dirks et al., 2015, does not make any claims that the bones below 1a are hominin, but instead they:

- differentiate between the deposits below 1a and those below 1b-e and indicate that bones weather out from the non-indurated parts of unit 2 below flowstones 1b-e

- never state that hominin bones occur in the consolidated Unit 2 sediments below Flowstone 1a. In all honesty, caves are really difficult to work in, and we admit that we did not get everything right the first time and did make several mistakes. We recognize that this caused some confusion by originally assigning all facies 2 sediments below Flowstones 1a-e to the same stratigraphic unit (Unit 2), and by making the mistake in Figure 3 in Dirks et al (2015). However, we are now trying to clear this up by more accurately differentiating between the indurated sediments below the older flowstone 1a (Unit 2) and the less indurated hominin- bearing sediments below the younger flowstones (Unit 3).

*4) Several critical errors in Table 1 (or the text) need to be fixed (see details below). The most important one is that sample RS18 is said to overly Unit 3 (hominin-bearing) in the text, but Unit 2 in the Table. This is a key date they use for anchoring the hominin fossils. Wondering if the change wasn't made in the table because they altered their unit assignments only after the dates were produced.*

The reviewer is correct in assuming that we altered the unit assignment after the dates came in as is usual in the stratigraphic allocations of sediments to units when chronostratigraphic constraints are acquired (as explained above). The real reason for the mix-up, however, is that an older version of the table was sent in by accident. The table has now been corrected and the text and figures are mutually consistent.

*5) Table 2–Table 7 need substantial revisions. I know the work was done in different labs, but that doesn't mean the tables should be completely different. They need to be totally reformatted to match each other so that comparisons can be made between the two labs. As is, they have different fields, different units, etc.*

This has now been done.

*6) The final age determination of the hominin fossils of 200-300 ka is not sufficiently explained. If based on ESR dates, then they need to more fully explain/justify why settled on 200-300 ka specifically as opposed to the ESR range of ~150-425 Ma? This appears to favor results of UW1810 and results from SCU/UOW (vs. GU/ANU). This seems rather arbitrary when in fact it is something that could be either modeled or calculated statistically. If it is just based off the younger age peak of 1767 (~200ka) and the older peaks of 1788 and 1810 (~275ka) in Figure 9, they should specifically state (and reference Figure 9).*

Good points, part of the problem with the paper was that the ESR results from the two labs, which are central to the age estimates for the fossils, had not been standardized; i.e. the two labs were using different model assumptions in calculating their ESR ages (i.e. their environmental dose rates). We have now standardized the modelling approach for ESR with fixed model parameters that have been varied these parameters between two scenarios to derive minimum and maximum age estimates. As a result the ESR results have been presented more clearly, ages and age ranges have now been better and more consistently constrained, and objective and transparent criteria (based on reported age ranges) have been applied consistently throughout the text and figures. We believe that has addressed the problems raised by the reviewer.

*Also, the yellow box in Figure 14 is not 200-300 ka, but more like 200-340 ka. Need to be consistent between text and figures. This is one of their main conclusions, yet not specifically supported.*

This has been done and this figure has been completely redrawn.

Reviewer #4 Minor Comments:

*- Line 88: Other papers format as refer to U.W. 101, not UW101. Change for consistency.*

Done.

*- Line 90: Lesedi Chamber papers says "some of which (facies and stratigraphic units) correlate to facies in the Dinaledi Chamber". Does not imply "sedimentologically distinct" as stated here.*

The fact that facies and units may correlate or look the same does not mean that the Dinaledi Chamber cannot be sedimentaologically distinct. For instance, certain types of rock, such as the orange mudstone will form in standing water in isolated pools within a cave chamber. Each pool may form very similar looking deposits, because the muds are largely derived from the insoluble product of dolomite dissolution during cave formation. So the units look similar and may even have formed at the same time. However, the cave chamber in which they formed may have been completely isolated. In this case isolation refers to the fact that the component of externally derived sediment contributing to the total sediment load is relatively small.

*- Lines 83 and 92: COH or CoH. Pick one.*

Done.

*- Line 167+: What specifically is used to correlate/group these discontinuous flowstone units across the chamber. You mention appearance is included (line 153), but no appearance is described. It seems that Group 2 is really just a group because it is not Group 1 or 3.*

We have tried to explain this a bit better; but in the end you are right, FG2 is everything that is not 1 or 3.

*- Line 194: Should be noted that in Dirks et al. (2015) Figure 3 the fossil below Flowstone 1a is attributed to hominin. What changed the identification?*

Accepted, a mistake was made in the figure in Dirks et al. (2015), but not in the text; see explanation above.

*- Line 200: How can Unit 2 sediments be derived both from local erosion/reworking of Unit 1 and as a debris cone below a vertical fracture system? They can't be both autochtonous and allochthnous in relation to the chamber. I believe this needs to be clarified. Are the Unit 1 clasts derived from Unit 1 equivalents outside the chamber?*

Valid concern and we have addressed this; see our detailed explanation above. We have changed the text to address this important issue by introducing facies 1c in Unit 1.

*- Line 213/218. This also goes for Unit 3. Is the Unit 3 material derived from reworked Unit 1 and 2 material outside the chamber, and brought in via the debris fracture/debris cone? Please clarify.*

As before, this has now been addressed.

*- Line 202: Why is there no mention of the baboon tooth in Unit 2 here?*

This has now been included.

*- Lines 203-260: This description doesn't seem to jive with what is depicted in Figure 7. The top cartoon should be more of a debris cone without a thick and even deposit running along the floor of the cave. Only AFTER the debris cone collapsed/eroded would Unit 2 material have spread/deposited along (evenly?) across the floor of the cavern. As currently depicted, if the debris cone in the top cartoon eroded, then the thickness of Unit 2 should be much, much thicker (not thinner) or you have to explain where all that collapse/eroded material went in the cavern (down the drains?).*

We have explained in the text in several places that the debris cone is eroded from below as material is moved down the drains; as this material is removed from below, the debris cone slumps inwards. We have further addressed this issue, which is a totally valid criticism, by referring to all the floor sediments as Unit 3 (as was originally done in Dirks et al., 2015), and by dividing Unit 3 into a lower (hominin free) and upper (hominin-bearing) member. The reason for changing this back to Dirks et al. (2015) is that all floor sediments are probably significantly younger than the Debris cone and are probably better grouped with Unit 3 than with Unit 2.

Unit 2 near the entrance is >780 ka old (from paleomag), and started eroding after formation of Flowstone 1a came to an end, which could be as late as 437 ka 9the lower age limit of Flowstone 1a). It is this erosional process of the debris cone, probably driven by sediment moving down the drain that causes the material to spread out along the floor.

We have updated the text and figures to reflect this better.

*- Line 211: It would be helpful to rephrase this sentence a bit to note how Unit 3 is differentiated from Unit 2. As is, they sound the exact same except for lithified (Unit 2) vs unlithified (Unit 3). Is that the only difference? Is that sufficient to re-assign the material below FS1c from Unit 2 to 3?*

As explained above. We come back to this issue extensively within the discussion section, but we have now clarified the issue up front as well by adding to the definition of Unit 2 as follows:

“however, the revised definition of Unit 2 accounts for the much more indurated nature and distinctively darker colour associated with deposits under Flowstone 1a, as well as the absence of hominin fossils, and the new geochronological constraints presented below”.

*- Line 218: Along with Figure 7, is the interpretation then that the hominin remains were brought in along with the sediments from outside the chamber and deposited as part of the debris cone? Please clarify.*

This issue has been discussed (and clarified) at length in Dirks et al. (2015), and a reference to this should suffice, as it is immaterial to the core aims of this paper. However, we do explain that fossils came in through the same entry route as the sediments that contributed to the debris cone as shown in Figure 8.

*- Line 246: With other macro-fossils now identified in Unit 2, what are the implications for chamber accessibility in the past? (Especially as there is little Unit 2 exposed, but it has fossils.)*

No difference to earlier interpretations; see Dirks et al., 2016 for a more extensive discussion of this. Again, this is an interesting issue, but not the subject of this paper.

*- Line 260-263. Need to provide some supporting evidence that Unit 1 was deposited during/after Unit 2 through modern times. Similarly, need to provide evidence that Unit 2 sediments are older than most of the Unit 1 deposits presently observed (no dating results presented at this point). What precludes it from being older? The only evidence provided (reworked into Unit 2) is the only concrete evidence provide for the sequence of the two units (and indicates older).*

Hopefully we have explained things more clearly by introducing facies 1c with an extended explanation of the stratigraphy in the chamber, and our sampling strategy. Of course, the data presented as part of this study will provide the age constraints that the reviewer is asking for here.

*- Line 294: Should include or refer to a Figure 2 photo(?) showing FS2 directly overlying U3 and hominin fossils.*

Okay.

- Line 297: "fossils entered" suggests they walked in along with the deposition as opposed to being deposited/reworked along with Unit 3. Perhaps better to say they were deposited or transported.

We disagree with this. “Entered” is an interpretation neutral term and can mean many different things. We intended it to be that way because Dirks et al., 2015 suggest that the fossils may have been carried in, dumped in or even walked in. Dirks et al., 2015 excluded the possibility that the fossils were transported in based on good sedimentological evidence, and nothing we have seen since has given us any reason to believe differently.

- Line 317: Dirks et al. (2015) noted that flowstone samples included a detrital component derived from associated muds in all tested pilot samples which confounded U-Pb dating because of the high and isotopically variable background of common Pb it carries. But now they are apparently all suitable? Need an explanation as to why they were undatable in the last publication, but are now magically all datable.

See our explanation under point 2 above. We have also expanded the text to make this more clear.

*- Line 332: Text says RS5 sample is from FS1, Table 1 says FS2. Which is it?*

It is FS1; this has now been updated.

*- Line 337+: Text uses Units, Table 1 uses Facies and they are not comparable! JCU/RS18 overlies Unit 3 (text) or Facies 2b (table)? This perhaps the most critical date as they argue it as the minimum for the fossils, but there is an error either in the text or the table.*

This has been corrected.

*- Line 368: Should state what the teeth are (e.g., upper premolar of H. naledi) either here or where each sample is discussed shortly later. Not just refer to figures that don't contain the information either (e.g., Figure 5, Figure 8). Shouldn't have to wait until materials section at the end to get some basic information.*

This has been corrected.

*- Line 375: Why are Table 6 presented before Table 4 (line 444)? Need to reorder.*

This has been corrected.

- Line 379: Needs a brief explanation/justification as to why U-Th ages should be regarded as minimum.

Explanations have been included, and a much longer section on how to evaluate the different dating techniques has been added to the discussion.

*- Line 426: There is no data on Sample 1841 in Table 7.*

This has been corrected.

*- Lines 427-435. Why are Sample 1767 and 1810 (and the age of Naledi in relation to it) being discussed under Sample 1841? This should have a different header.*

This has been corrected.

*- Line 493: Should provide radial plots for the samples for evaluation/visualization of the data.*

We don’t have these data.

*- Lines 507-511: Even if one agrees with the time-transgressive argument of Unit 1 (and that's a big if), these are the ages you have at the moment. Similarly, they still need to be reconciled with the flowstone ages that cover them, which is brushed aside here.*

Valid point, we have rewritten the text to address this in more detail.

*- Line 572/573: This totally ignores the results of RS23 from the BASE of FS1a which has a date of 683-449 Ka and is not compatible with the pmag of older than 780ka. The authors focus on the top sample RS22 as it fits their pmag, but seem to selectively ignore the data that doesn't fit. This was not discussed in the paleomag section and it is totally omitted here.*

No, this is a misunderstanding, we actually did say that the dating sample corresponds to the mid-point (i.e. Phase B) in the Paleomag sample (the various layers in the flowstone taper out laterally). In the description of the palaeomagnetic sample we say towards the end of the description: “Phase B and Phase C flowstone correlate with samples RS22 and RS23 that were collected for U-Th dating.” We have rewritten the text and figures to make this clearer and say this earlier. We have also added a new Figure 2 to show these relationships more clearly.

*If RS23 is not from the base, or is from higher up and in the Normal section of FS1a, then that needs to be properly documented here. In Figure 13 they state RS23 as "mid", but in Table 1 it is "base" and in the description of the sample (Line 912) and Figure 2 it is at the base.*

It was not intended to be confusing, but I can see it is. The paleomag sample was taken at a different spot from the U-Th sample and further back (i.e. closer to drip point source – see our new Figure 2) along flowstone 1a. The lower unit A measured in the paleomag analysis tapers out along the dip direction of the flowstone (as can be seen in Figure 13), whilst units B and C continue. Thus, the base of flowstone 1a where the sample for U-Th dating was taken corresponds to the mid-section of flowstone 1a in the place where the paleomag sample was taken. We have now changed the text to better explain this.

*- Lines 568-595: It seems like the determination to reallocate the sediments associated to Flowstones 1c-e from Unit 2 to Unit 2 was based more on the geochronology results than sedimentological analysis. Different levels of induration could be the result of other processes besides being different depositional units. It seems like they are changing their interpretation post-hoc to fit their new ages rather than based on the actual sedimentology.*

As explained above.

*- Lines 604-607: Their argument for Unit 1 to be time-transgressive is not very convincing. All 3 of their samples from 2 different parts of the cave say Unit 1 is younger than Unit 2. It doesn't match their interpretation of Unit 2 being reworked material Unit 1 so they propose that it is time-transgressive, yet they provide no data to support this. Seems that this hypothesis could be tested by additional sampling to find pieces of reworked Unit 1 in Unit 2 (or 3) that are suitable for OSL analysis. If dramatically older, then I'd believe their conclusion. If not, then they need to seriously re-evaluate their current ages of either Unit 1 or Unit 2.*

No, we clearly state that the fragments of Unit 1-like mudstone were derived from an earlier deposit of laminated orange mudstone that was similar in composition to the Unit 1 sediments; i.e. we acknowledge that earlier deposits of Unit 1 like material must have been present in the cave, and have contributed to the accumulation of Unit 2. In other words we know that the ages for Unit 1 as currently preserved in the cave are younger that the ages of Unit 2, and we also know that there are other deposits of unit-1 like material higher in the cave that may be the source for the fragments in Unit 2, and that we have not been able to test in this study (please note that the fractures in which these sediments occur are extremely hard to access). We have now made this more explicit in the text and we have added facies 1c to specifically reflect these remanants of Unit 1 higher in the cave. The new Figure 2, hopefully also helps in explaining this better.

Regarding the hypothesis testing, we have established that Unit 2 is older than 780 ka. It would be impossible to sample Unit 1-like fragments that are incorporated in Unit 2 for OSL analysis; these fragments are much too small for that.

*- Line 640/641: Again, Table 1 says RS18 overlies Facies 2b (=Unit2?), not Unit 3. Perhaps it was Unit 2 until they decided to change it Unit 3 because of the age (i.e., changing based on age, not sedimentology).*

Corrected. Also, getting an age to establish stratigraphy is a normal process in stratigraphy.

*- Lines 644-647: This is confusing sediment age with depositional age. Reworking sediment doesn't change the age of the sediment, it simply reflects a different period of deposition. It would be more appropriate to say that Unit 3 is a reworked unit and that different deposits of it in the chamber may have been deposited at different times. However, it is also worth noting that although H. naledi may have been deposited over an extended period of time, they also may not have. There is not sufficient evidence presented to say one way or the other.*

We have presented the arguments why we think *H. naledi* may have been deposited over a period of time, but we have not expressed an absolute view on this and I think we are very careful in our wording regarding this; I think this is clear in the text where we state: “Staining patterns on bone fragments, skeletal element representation, and the fact that bones can be seen to weather out from erosional remnants of Unit 3, indicate that part of the fossil assemblage has been reworked. The presence of well-articulated remains in the excavation pit, away from the chamber entrance, indicates that some of the remains entered the cave intact. The mixed taphonomic signature suggests that fossils entered the cave over a period of time, which is minimally assumed to be during deposition of Unit 3, and before deposition of Flowstone 1c. Fossil entry may have continued as sediment accumulations of Unit 3 near the entry shaft were reworked and redistributed along the cave floor (Figure 7).”

*- Line 657: This is cherry picking the U-Th tooth data as it is only 1 (the oldest) of 14 results. Should just stick to the ESR data.*

No this is not cherry-picking, but is a function of how the combined U-Th-ESR technique operates. Note that the U-Th ages in the teeth reflect U-uptake events (which are an integral part that dictates the models applied for the ESR dating); they do not reflect a true age for the teeth, but provide an estimate for the minimum age.

Various uptake events can, and are likely to overprint each other, so therefore we look for the oldest uptake event we can find in the teeth, which in this case is the 200 ka event. It just so happens that we have found one spot in the teeth that records this old age reflecting early U-uptake; this age will, therefore, set a likely minimum age for the fossils (compare this to detrital zircon ages where the youngest zircon age will be closest to the actual depositional age of the sediment even if older grains are far more numerous!).

*- Line 719: Why 200-300 ka? If based on ESR dates, then need to explain/justify why settled on 200- 300 ka specifically as opposed to the ESR range of ~150-425 Ma? This appears to favor results of UW1810 and results from SCU/UOW (vs. GU/ANU). This seems rather arbitrary when in fact it is something that could be either modeled or calculated statistically. If it is just based of the younger age peak of 1767 (~200ka) and the older peaks of 1788 and 1810 (~275ka) in Figure 9. Then should specifically state (and reference Figure 9).*

See our explanations for this above. This has been extensively reworded.

*Also, the yellow box in Figure 13 is not 200-300 ka, but more like 200-340 ka. Need to be consistent between text and figures. This is one of their main conclusions, yet not specifically supported.*

This has been rewritten and corrected.

*- Lines 729+: Appreciate the detailed materials and methods.*

*- Figure 1. Why not use the update surface map from the Elliot et al. submission that also includes the location of domal stromatolites, chert markers and mylonite markers? Should at least use consistent maps between papers.*

Done.

*- Figure 12'm assuming that this wasn't actually analyzed as Phase A and B are only a cm thick yet samples were 2-2.5cm cores.*

You can see from 12A that there is sufficient material for analysis. 12B is an off-cut for the close-up photo.

*- Table 1: It would be helpful to sort the table by flowstone group (or possible age), not HW sample number to compare proposed clusters of flowstone ages easier. Blind duplicates of the same sample or base/top of the same flow shouldn't be rows apart. Also, text only refers to Units, table refers to Facies, and they are not comparable! Why are RS10 and RS5 listed if there is not data? Should delete (similar to no HW to pair with RS1).*

Table has been cleaned up.

*Text says RS5 sample is from FS1, Table 1 says FS2. Which is it?*

Corrected.

*Text says RS18 overlies Unit 3, Table says RS18 overlies Facies 2b (=Unit2?). Which is it?*

Corrected.

*- Table 2: Need to be formatted exactly the same (same fields, units, etc.) for a proper comparison.*

Corrected.

*Data for RS1 is missing from Table 3. As it is a blind duplicate it needs to be included so the blind comparison can be compared.*

The blind duplicate was only done in the lab external to JCU (so that we could check the results from that lab). Because this is a double blind, the results from the external lab at UoM serve as a check on the JCU results.

*Sorry, but the different formatting of Figure 2–Figure 7 is sloppy.*

Corrected.

*- Table 4 & 5. Again, why are these tables formatted differently and with different information.*

Corrected.

*Also, why are they presented before Table 5, but discussed/referenced after them in the text?*

As explained above.

*- Table 6 & 7. Again, why are these tables formatted differently and with different information?*

As explained above.

*- Table 6 is 4 tables, not 1. What do * represent? Needs to be explained.*

Corrected.

*- Table 7: Missing data for Sample 1841.*

Corrected.

Reviewer #5:

*[…] Dating that places these specimens at 200-300 ka suggests several scenarios.*

*1) The dating is correct.* H. naledi *originated at some unknown time, perhaps 1-3 Ma, and survived until the Middle Pleistocene, in a similar fashion to the proposed survival of* H. floriensis*. Note that* H. floriensis *was isolated from other species;* H. naledi *would have co-existed with possibly several other species of hominins. In itself, that is plausible; we are learning that the existence of a single hominin species is quite recent. However, this hypothesis conflicts with the evolutionary assumption that more primitive species cannot compete with more advanced, and thus to be convincing it would need to be bolstered by some evidence of an advantage that compensated for primitive features or an analysis that these features were irrelevant.*

Interesting, but this comment is better referred to the Berger et al paper.

*2) The dating is correct but the fossils are reworked from some other deposit. This hypothesis has been rejected on the grounds that there is no known entrance into the cave other than the one found by this group. It is further challenged by the sheer number of finds; survival of so many pieces is more plausible over 300 ka than 1 Ma.*

Note that this argument has been principally rejected based on sedimentological grounds; there is no evidence of (a large amount of) coarse-grained material being transported into the chamber from outside! This is made clear in the text.

*3) The dating has been done with great precision and care, but there is a factor of which the analysts were unaware, or perhaps that could not be measured.*

*Reviewing a 96-page manuscript means that the reviewer must concentrate on a portion of it. Of the various dating methods used, only the combined ESR-U series method deals specifically with the fossil ages. It seems reasonable therefore to focus on the details of this method in commenting on the manuscript.*

*Let us consider the scenarios in reverse order. By definition, something that cannot be measured cannot play a role in calculations. Of the known factors, the only possible problem with the dating lies in the evaluation of the external dose rate. The cave has experienced significant transformations over time. The external dose rate is as much as 80-90% of the total dose rate for the hominin teeth. The text notes that sediment radioisotope contents are in Table 8; that table lists sediments from Unit 1 for OSL, not Unit 3.*

OSL was done on different sediments (Unit 1) compared to the sediments in which the teeth were embedded (Unit 3); this is clearly explained in the text. The average U, Th, K values of Unit 3 are listed in Table 4.

*A reference is made to 'average' sediment values; it would be of interest to say how many samples were used.*

This is explained in the methodology section and shown in Figure 10.

*Also, the values for Unit 2 are very similar to those for Unit 3, reinforcing the suggestion (p. 8, line 213) that Unit 3 reflects reworking of Unit 2. On the whole, there seems little leeway for significant changes in external dose rate.*

The reviewer is correct that there is little leeway. However we have significantly expanded on this section and have done additional modelling to make sure we fully understand the issues associated with the external dose rates (see the detailed updates in this regard in the results and methodology sections, and the introduction of maximum and minimum age scenarios).

*What about reworking? To say that 'there is no evidence that the present entrance has significantly changed' is not the same as saying that there is evidence it could not possible have changed, the latter being an unreasonable standard, of course. Internal reworking – where the fossils were originally deposited in a thicker Unit 2 and eroded onto a deflational surface in what is now Unit 3 – would not be likely to change the age, at least not from 200 ka to 1 Ma, since as noted in the preceding paragraph the enclosing sediment remains essentially the same.*

Exactly.

*That leaves the first scenario, which is one that is not dealt with in this manuscript. It will be interesting to see what sorts of interpretation are made along that line.*

This however has been dealt with in the accompanying Berger et al paper. We have now included some of the main conclusions from that paper in this paper as well in response to some of the other review comments.

*Therefore this manuscript needs only minor revision. The authors should explain more fully the number and locations of sediment samples, preferably adding yet one more table instead of referring to Table 8 (p. 38, line 1203).*

With the rewrite of the ESR section this has been done.

Reviewer #5 Minor Comments:

*Page 15, line 447: 'did' should be 'was'. Page 41, line 1296: 'enable one to differentiate'.*

*As noted in the review, the authors have not submitted sediment data for ESR dating that indicates how many sediment samples were used for each sample and where they came from.*

Corrected; this has been made clearer.

[Further revisions were requested before acceptance.]

In making further revisions to the manuscript we have reverted back to a more detailed description of Facies and Units that allows us to better address the concerns raised by the reviewers. Initially we had tried to keep things relatively simple by omitting detail on facies and Unit descriptions, but that results in over-simplification as correctly pointed out by the reviewers. We have explained our corrections in more detail below.

*In addition, there was discussion among the reviewers during the consultation process concerning the last line of the Abstract and the penultimate paragraph of the Discussion (that begins on line 1009), which were viewed as unnecessarily critical of prior authors' work and of the challenges in accurately dating hominin fossils.*

We have changed the wording to appear less critical.

*In addition, at the end of the Abstract, it may be better to simply state that "These dating results indicate a much younger age for the* H. naledi *fossils than have previously been hypothesized based on their morphology."*

This has been done.

*Technical Points:*

*1) Lines 175-178: The addition is appreciated, but a sentence at least on how the material "accumulated inside fractures and along ledges" needs to be added. I would also drop the last sentence in this paragraph and move it to the discussion of Unit 1 (out of place here).*

We have followed both recommendations; this has now been done.

*2) As a result of the revised units and stratigraphy (esp. "facies 1c" and "informal upper and lower members of Unit 3), the authors have now introduced inappropriate and inconsistent terminology. This is not just an issue of semantics.*

*Lines 217-229: The introduction of facies 1c is problematic, not as a subunit in general, but as a subfacies in particular. The confusion and misuse of facies happens often and here is a chance to correct it. Quoting Maill (facies guru), in sedimentology, "lithofacies are defined on the basis of its distinctive lithologic features, including composition, grain size, bedding characteristics, and sedimentary structures. Each lithofacies represents an individual depositional event. Lithofacies may be grouped into lithofacies associations or assemblages, which are characteristic of particular depositional environments". Thus, the basis of different subfacies here based on "relative topographic position in the cave" (i.e., facies 1c) is not valid criteria. Specifically, facies 1a is fine (non-lithified, horizontally laminated, orange mud). Facies 1b is fine (laminated mud with fine sand and rodent remains) and distinct from 1a. Facies 1c is "similar in appearance and composition to the laminated, muddy sediments of facies 1a, but they occur "in different places". This is not a properly differentiated as a separate facies. Facies 1a in a different location in the cave topography is still facies 1a unless it can be differentiated appropriately. There are many examples in sed/strat where the same/similar facies are repeated multiple/different times in a succession. This is again an issue at the end of the paragraph where the facies differ in "relative age and position within the cave chamber", which is not a characteristic of facies. Facies 1c needs to be differentiated from 1a on the basis of composition, grain size, bedding, or sedimentary structures to be valid.*

We have retained Facies 3c, but we have expanded the descriptions of the different facies and we have included bed-form characteristics that clearly differentiate Facies 1c from Facies 1a and 1b. We now state: “Deposits of Facies 1c are similar in composition to the laminated, muddy sediments of Facies 1a, but they are distinguished from Facies 1a by having non-horizontal (including subvertical) laminations with limited lateral continuity and curved surfaces. Sediment of Facies 1c is consistently observed along chert ledges, and in solution pockets and fractures in the Dinaledi Chamber, and are interpreted to have originated as localised accumulations of muddy material that entered the chamber through thin (mm-scale) fracture networks. The mud is mostly the product of the cave formation process, representing the insoluble residue left over when cavities develop via dissolution in dolomite (Dirks et al., 2015).”

*Lines 270-276: Unit 1 is divided (incorrectly) into 3 subfacies. Unit 3 is divided into 2 informal members. This inconsistency is confusing and unnecessary. Also, members are formal lithostratigraphic units of formations. It is OK to provisionally have informal members, but this is almost always in the context of a sequence where there are other formal members (as part of a formation), which this sequence does not have, nor is it part of a defined formation. For example, you could have formal Members A, B, C, D and then informal members E and F, but randomly throwing in informal members into this context is exceptionally odd. Ironically, the authors could subdivide Unit 3 into facies 3a and 3b as the presence/absence of fossils is a characteristic that can be used to differentiate facies. Also, what happens if they do ultimately find H. naledi fossils in the "lower member"? Something that would easily fix all of this is if they just used informal sub-units for all (e.g., 1a, 1b, 1c, 2, 3a, 3b), that way they can be consistent and define them as they wish as units have no formal definition while facies and member do. The fact that all of the sedimentary material in the cave appears to multiple phases of reworking the same material over and over suggests everything is genetically related and easily dealt with as a series of related units.*

We think it is better not to introduce Facies 3A and 3B for the reasons mentioned by the reviewer: there is a possibility in future that *naledi* material could be discovered in the Unit 3 sediment, which now appear devoid of fossils. We have now done the following:

As explained above we have retained Facies 1A, 1B and 1C,We have removed all reference to members, and instead have called the upper and lower part of Unit 3, sub-unit 3a and sub-unit 3b as suggested by the reviewers.Likewise, we have introduced two sub-units to describe Unit 1 sediments along the cave floor.We have made it clear that Facies 1c sediment, which accumulates along ledges and in cavities along the cave walls and along the entry zone, could have been part of a further sub-unit of Unit 1, which we hypothesize, may have existed in the cave.We have changed the text and figures throughout to be consistent with these new subdivisions.We have been very careful in our wording not to be too definitive about the provenance of Facies 1c material that may have ended up in Unit 2.

*Lines 304-315 (and elsewhere): Again, until other ages are generated for facies 1c subfacies or its stratigraphic relationship/contacts with other units can be observed/demonstrated, the time-transgressive explanation for Unit 1, and the relationship between Units 1 and 2 are just a hypothesis. Similarly, their assertion that facies 1c (as opposed to 1a or 1b) must have been the unit that provided the source material to Unit 2 (now 3?) is not confirmed by evidence and, again, is just a hypothesis. The authors are very definitive about their interpretation: "it is clear that older laminated orange mud deposits ascribed to facies 1c in Unit 1 occurred above Unit 2 and must have provided the source material now found as much clasts within Unit 2". As there are no dates for facies 1c, and as facies 1c is lithologically the same as 1a, this is a hypothesis, not an unquestionable fact. I have no problem if this is presented as a hypothesis: it might even be likely, but it is not definitive and I think the authors need to back away from such a claim.*

This wording has been changed to reflect that this is an interpretation and not definite.

*The authors note the challenges of OSL dating these deposits, yet rely almost exclusively on the OSL ages for a "young" facies 1a/b argument. An alternative explanation is that Unit 1a/b is older than Unit 2 and the OSL dates are incorrect (aliquots as opposed to single grain). Or perhaps the 1a looking material was brought into the chamber from elsewhere where similar depositional environments also produced laminated orange mud. This all needs to be demonstrated or proposed as a hypothesis (which can be tested by more/future dating even if far down the line).*

Point taken and the wording has been changed as explained above.

*3) Somewhere, the authors should to be more specific of the possible age of H. naledi from this chamber, versus the best fit age of H. naledi specimens dated here. For example, flowstone group 1c does provide a minimum age for the specimen that is stuck underneath it, and the ESR ages do provide a maximum age for the 2 teeth. The authors suggest that the fossils entered the cave over a period of time and may have continued during the deposition of upper Unit 3 (lines 299-303). As such, I have no problem with the authors saying those 3 particular fossils have a best fit of 236-335 ka, but does anything preclude the other hominid specimens from being associated with older or younger parts of upper unit 3 with the potential to span approximately 90ka (min of above RS1/15 above U3) to 414 ka (max of OSL5)? At minimum, a red dashed vertical line should be added to Figure 14 at ~414 ka for the oldest possible age to match the minimum line at ~100 ka.*

In the abstract and in various parts of the text we explain that we have ages that constrain the fossil-bearing unit (sub-unit 3b in our new terminology), that we have ages for 3 fossil teeth themselves, which provide a tighter age constraint and that within our preferred age range for the fossils we think that the maximum age model is most likely considering the high degree of Rn loss in the sediments. In other words, we are quite specific on how we think the results should be interpreted, and in doing so we closely followed recommendations made by the reviewers during the earlier review round. To address the above comment we have adjusted the text, and we have added further detail to Figure 14, following the recommendation made.

*4) Line 465: This sentence is confusing as presently constructed. It seems to report that something is both younger than unit 1 and older than unit 1. It would be better to write "younger than ages derived from facies 1a, but slightly older than ages derived from facies 1b in this location."*

Corrected.

*5) The U-Th ages must be quoted as just ka, using kyr BP is a radiocarbon convention and does not apply to U-Th - there is no industrial carbon adjustment to U-Th ages, so to quote them as kyr BP is misleading and incorrect.*

Reference to kyr BP has been removed from the text.

*6) In Table 4, "Dose" should be "De" to prevent confusion with subsequent discussion of artificial doses given to samples. Thicknesses are μm, not mm and according to the table, 100 μm was removed from the dentine and the sediment, not enamel and dentine, due to misplaced headers. The removal from dentine was not mentioned in the methods section.*

Mistakes in this table have been corrected. No material was removed from sediment; this line was mistakenly duplicated in the table.

*7) There was no dentine on the baboon tooth (line 618) but would a modern analogue suggest that there should have been dentine on at least the inside? It probably doesn't matter since the dentine survival time would be a relatively small portion of the entire burial time, but the authors might comment on this.*

We don’t know; there is no visible evidence for dentine, and any statement on this would be a guess that cannot be easily tested. Also a small amount of dentine if it had existed would make little difference regarding the result. We prefer not to speculate here.

*8) Figure 8: With the addition of another unit, it is very challenging to differentiate the little sliver of Unit 2 (below Flowstone 1a) and Unit 3 upper because of the similarity in color, especially in the upper panel. It would be helpful to address this. Also, given the revisions, what now is the explanation/model for the deposition of Unit 2? Virtually all of Unit 2 is now assigned to lower Unit 3. In the previous submission, Unit 2 was under the flowstone and underneath the hominid fossils on the cave floor. Now it is stuck up under the uppermost flowstone and nowhere modeled to be anywhere on the cave floor, just completely washed away(?). While the authors do touch upon this in the text, they should consider adding back in a revised panel to explain Unit 2 a bit better.*

We have revised Figure 8 to reflect this. The issue of reworking is explained in quite a bit of detail in the text as well.

[Further revisions were requested before acceptance.]

[…] Perhaps one of the major continuing complications and challenges in these revisions is the effort to maintain consistency with the earlier descriptions in the Dirks et al. 2015 paper. Simply stated, the reviewers feel strongly that the use of facies as units in Dirks et al. 2015 was incorrect; updated stratigraphy needs to be used in the present paper alongside a simple, clear statement of change from the original interpretations. Editorially, we agree that the present paper should standalone and use the most up-to-date, appropriate descriptions and approach. Doing otherwise is introducing complication and would be perpetuating confusion.

In making further changes to the manuscript we have referred directly to the detailed recommendations made by reviewer 4 in review round two. This means that we have now exactly followed the reviewer suggestions to abandon the use of facies and refer to all the stratigraphic entities as informal units and subunits. We have only retained mention of facies where we make direct reference to Dirks et al., 2015, and such references have been made explicit. Otherwise we have used Units 1, 2 3 and sub-units 1a, 1b, 1c and 3a, 3b throughout, as recommended. The distribution of these sub-units is illustrated in Figure 2. These changes mainly affect the section on ‘lithologic and stratigraphic context for dating’, which has been updated. The changes will make it much easier for people to read and understand.

As a point of correction, Dirks et al (2015) do not use facies as units. However, in response to reviewer 4, all of the previous concerns raised with respect the use of facies, units and sub-units have now been addressed.